# A single-cell time-lapse of mouse prenatal development from gastrula to birth

Chengxiang Qiu[1,18 ✉], Beth K. Martin[1,18], Ian C. Welsh[2,18], Riza M. Daza[1], Truc-Mai Le[3], Xingfan Huang[1,4], Eva K. Nichols[1], Megan L. Taylor[1,3], Olivia Fulton[1], Diana R. O'Day[3], Anne Roshella Gomes[3], Saskia Ilcisin[3], Sanjay Srivatsan[1,5], Xinxian Deng[6], Christine M. Disteche[6,7], William Stafford Noble[1,4], Nobuhiko Hamazaki[1,8], Cecilia B. Moens[9], David Kimelman[1,10], Junyue Cao[11], Alexander F. Schier[12,13], Malte Spielmann[14,15,16], Stephen A. Murray[2], Cole Trapnell[1,3,13,17] & Jay Shendure[1,3,8,13,17 ✉]

The house mouse (*Mus musculus*) is an exceptional model system, combining genetic tractability with close evolutionary affinity to humans[1,2]. Mouse gestation lasts only 3 weeks, during which the genome orchestrates the astonishing transformation of a single-cell zygote into a free-living pup composed of more than 500 million cells. Here, to establish a global framework for exploring mammalian development, we applied optimized single-cell combinatorial indexing[3] to profile the transcriptional states of 12.4 million nuclei from 83 embryos, precisely staged at 2- to 6-hour intervals spanning late gastrulation (embryonic day 8) to birth (postnatal day 0). From these data, we annotate hundreds of cell types and explore the ontogenesis of the posterior embryo during somitogenesis and of kidney, mesenchyme, retina and early neurons. We leverage the temporal resolution and sampling depth of these whole-embryo snapshots, together with published data[4–8] from earlier timepoints, to construct a rooted tree of cell-type relationships that spans the entirety of prenatal development, from zygote to birth. Throughout this tree, we systematically nominate genes encoding transcription factors and other proteins as candidate drivers of the in vivo differentiation of hundreds of cell types. Remarkably, the most marked temporal shifts in cell states are observed within one hour of birth and presumably underlie the massive physiological adaptations that must accompany the successful transition of a mammalian fetus to life outside the womb.

Since 2017, many studies have applied single-cell methods to characterize biological development at the scale of the whole organism[7–17]. Most such studies are time series, in which each embryo is analysed at one developmental stage—by profiling of transcription via single-cell RNA sequencing (scRNA-seq) or chromatin accessibility via single-cell sequencing assay for transposase-accessible chromatin (scATAC-seq)—resulting in a series of snapshots that can be pieced together, analogous to the single frames that are put together to create a film. Inevitably, there are trade-offs between the developmental span studied, the temporal resolution and the sampling depth of the snapshots taken. For example, 2 studies intensely profiled mouse gastrulation, together quantifying gene expression in 150,000 cells from more than 500 embryos spanning embryonic day (E)6.5 to E8.5[7,17], and another study profiled 2 million nuclei from 61 embryos spanning E9.5–E13.5[14]. We recently integrated such scRNA-seq datasets to

produce an initial tree of mouse developmental cell states spanning E3.5–E13.5[8]. However, early organogenesis was coarsely sampled (with 24-h intervals), and the remainder of prenatal development remained unsampled at the whole-organism scale, limited in part by the sheer number of cells.

## Ontogenetic staging

To progress towards a more comprehensive, continuous view of transcriptional dynamics throughout prenatal development, we sought to deeply sample single nuclei from mouse embryos precisely staged at 2- to 6-h intervals spanning late gastrulation (E8) to birth (postnatal day (P)0). In staging embryos, we distinguish between gestational age and developmental progression. Mouse gestational age, based on the observation of a vaginal plug for which noon on that day is declared

[1]Department of Genome Sciences, University of Washington, Seattle, WA, USA. [2]The Jackson Laboratory, Bar Harbor, ME, USA. [3]Brotman Baty Institute for Precision Medicine, Seattle, WA, USA. [4]Paul G. Allen School of Computer Science and Engineering, University of Washington, Seattle, WA, USA. [5]Medical Scientist Training Program, University of Washington, Seattle, WA, USA. [6]Department of Laboratory Medicine and Pathology, University of Washington, Seattle, WA, USA. [7]Department of Medicine, University of Washington, Seattle, WA, USA. [8]Howard Hughes Medical Institute, Seattle, WA, USA. [9]Division of Basic Sciences, Fred Hutchinson Cancer Center, Seattle, WA, USA. [10]Department of Biochemistry, University of Washington, Seattle, WA, USA. [11]Laboratory of Single-Cell Genomics and Population dynamics, The Rockefeller University, New York, NY, USA. [12]Biozentrum, University of Basel, Basel, Switzerland. [13]Allen Discovery Center for Cell Lineage Tracing, Seattle, WA, USA. [14]Max Planck Institute for Molecular Genetics, Berlin, Germany. [15]Institute of Human Genetics, University Hospitals Schleswig-Holstein, University of Lübeck and Kiel University, Lübeck, Kiel, Germany. [16]DZHK (German Centre for Cardiovascular Research), Partner Site Hamburg, Lübeck, Kiel, Germany. [17]Seattle Hub for Synthetic Biology, Seattle, WA, USA. [18]These authors contributed equally: Chengxiang Qiu, Beth K. Martin, Ian C. Welsh. ✉e-mail: cxqiu@uw.edu; shendure@uw.edu

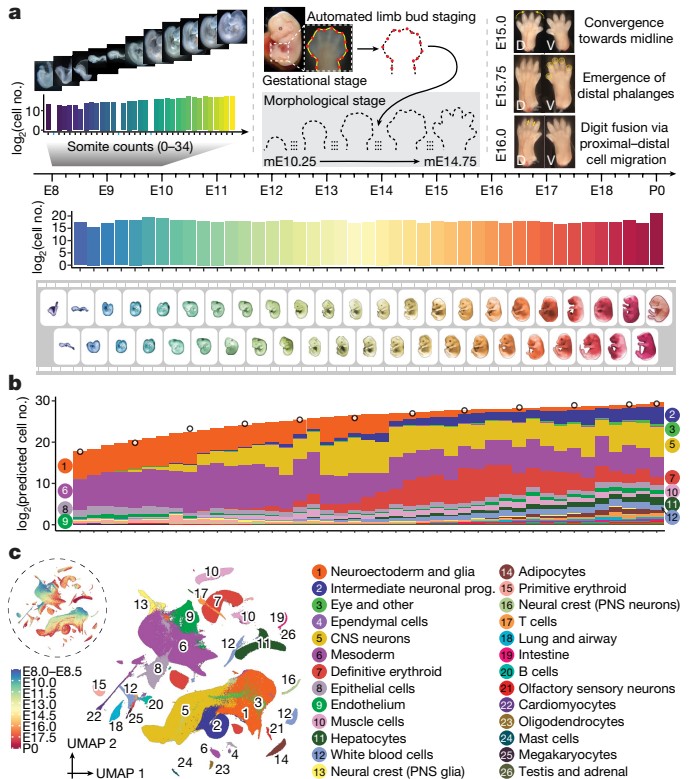

**Fig. 1 | A single-cell transcriptional time-lapse of mouse development, from gastrula to pup. a**, Embryos were collected and precisely staged based on morphological features, including by counting somite numbers (up to E10) and an automated process that leverages limb bud geometry (E10–E15) (Methods). Each embryo was assigned to one of 45 temporal bins at 6-h increments from E8 to P0, and to more highly resolved 2-h bins at earlier timepoints based on somite counts (0–34 somites). The first three bins (E8.0, E8.25 and E8.5) are combined. Embryos with somite counts 1, 13 and 19 are missing from the series (blue ticks in sub-axis). The number (log₂ scale) of nuclei profiled at each timepoint, is shown adjacent to the horizontal timeline, for 2-h bins (0–34 somites) for E8–E10 and for 6-h bins for E8–P0. **b**, Composition of embryos from each 6-h bin by major cell cluster. The y axis is scaled to the estimated cell number (log₂ scale) at each timepoint. In brief, we isolated and quantified total genomic DNA from whole embryos to estimate cell number at 12 stages (1-day bins, highlighted by black circles), and then predicted cell number at 43 timepoints using polynomial regression (Methods). **c**, Two-dimensional uniform manifold approximation and projection (UMAP) visualization of the whole dataset. The inset dashed circle shows the same UMAP coloured by developmental stage (plotting a uniform number of cells per timepoint). Colours and numbers in **b**,**c** correspond to the 26 listed major cell cluster annotations. Prog., progenitor.

E0.5, only loosely approximates the time elapsed since conception. Stochastic differences in the timing of mating or fertilization, together with genetic factors and litter size, can result in significant variation among embryos of identical gestational age[18]. Conversely, embryonic morphogenesis is highly ordered, reproducible, and inherently reflective of an embryo's developmental age with respect to absolute position within a morphogenetic trajectory and the dynamic progression of underlying cell states[9,19]. Therefore, we staged embryos by well-defined morphological criteria—for example, somite number and limb bud geometry—initially to 45 temporal bins at 6-h increments from E8 to P0 (Fig. 1a and Extended Data Fig. 1). From a total of 523 embryos staged at the Jackson Laboratory, we selected 75 for whole-embryo scRNA-seq, targeting 1 embryo for every somite count from 0 to 34 (2-h increments) and one embryo for every 6-h bin from E10 to P0 (Supplementary Table 1).

## Whole-embryo scRNA-seq

Flash-frozen embryos were shipped to the University of Washington, where they were pulverized and subjected to an optimized protocol for single-nucleus transcriptional profiling by combinatorial indexing[3] (sci-RNA-seq3). Sequencing data were generated across 15 sci-RNA-seq3 experiments and 21 Illumina Novaseq runs (Supplementary Tables 1 and 2). In total, 160 billion reads were demultiplexed, trimmed, mapped, deduplicated and grouped on the basis of constituent cellular indices. Following aggressive filtering of low-quality nuclei and potential doublets, the resulting cell-by-gene count matrix includes transcriptional profiles for 11,441,407 nuclei from 74 embryos spanning E8 to P0 (Fig. 1a and Extended Data Fig. 2a–f), 1% of which (somite counts 0–12) were previously reported[8]. On average, 154,614 nuclei were profiled per embryo (range 1,700 to 1.6 million; Fig. 1a and Supplementary Table 1).

This dataset greatly improves upon our previous single-cell atlas of mouse organogenesis[14] with respect to sampling depth (from 2 million to 11.4 million nuclei), profiling depth (median 671 to 2,545 unique molecular identifiers (UMIs) per nucleus), temporal resolution (24-h to 2- to 6-h intervals) and developmental span (E9.5–E13.5 to E8–P0). In performing quality control, we found that cells from the same or adjacent stages but profiled in different experiments were well integrated (Extended Data Fig. 2g–i). Furthermore, principal component analysis (PCA) of pseudobulked RNA-sequencing (RNA-seq) profiles resulted in a major first component that strongly correlated with developmental time (PC1 = 77%; Extended Data Fig. 2j). Ambient noise due to RNA leakage or barcode swapping was present at low levels (Extended Data Fig. 2k).

What kind of 'shotgun cellular coverage' of the mouse embryo are we achieving? Leveraging total DNA quantification of staged embryos, we estimate that the embryo grows 3,000-fold from E8.5 to P0 (210,000 to 670 million cells), with its cellular doubling time slowing from around 6 h to 1.5 days (Fig. 1b, Extended Data Fig. 2l,m and Supplementary Table 3). Thus, even with the many nuclei profiled here, our cellular coverage remains modest, ranging from 0.5-fold for early stages (summing 6 embryos, somite counts 7–12) to 0.002-fold immediately before birth (summing 6 embryos, E17.5–E18.75).

## Cell-type annotation

To get our bearings, we used Scanpy[20] to generate a global embedding of the 11.4 million cell × 24,552 gene count matrix, and annotated 26 major clusters on the basis of marker genes (Fig. 1b,c and Supplementary Table 4). As expected, cell clusters whose proportions decline over developmental time either stream towards derivatives (for example, neuroectoderm and glia to central nervous system (CNS) neurons and intermediate neuronal progenitors) or are displaced by functionally analogous but developmentally distinct lineages (for example, primitive erythroid to definitive erythroid). However, the resolution of these major clusters was somewhat arbitrary and affected by abundance. To balance the resolution, we performed another iteration of clustering and annotation, resulting in 190 labelled cell types (Extended Data Fig. 3 and Supplementary Table 5). These annotations are preliminary, and we welcome their refinement by the community.

We also performed deeper dives into the ontogenesis of the posterior embryo during somitogenesis, kidney, mesenchyme, retina and early neurons. These analyses, summarized below, illustrate the richness of this dataset and highlight opportunities for its further exploration.

## Posterior embryo during somitogenesis

Neuromesodermal progenitors (NMPs) are a population of bipotent cells with both neural (spinal cord) and mesodermal (trunk and tail somites) derivatives[21]. Towards extending our previous investigations of NMP heterogeneity[8], we re-embedded 121,118 cells from all

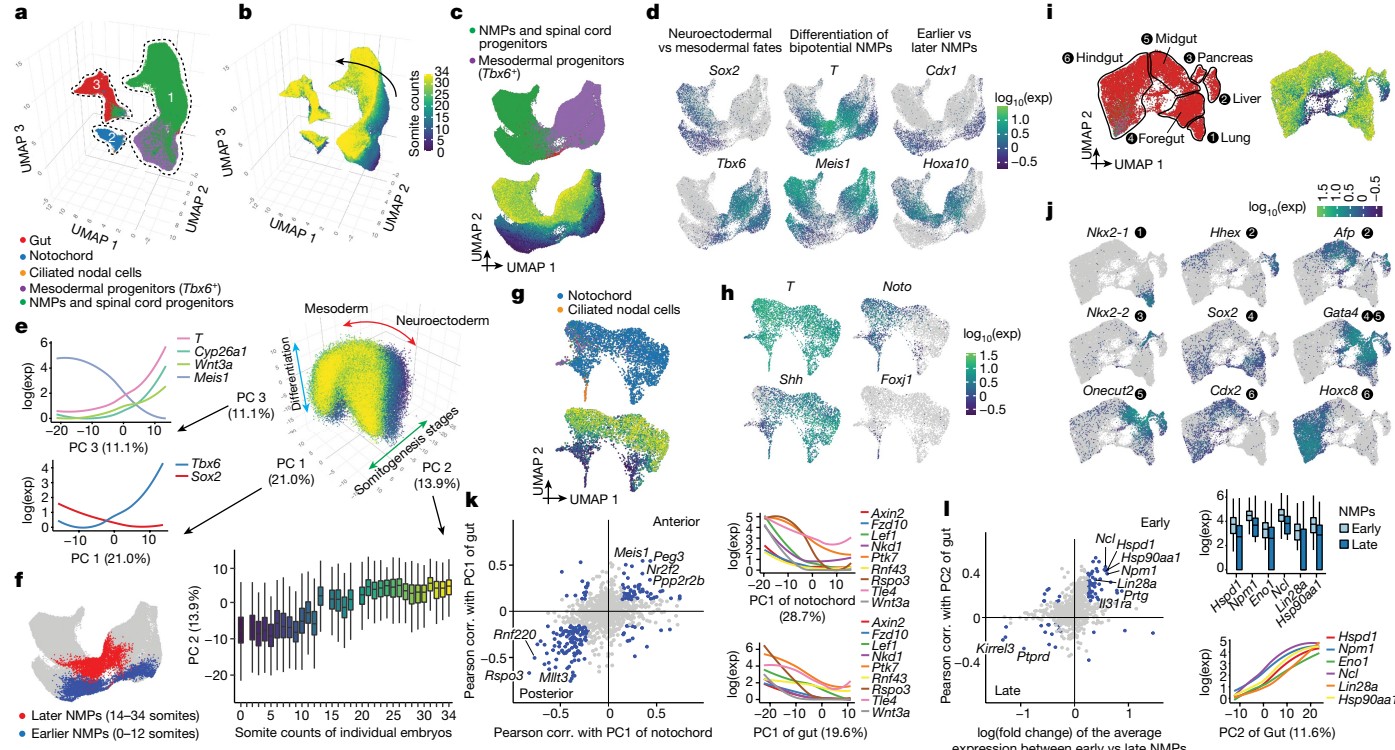

**Fig. 2 | Transcriptional heterogeneity in the posterior embryo during early somitogenesis. a**, Re-embedded 3D UMAP of 121,118 cells from selected posterior embryonic cell types at early somitogenesis (somite counts 0–34; E8–E10). Three clusters are identified. **b**, The same UMAP as in **a**, coloured by somite counts. **c**, Re-embedded 2D UMAP of cells from cluster 1. **d**, The same UMAP as in **c**, coloured by marker gene expression for NMP subpopulations (Supplementary Table 12). Exp, expression. **e**, 3D visualization of the top three principal components of gene expression variation in cluster 1. Correlations between top three principal components and the normalized expression of selected genes (left) or somite counts (bottom). **f**, The same UMAP as in **c**, with earlier (*n* = 4,949 cells) and later (*n* = 3,910 cells) NMPs highlighted. NMPs: *T*⁺ (raw count ≥ 5) and *Meis1*⁻ (raw count = 0). **g**, Re-embedded 2D UMAP of cells from cluster 2. **h**, The same UMAP as in **g**, coloured by marker gene expression

for notochord or ciliated nodal cells (*Foxj1*⁺). **i**, Re-embedded 2D UMAP of cells from cluster 3. Black circles highlight gut cell subpopulations. **j**, The same UMAP as in **i**, coloured by marker gene expression for gut cell subpopulations (Supplementary Table 12). **k**, Left, Pearson correlation (corr.) with PC1 of notochord or gut for highly variable genes. Right, gene expression of selected Wnt signalling genes versus PC1 of notochord or gut. **l**, Left, fold changes between early and late NMPs and Pearson correlation with PC2 of gut are plotted for highly variable genes. Right, gene expression of selected genes (several MYC targets, *Lin28a* and *Hsp90aa1*) versus early and late NMPs or PC2 of gut. In **c,g,i**, cells are coloured by either initial annotations or somite counts. Box plots in **e** (*n* = 98,545 cells) and **l** (*n* = 8,859 cells) represent inter-quartile range (IQR) (25th, 50th and 75th percentile) and whiskers represent 1.5× IQR.

somite-staged embryos (0–34 somites) initially annotated as NMPs and spinal cord progenitors, mesodermal progenitors (*Tbx6*), notochord or gut (Fig. 2a–c).

First focusing on NMPs and their immediate derivatives (cluster 1 in Fig. 2a), we performed PCA on highly variable genes. The top three principal components, which explain nearly half of transcriptional variation, appear to correspond to neural versus mesodermal fate (PC1), developmental stage (PC2) and bipotentiality versus differentiation towards either fate (PC3) (Fig. 2d,e and Supplementary Table 6). Assuming that PC3 tracks differentiation consistently between neural versus mesodermal fates, our data suggest that being brachyury-positive (*T*⁺) and *Meis1*⁻ may better indicate bipotency than being *T*⁺ and *Sox2*⁺, consistent with recent studies of NMPs' genetic dependencies[22–24] (Fig. 2e,f). *Cyp26a1* (whose gene product inactivates retinoids) and *Wnt3a* (involved in canonical Wnt signalling) were also strongly correlated with bipotency.

We observe marked contrasts between earlier (0–12 somites) and later (14–34 somites) NMPs, which may correspond to the 'trunk-to-tail' transition[25] (Fig. 2c–f). This observation is consistent with differences between NMPs from microdissected E8.5 versus E9.5 embryos[26], implicating many of the same genes (for example, *Cdx1* (early) and *Hoxa10* (late); Fig. 2d and Supplementary Table 7). However, given concern about batch effects, we profiled an additional 12 embryos (8–21

somites). This new experiment validated and refined the estimated timing of this transition (Extended Data Fig. 4a–f).

Another cell type marked by the master transcriptional regulator *T* is the notochord (cluster 2 in Fig. 2a). In 0–12 somite embryos, we observe distinct notochordal subsets, one expressing *Noto* (notochord homeobox) and another *Shh* (sonic hedgehog) (Fig. 2g,h). As somitogenesis progresses, the inferred derivatives of these subsets remain distinguishable. The *Noto*⁺ subset is marked by posterior *Hox* genes, Notch and Wnt signalling, and mesodermal differentiation modules (Extended Data Fig. 4g). Within this subset, we identify a few cells that strongly express *Foxj1* and motile ciliogenesis genes. These ciliated nodal cells, which set the left–right axis[27], are both extremely rare and transient, peaking at the 2-somite stage (Fig. 2g,h and Extended Data Fig. 4h).

By contrast, the inferred derivatives of the *Shh*⁺ subset express genes involved in neurogenesis and synaptogenesis—for example, *Sox10*, *Bmp3*, *Nrg1* and *Erbb4* (Extended Data Fig. 4i). We speculate that the *Noto*⁺ subset corresponds to posterior notochord, arising from the node, whereas the *Shh*⁺ subset corresponds to anterior mesendoderm (that is, anterior head process and possibly prechordal plate), arising by condensation of dispersed mesenchyme and possibly contributing to forebrain patterning[28–31]. These presumably anterior–posterior differences are a major source of notochordal heterogeneity (PC1 = 29%; Supplementary Table 8).

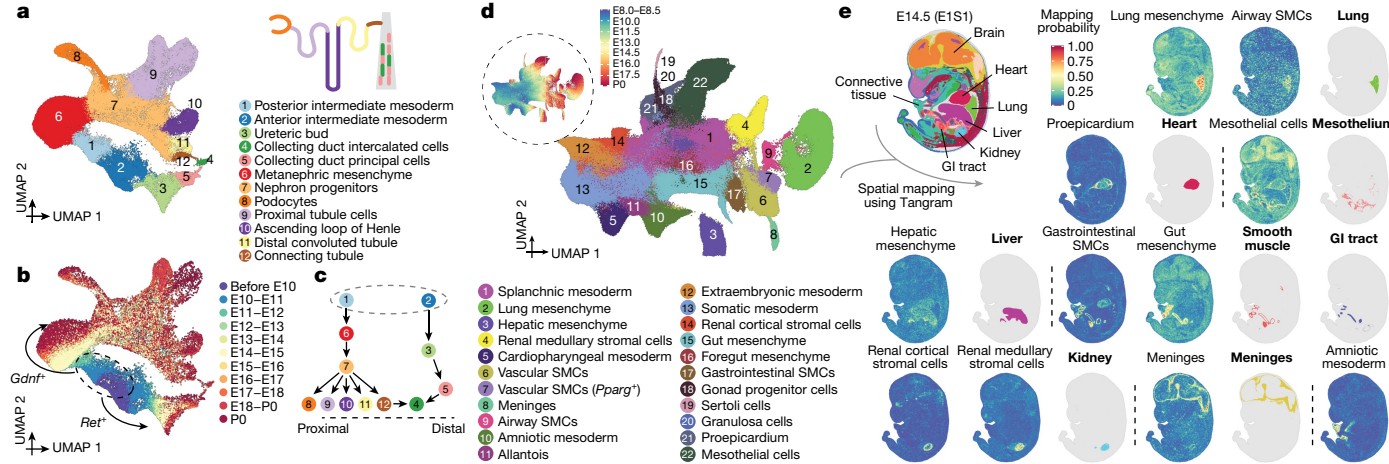

**Fig. 3 | Diversification of the intermediate mesoderm and LPM.**
**a**, Re-embedded 2D UMAP of 95,226 cells corresponding to renal development. A schematic of a nephron is shown at the top right. **b**, The same UMAP as in **a**, coloured by developmental stage (plotting a uniform number of cells per time window). The inset dashed circle highlights posterior and anterior intermediate mesoderm, with arrows highlighting derivative trajectories expressing *Gdnf* and *Ret*, respectively. **c**, Manually inferred relationships between annotated renal cell types. Label annotations in **a**. **d**, Re-embedded 2D UMAP of 745,494 cells from lateral plate and intermediate mesoderm derivatives. The inset

dashed circle highlights the same UMAP with cells coloured by developmental stage. SMC, smooth muscle cell. **e**, The spatial origin of each lateral plate and intermediate mesoderm derivative was inferred based on a public dataset, Mosta[46], together with our data and the Tangram algorithm[47] (Methods). An image of one selected section (E1S1) from E14.5 of the Mosta data is shown at the top left with major regions labelled. The spatial mapping probabilities across voxels on this section for selected subtypes are shown (non-bold label), with the regional annotation appearing to best correspond to the inferred spatial pattern shown alongside (labelled in bold). GI, gastrointestinal.

Turning to gut (cluster 3 in Fig. 2a), we again observe distinct progenitor subsets that transition to a continuum as somitogenesis progresses (Fig. 2i). A major aspect of this continuum also reflects anterior–posterior patterning, with subsets corresponding to lung, liver, pancreas, foregut, midgut and hindgut progenitors (PC1 = 20%; Fig. 2j and Supplementary Table 9). As *T* is classically associated with notochord and posterior mesoderm, we were initially surprised by strong *T* expression in the putative posterior hindgut, coincident with posterior *Hox* genes (Extended Data Fig. 4j). However, this pattern has been documented[32], and is consistent with the ancestral role of *T* in closing the blastopore[33] as well as hindgut defects in *Drosophila brachyenteron* and *Caenorhabditis elegans mab-9* mutants[34,35].

Of note, there is strong overlap between genes underlying the inferred anterior–posterior axis of axial mesoderm (notochord; PC1; $n = 591$) and endoderm (gut; PC1; $n = 502$) (198 overlapping genes, 86% directionally concordant; $P < 10^{-28}$, $\chi^2$-test; Fig. 2k and Supplementary Table 10). Concordantly posterior-associated genes are highly enriched for Wnt signalling and posterior *Hox* genes. One model to explain these overlaps between germ layers is that they are residual to the common origin of anterior mesendodermal derivatives from early and mid-gastrula organizers (anterior head process, prechordal plate and anterior endoderm) versus posterior mesendodermal derivatives from the node[28] (notochord and posterior endoderm). Alternatively, they could be explained by physically coincident progenitors of these germ layers being exposed to similar patterns of Wnt signalling.

A second overlap between germ layers involves genes correlated with early versus late somite counts in NMPs ($n = 257$) versus gut (PC2; $n = 502$) (82 overlapping genes, 70 (85%) directionally concordant; $P < 10^{-15}$, $\chi^2$-test) (Fig. 2l and Supplementary Table 11). Given concern about batch effects, we re-examined the aforementioned replication series (8–21 somite embryos). Seventy-seven per cent of the overlapping, concordant genes replicated in terms of directionality-of-change between early versus late NMPs and gut (54 out of 70; expected value 25%; Extended Data Fig. 4a–f). Genes reproducibly associated with early stages in both germ layers were strongly enriched for MYC targets, and included *Lin28a*, a deeply conserved regulator of developmental timing[36]. Other genes, such as *Npm1* and *Hsp90* isoforms are plausibly

associated with batch effects. However, analysis of a module of genes correlated with *Npm1* revealed that this module declined with developmental time across the entire time series, rather than being correlated with batch variables (Extended Data Fig. 4k,l).

## Intermediate and lateral plate mesoderm

Above, we investigated aspects of axial and paraxial mesoderm, which give rise to notochord and somites, respectively. Next, we focus on the transition from intermediate mesoderm to nephrons, and lateral plate mesoderm (LPM) to organ-specific mesenchyme.

Our aim was to explore the continuum of transcriptional states that span the transition from intermediate mesoderm to functional nephrons. Re-embedding 95,226 relevant cells, we observe two major trajectories, one corresponding to posterior intermediate mesoderm→renal tubules, and another corresponding to anterior intermediate mesoderm→collecting ducts (Fig. 3a–c). In late gastrulation, posterior (*Gdnf*+) and anterior (*Ret*+) intermediate mesoderm[37,38] initially progress to metanephric mesenchyme and ureteric bud states, respectively, then onwards to functional components of the nephron (Extended Data Fig. 5a–c). Cells annotated as podocytes and proximal tubule cells but unexpectedly appearing as early as E10.5 may correspond to mesonephric tubules[37]. Metanephric mesenchyme and ureteric bud states persist through P0, presumably reflecting ongoing nephrogenesis, which continues for a few days after birth[39]. The apparent bifurcation of proximal tubule cell states at later stages corresponds to major differences in cells obtained before versus after birth (Extended Data Fig. 5d). We return to this observation further below.

Both tip and stalk cells are identified within the ureteric bud—the tip cells giving rise to the collecting duct, and the stalk cells giving to the ureter[40,41] (Extended Data Fig. 5e). Notably, we observe transcriptional 'convergence' of the posterior and anterior trajectories in collecting duct intercalated cells (cluster 4 in Fig. 3a,b). More detailed investigation supports a contribution of the posterior trajectory to the collecting duct, consistent with recent lineage tracing experiments demonstrating a dual origin for intercalated cell types from distal nephron and ureteric lineages[42] (Fig. 3c and Extended Data Fig. 5f–h).

The LPM is considerably more complex than the axial, paraxial and intermediate mesoderms[43]. Although some LPM derivatives have been intensely studied (for example, limb and heart), others remain poorly understood, in particular the mesoderm lining the body wall and internal organs. This aspect of LPM gives rise to a remarkable diversity of cell types and structures (including fibroblasts, smooth muscle, mesothelium, pericardium, adrenal cortex and others) and its reciprocal interactions with other germ layers has a key role in organ patterning[44,45].

To annotate understudied LPM derivatives, we leveraged spatial transcriptomic data to impute coordinates for our cells[46,47], which enabled us to annotate 22 subtypes of the LPM and intermediate mesoderm major cluster, including cardiac (proepicardium), brain (meninges), lung, liver, foregut and gut mesenchyme, and airway versus gastrointestinal versus vascular smooth muscle (Fig. 3d,e, Extended Data Fig. 6 and Supplementary Table 12). Two subtypes spatially mapped to the kidney, one to the cortex and the other heterogeneously, which we term renal cortical stromal cells and renal medullary stromal cells, respectively[48] (Fig. 3d,e and Extended Data Fig. 7a–c). Although both express *Foxd1*⁺, focused analyses suggest distinct origins, with renal cortical stromal cells appearing to derive from the intermediate mesoderm and metanephric mesenchyme, and renal medullary stromal cells appearing to derive from LPM (Extended Data Fig. 7d,e). However, lineage tracing experiments would be necessary to provide conclusive evidence for this. Of note, renal medullary stromal cells exhibited heterogeneity along what may be a cortical–medullary spatial axis (Extended Data Fig. 7f).

The temporal resolution of our studies enables us to narrow the window during which various organ-specific mesenchymes are specified (Extended Data Fig. 8a). We also applied a mutual nearest neighbours (MNN) heuristic to identify putative precursors of each subtype (Extended Data Fig. 8b–g)—for example, subsets of splanchnic mesoderm most highly related to foregut mesenchyme, hepatic mesenchyme or proepicardium—which may correspond to the 'territories' in which these organ-specific mesenchymes are induced (Extended Data Fig. 8b–d). For example, hepatic and foregut mesenchyme are distinguished both from one another as well as from their inferred progenitors by *Gata4* and *Barx1* expression, respectively[49,50]. However, their inferred progenitors are also distinct from one another, with inferred hepatic mesenchymal progenitors expressing a programme of epithelial–mesenchymal transition and inferred foregut mesenchymal progenitors expressing multiple guidance cue programmes (for example, semaphorins, ephrins, SLIT family proteins and netrins) (Extended Data Fig. 8c and Supplementary Table 13).

## From patterned neuroectoderm to neurons

We now turn from mesoderm to neuroectoderm. Relative to our previous studies[14], optimizations of sci-RNA-seq3 have markedly improved our ability to distinguish neuronal subtypes. For example, in Supplementary Note 1, we describe the timing and trajectories of prenatal diversification of the retina. In that context, we can distinguish 15 retinal ganglion subtypes by P0, on par with expectation[51], each well defined by specific transcription factor combinations (Extended Data Fig. 9a–l and Supplementary Table 14).

In our earliest embryos (0–12 somites), we previously defined a continuum of cell states that correlated with anatomical patterning of the 'pre-neurogenesis' neuroectoderm[8]. Extending this analysis through early organogenesis (E8–E13), we observe clusters corresponding to territories that will give rise to the major regions of the mammalian brain (Fig. 4a and Extended Data Fig. 9m). As development unfolds further, we observe many trajectories of neurogenesis arising from these inferred territories (Fig. 4b,c).

Beginning as early as the 16-somite stage, most neuronal diversity derives from direct neurogenesis (Fig. 4d), including motor neurons, cerebellar Purkinje cells, Cajal–Retzius cells and many other subtypes

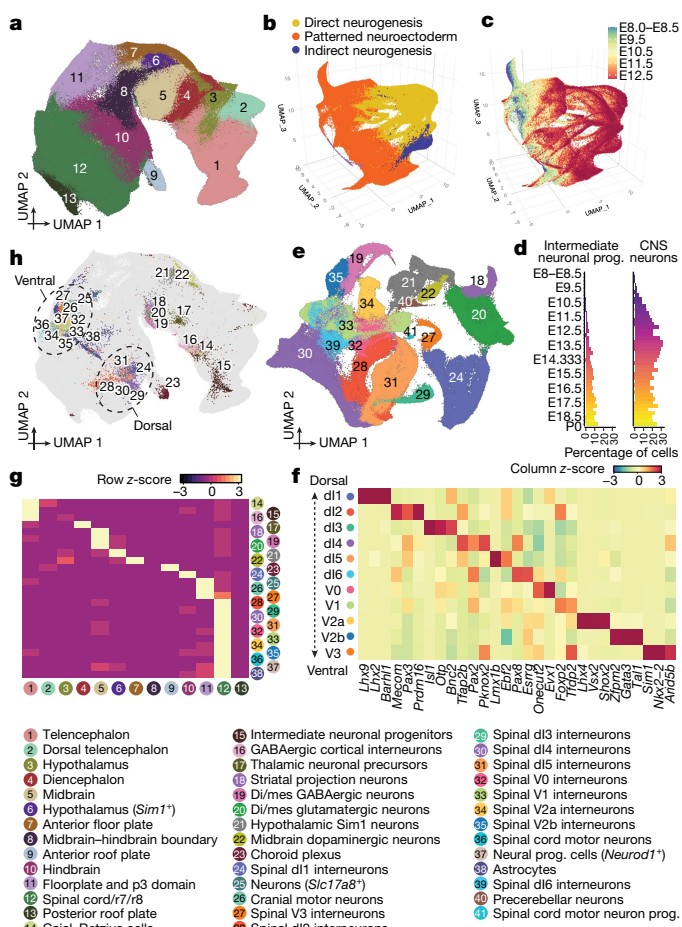

**Fig. 4 | The emergence of neuronal subtypes from the patterned neuroectoderm. a**, Re-embedded 2D UMAP of 1,185,052 cells, corresponding to different neuroectodermal territories from neuroectoderm and glia; major cell clusters, from stages before E13. **b**, Re-embedded 3D UMAP of 1,772,567 cells from neuroectodermal territories together with derived cell types, from stages before E13. Patterned neuroectoderm comprises neuroectoderm and glia and choroid plexus; direct neurogenesis comprises CNS neurons; indirect neurogenesis comprises intermediate neuronal progenitors. **c**, The same UMAP as in **b**, coloured by timepoint. **d**, Composition of embryos from each 6-h bin by intermediate neuronal progenitor (left) and CNS neuron (right) major cell clusters. **e**, Re-embedded 2D UMAP of 296,020 cells (glutamatergic neurons, GABAergic neurons, and spinal cord dorsal and ventral progenitors) from stages before E13. **f**, The top 3 transcription factor markers of the 11 spinal interneurons. Marker transcription factors were identified using the FindAllMarkers function of Seurat v3[63]. The heat map shows mean gene expression values per cluster, calculated from normalized UMI counts. **g**, The row-scaled number of MNN pairs identified for each derivative cell type between its earliest 500 cells and cells from neuroectodermal territories. Some derivative cell types are excluded owing to low cell number or MNN pairs. **h**, The same UMAP as in **a**, but with inferred progenitor cells coloured by derivative cell type with the most frequent MNN pairs. Dotted circles highlight the dorsal and ventral spinal interneuron neurogenesis domains of the hindbrain and spinal cord. Di/mes, diencephalon and mesencephalon.

1 Telencephalon
2 Dorsal telencephalon
3 Hypothalamus
4 Diencephalon
5 Midbrain
6 Hypothalamus (*Sim1*⁺)
7 Anterior floor plate
8 Midbrain–hindbrain boundary
9 Anterior roof plate
10 Hindbrain
11 Floorplate and p3 domain
12 Spinal cord/r7/r8
13 Posterior roof plate
14 Cajal–Retzius cells
15 Intermediate neuronal progenitors
16 GABAergic cortical interneurons
17 Thalamic neuronal precursors
18 Striatal projection neurons
19 Di/mes GABAergic neurons
20 Di/mes glutamatergic neurons
21 Hypothalamic Sim1 neurons
22 Midbrain dopaminergic neurons
23 Choroid plexus
24 Spinal dl1 interneurons
25 Neurons (*Slc17a8*⁺)
26 Cranial motor neurons
27 Spinal V3 interneurons
28 Spinal dl2 interneurons
29 Spinal dl3 interneurons
30 Spinal dl4 interneurons
31 Spinal dl5 interneurons
32 Spinal V0 interneurons
33 Spinal V1 interneurons
34 Spinal V2a interneurons
35 Spinal V2b interneurons
36 Spinal cord motor neurons
37 Neural prog. cells (*Neurod1*⁺)
38 Astrocytes
39 Spinal dl6 interneurons
40 Precerebellar neurons
41 Spinal cord motor neuron prog.

(CNS neurons sub-panel of Extended Data Fig. 3). Indirect neurogenesis[52] has a later start, with intermediate neuronal progenitors first detected at E10.25, later giving rise to deep-layer neurons, upper-layer neurons, subplate neurons, and cortical interneurons (Fig. 4d and Extended Data Fig. 10a,b). Although many subtypes deriving from direct neurogenesis are easily distinguished, the majority (55%) of these 2.1 million cells could initially only be coarsely annotated as glutamatergic or GABAergic (γ-aminobutyric acid-producing) neurons

or dorsal or ventral spinal cord progenitors. To leverage the greater heterogeneity evident at early stages as these trajectories 'launch' from the patterned neuroectoderm, we re-analysed the pre-E13 subset. This facilitated much more granular annotation, while also highlighting sources of heterogeneity—for example, anterior versus posterior or inhibitory versus excitatory (Fig. 4e, Extended Data Fig. 10c,d and Supplementary Table 12).

Among these more refined annotations of direct neurogenesis derivatives were 11 spinal interneuron subtypes; similar to retinal ganglion subtypes, these were well defined by transcription factor combinations[53] (Fig. 4f and Supplementary Table 15). The top principal components of transcriptional heterogeneity among spinal interneurons appear to correspond to neuronal differentiation (PC1 and PC2), glutamatergic versus GABAergic identity (PC3), and dorsal versus ventral identity (PC4) (PC1–4 (50%); Extended Data Fig. 10e,f and Supplementary Table 16).

We next sought to infer the progenitors from which various neuronal and non-neuronal cell types derive. First, we took pre-E13 cells annotated as astrocytes, choroid plexus or any direct or indirect neurogenesis derivative, and co-embedded them with cells of the patterned neuroectoderm. Next, for each derivative cell type in the co-embedding, we selected the 500 'youngest' cells, identified their patterned neuroectoderm MNNs and then mapped these back to our original embedding of patterned neuroectoderm (Fig. 4g,h). The resulting distribution of inferred progenitors is considerably more granular than our annotations of anatomical territories (compare Fig. 4h with Fig. 4a).

For non-neuronal subtypes, the inferred progenitors of the choroid plexus overwhelmingly map to the anterior roof plate (91%), with a minor subset in the dorsal diencephalon (5%), although this balance is likely impacted by the time window of this analysis[54] (E8–E13). Inferred astrocyte progenitors exhibit a more complex distribution, with VA2 progenitors primarily assigned to the spinal cord, r7 and r8 (83%) and hindbrain (16%), and VA3 progenitors to the spinal cord, r7 and r8 (57%) and floorplate and p3 domain[55] (32%) (Extended Data Fig. 10g–j). VA1 astrocytes arise later than VA2 and VA3 astrocytes, and were not present in sufficient numbers for their progenitors to be inferred.

For neuronal subtypes, inferred progenitors largely fall within the expected territories, but with considerable granularity (Fig. 4h). For example, inferred progenitors of dorsal and ventral spinal interneurons cluster distinctly. Although the progenitors of three neuronal subtypes (cerebellar Purkinje neurons, precerebellar neurons and spinal dI6 interneurons) were not clearly defined by the method described above, an iterative variant of the MNN heuristic suggested that cerebellar Purkinje neurons and dl2 spinal interneurons have common or at least transcriptionally similar progenitors, which may have confounded the original analysis (Extended Data Fig. 10k).

We next examined how the identities of neuronal subtypes are established and maintained[56]. We identified transcription factors specific to each of the 11 spinal interneuron subtypes (median 53 per subtype; Fig. 4f and Supplementary Table 15). However, within each subtype, these transcription factors exhibit complex temporal dynamics, with most only expressed transiently (Extended Data Fig. 10l). Focusing on spinal interneurons dl1–dl5, we could also identify transcription factors specific to the inferred progenitors of each subtype, relative to the inferred progenitors of other dorsal spinal interneurons (Extended Data Fig. 10m, left). Most of these were basic helix–loop–helix or homeodomain transcription factors[57]. However, consistent with the transitional expression of other subtype-specific transcription factors, their expression was generally not maintained for very long after neuronal specification (Extended Data Fig. 10m, right).

Finally, we sought to systematically delineate the timing of differentiation (Extended Data Fig. 10n). This analysis suggests that the emergence of each derivative cell type from the patterned neuroectoderm is both cell-type-specific and modestly asynchronous. For example, about 95% of inferred progenitors of dl2 spinal interneurons are from 20-somite to E11 stage embryos, whereas 95% of dl4 spinal interneurons inferred progenitors are from 27-somite to E11.75 stage embryos.

Together, these analyses are consistent with a model articulated by Sagner and Briscoe[56] in which both spatial and temporal factors heavily contribute to the specification of neuronal subtypes as they emerge from the patterned neuroectoderm. Furthermore, they highlight the complexity of this process not only at the initiation of each neuronal subtype, but also over the course of their early maturation—for example, at 6-h resolution, we can observe each spinal interneuron subtype expressing a dynamic succession of developmentally potent transcription factors (Extended Data Fig. 10l).

## A cell-type tree from zygote to birth

A primary objective of developmental biology is to delineate the lineage relationships among cell types. Transcriptional profiles of single cells do not explicitly contain lineage information. However, assuming that a continuity of transcriptional states spans all cell-type transitions, we can envision a tree accurately relating cell types based solely on scRNA-seq data[58]. Indeed, we and others have constructed such trees for portions of worm, fly, fish, frog and mouse development[7,9–14,17].

On the basis of these learnings, we constructed a rooted tree of cell types that spans mouse development from zygote to birth, based on four published datasets[4–7] (110,000 cells; E0–E8.5) and the dataset reported here (11.4 million cells; E8–P0) (Supplementary Table 17). Challenges included the heterogeneity of technologies used to generate the data, that cells' transcriptional states are only loosely synchronized with developmental time, the multiple scenarios by which cell state manifolds may be misleading[58], and finally, the sheer complexity of this organism. To overcome these challenges, we took a heuristic approach.

First, we split cells into 14 subsystems to be separately analysed and subsequently integrated (pre-gastrulation, gastrulation, and 12 organogenesis and fetal subsystems; Supplementary Tables 17 and 18).

Second, dimensionality reduction was performed on each subsystem and 283 cell-type nodes were defined, largely but not entirely corresponding to our original cell-type annotations (Supplementary Table 19 and 20). The cells comprising each node derived from a single data source, but usually from multiple timepoints within that data source.

Third, we sought to draw edges between nodes (Fig. 5a–f). Within each subsystem, we identified pairs of cells that were MNNs in 30-dimensional PCA space. Although the overwhelming majority of MNNs occurred within nodes, some MNNs spanned nodes, presumably enriched for bona fide cell-type transitions. Each possible edge (that is, node pair) was ranked based on a normalized count of inter-node MNNs (Supplementary Table 21). The MNN approach is robust to technical factors or parameter choices (Extended Data Fig. 11a–c and Supplementary Note 2).

Fourth, we manually curated the top 1,155 candidate edges for biological plausibility (Extended Data Fig. 11d), leaving 452 edges, which we further categorized as likely reflecting 'developmental progression' or 'spatial continuity' (Supplementary Table 22). Notably, where nodes were connected to multiple other nodes, distinct subsets of cells were generally involved in each edge, and inter-node MNN pairs exhibited temporal coincidence (Fig. 5a–f). As only a handful of cells were profiled in the pre-gastrulation subsystem, its edges were added manually.

Finally, to bridge subsystems, we performed batch correction and co-embedding of selected timepoints from different data sources, resulting in a third category of 'dataset equivalence' edges (Extended Data Fig. 11e–h). Ten of the organogenesis and fetal development subsystems could be linked to equivalent cell-type nodes in the gastrulation subsystem in a data-driven manner, and two required edges to be manually added based on biological plausibility. Altogether, we added 55 inter-subsystem edges.

The resulting developmental cell-type tree, spanning E0 to P0, can be represented as a rooted, directed graph (Fig. 5g).

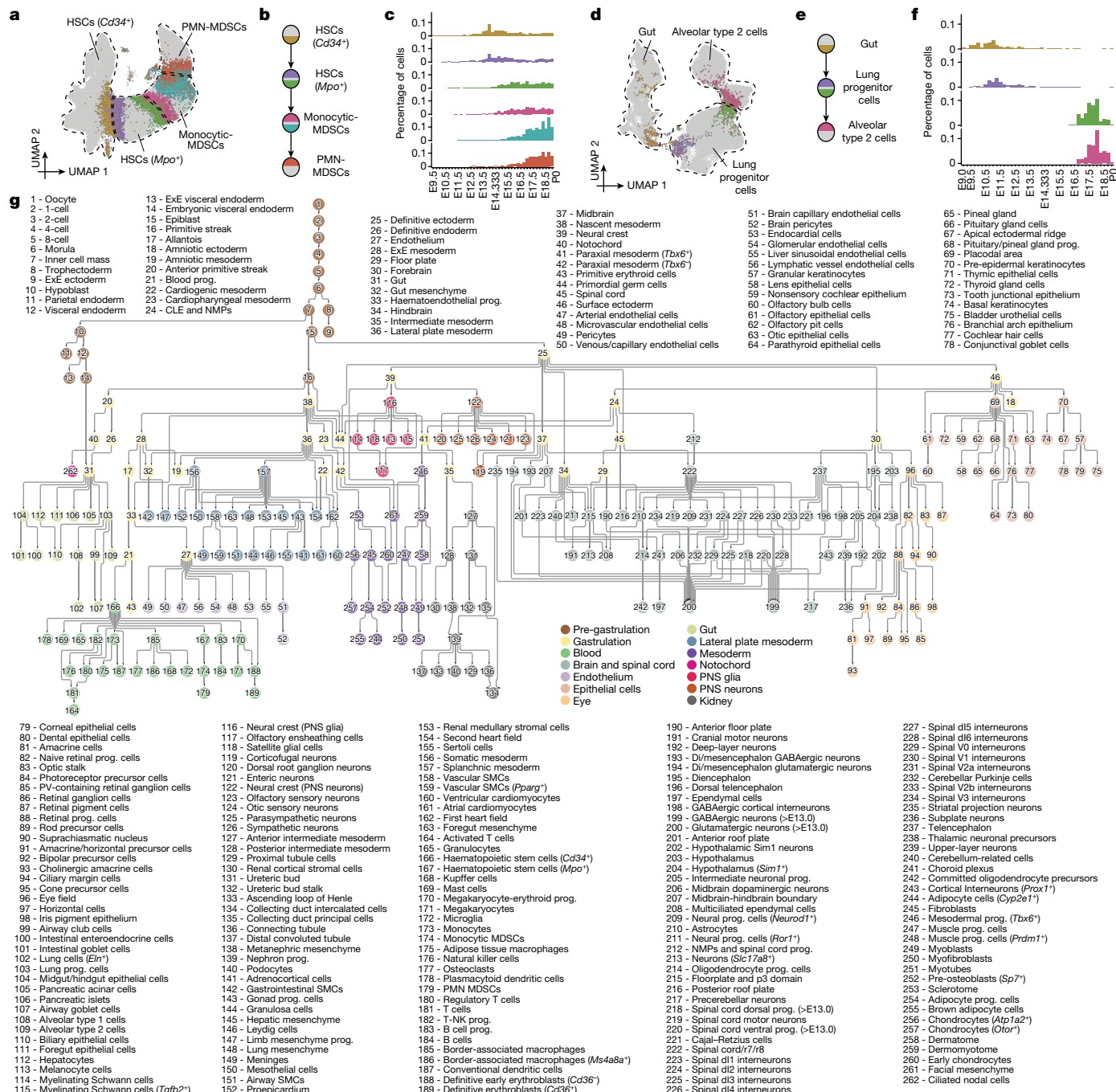

**Fig. 5 | A data-driven tree relating cell types throughout mouse development, from zygote to pup. a**, Illustration of the basis for the edge inference heuristic. Re-embedded 2D UMAP of 101,001 cells from *Cd34*+ HSCs, *Mpo*+ HSCs, monocytic myeloid-derived suppressor cells (MDSCs) and PMN MDSCs within the 'blood' subsystem. Cells involved in MNN pairs that bridge cell types are coloured. **b**, Inferred lineage relationships between annotated cell types in **a**, with corresponding colour scheme. **c**, The percentage of inter-cell-type MNN cells (*y* axis) versus the total number of cells profiled from embryos from the corresponding time bin, with colour scheme as in **a**,**b**. **d**, Additional illustration of the basis for the edge inference heuristic. Re-embedded 2D UMAP of 71,718 cells from gut, lung progenitor cells and alveolar type 2 cells within the 'gut' subsystem. Cells involved in MNN pairs that bridge cell types are coloured. **e**, Inferred lineage relationships between annotated cell types in **d**, with corresponding colour scheme. **f**, The percentage of inter-cell-type MNN cells (*y* axis) versus the total number of cells profiled from embryos from the corresponding time bin, with colour scheme as in **d**,**e**. **g**, A rooted, directed graph corresponding to development of a mouse, spanning E0 to P0 (Methods). For presentation purposes, we removed most 'spatial continuity' edges and merged nodes with redundant labels derived from different datasets, resulting in a rooted graph comprising 262 cell-type nodes and 338 edges. Nodes are coloured and labelled by each of the 14 subsystems. CLE, caudal lateral epiblast; ExE, extra-embryonic; NK-T cell, natural killer T cell; PV, parvalbumin.

## Key drivers of cell-type transitions

We next sought to test which transcription factors or other genes sharply change in expression with the emergence of each cell type.

First, for each directional cell-type transition edge between two nodes in the graph (A→B), we identified both 'inter-node' MNNs, as well as 'intra-node' MNNs of the inter-node MNNs. Rather than considering the entirety of A versus B, this heuristic focuses our attention on the

cells most proximate to each cell-type transition (groups 1→2→3→4 in Extended Data Fig. 11i,j). Next, we identified differentially expressed transcription factors (DETFs) and differentially expressed genes (DEGs) across each phase of the modelled transition—that is, early (1→2), inter-node (2→3) and late (3→4). Notably, the early phase is within node A, which may facilitate identification of changes that precede the A→B transition itself.

We applied this heuristic to 436 edges of the rooted tree shown in Fig. 5g, nominating ranked lists of median 28 (IQR 12–51) DETFs and 171 (IQR 76–389) DEGs per edge (Supplementary Tables 23 and 24). Most genes were nominated for only one or a few edges, with outliers that may have more general roles in cell-type specification (Extended Data Fig. 11k,l). Many of the top-ranked upregulated DETFs for the early phase of a transition correspond to an established driver of the derivative cell type (for example, *Mitf* for melanocytes, *Ebf1* and *Pax5* for B cell progenitors, *Lef1* for B cells and *Zfpm1* for megakaryocyte–erythroid progenitors). We also nominated potentially novel drivers that warrant further investigation (including *Tcf7l2* for Kupffer cells, *Ltf* for monocytic myeloid-derived suppressor cells, *Esrrg* for dorsal telencephalon-derived choroid plexus, *Zfp536* for myelinating Schwann cells and *Rreb1* for adipocyte progenitors) (Supplementary Table 23).

Digging into a well-studied transition, *Sox17* is the sole upregulated DETF during the early phase of the anterior primitive streak→definitive endoderm transition, whereas other transcription factors (*Elf3, Sall4, Hesx1, Lin28a, Hmga1* and *Ovol2*, but not *Sox17*) are upregulated during the transition itself (Supplementary Table 23). Non-transcription factor DEGs specific to the early phase of this transition include *Cer1*, ADP/ATP translocase 1 (*Slc25a4*) and *Slc2a3* (also known as *Glut3*) (Supplementary Table 24). To examine this further, we subjected all cells participating in groups 1–4 of this transition to conventional pseudotime analysis[14]. This analysis supported the upregulation of *Sox17* as preceding other nominated transcription factors, and further highlighted *Cer1* as the only non-transcription factor DEG with *Sox17*-like kinetics (Extended Data Fig. 11m,n).

A more complex example involves *Cd34*+ haemopoietic stem cells (HSCs), which in the graph are the origin of a dozen cell types (Extended Data Fig. 11o). Notably, although *Cd34*+ HSCs constitute a single node, the cells composing this node are very heterogeneous, with distinct subsets participating in the MNN pairs that support edges to various lymphoid, myeloid and erythroid derivatives (Extended Data Fig. 11p,q). Correspondingly, the heuristic nominates different transcription factors as early regulators of each transition—for example, *Ebf1* for B cells and *Id2* and *Nfatc2* for conventional dendritic cells (Extended Data Fig. 11r).

## Marked changes immediately after birth

As touched on above, we anecdotally noticed that proximal tubule cells deriving from P0 pups were unusually well-separated from those deriving from late-stage fetuses (Extended Data Fig. 5d). A similar phenomenon was noted for hepatocytes, adipocytes, and various lungs and airway cell types (Fig. 6a). This contrasts sharply with the bulk of the time-lapse, in which cells of a given type were overwhelmingly well mixed across adjacent timepoints. Concerned this was due to batch effects or the pitfalls of over-interpreting UMAPs[59], we conducted a timepoint correlation analysis, testing for each cell type whether the *k*-nearest neighbours of cells of a given timepoint were derived from the same or different timepoints. In this framing, a low proportion of neighbours from different timepoints suggests a temporally abrupt change in transcriptional state. For nearly all cell types, P0 cells exhibited a lower proportion for this metric than all other timepoints (Fig. 6b). Although a trivial explanation would be a longer interval between E18.75 and P0 than 6 h, the pattern was highly non-uniform across cell types, with extreme examples including the aforementioned cell types as well as

various endothelial and blood lineages. In sharp contrast, P0 cells from most neuronal cell types were relatively well mixed with cells deriving from earlier timepoints.

To validate this phenomenon, we collected nine pups from a single litter. Three were delivered vaginally, and the remaining six by caesarean section (C-section) and euthanized either immediately (2 pups), or after 20, 40, 60 or 80 min (1 pup each) (Fig. 6c and Extended Data Fig. 12a). Nuclei from these nine pups were analysed in a new sci-RNA-seq3 experiment, which yielded nearly one million additional single-cell profiles (Extended Data Fig. 12b and Supplementary Tables 1 and 2).

We applied timepoint correlation analysis to 24 major cell clusters identified in the 6 C-section embryos, as above except treating time after C-section as a continuous variable. Once again, hepatocyte, adipocyte and lung and airway cells were major outliers, validating our initial finding and narrowing the window in which these abrupt changes emerge to the first hour of extrauterine life (Fig. 6d,e and Extended Data Fig. 12c,d). Although we cannot fully rule out technical artefacts, we took care to minimize handling and stress prior to euthanasia and immediate snap freezing. Furthermore, it is plausible that rapid changes in transcriptional programmes might be physiologically necessary owing to the profound differences between the placental and extrauterine environments. In examining DEGs of rapidly changing cell types, either in E18.75 versus P0 embryos or across the C-section time series, we see clues that support this interpretation (Supplementary Tables 25 and 26).

For example, in hepatocytes, genes involved in gluconeogenesis are sharply upregulated, including *Ppargc1a*, which encodes PGC-1α, a master regulator of hepatic gluconeogenesis, as well as *Pck1*, *G6Pc* and *Got1*, which encode key enzymes in this pathway (Fig. 6f). Aspects of these changes have previously been linked to changes in key nutritional hormones immediately after birth and are presumably necessary for maintaining normoglycaemia in the wake of being abruptly cut off from maternal nutrients[60]. In brown adipocytes, we observe sharp upregulation of *Irf4*, a cold-induced master regulator of thermogenesis, and again of *Ppargc1a*, which in adipocytes has a different role than in the liver, as PGC-1α partners with IRF4 to drive the expression of *Ucp1* and uncoupled respiration[61], presumably to maintain body temperature upon transition to the extrauterine environment[62] (Fig. 6f).

The time elapsed between vaginal births and the collection of pups was not precisely captured in the replication experiment. However, on co-embedding cells derived from vaginally birthed pups with those delivered by C-section for the three most relevant major cell clusters, timepoint correlation analysis suggested they were collected within 1 h of birth (Extended Data Fig. 12e). However, this assumes similar kinetics for these rapid transcriptional changes in C-section versus vaginally delivered pups. On more detailed inspection, the patterns are considerably more complex, with certain clusters appearing to be specific to vaginally birthed pups (Extended Data Fig. 12f and Supplementary Table 27).

## Discussion

We profiled the transcriptional states of 12.4 million nuclei from 83 precisely staged embryos spanning late gastrulation (E8) to birth (P0), with 2-h temporal resolution during somitogenesis, 6-h resolution to birth, and 20-min resolution immediately postpartum. Despite the scale of the study, the project was driven by a small number of individuals, and not a formal production team. All embryo staging was performed by I.C.W., nearly all data production was done by B.K.M. and all computational analyses were done by C.Q. Nearly all experiments and analyses were completed within one year. Direct costs of reagents and labour were around US$70,000, and sequencing cost around US$300,000. This single dataset is equivalent to about 30% of the aggregated corpus of the Human Cell Atlas Data Portal (https://data.humancellatlas.org/) as of March 2023.

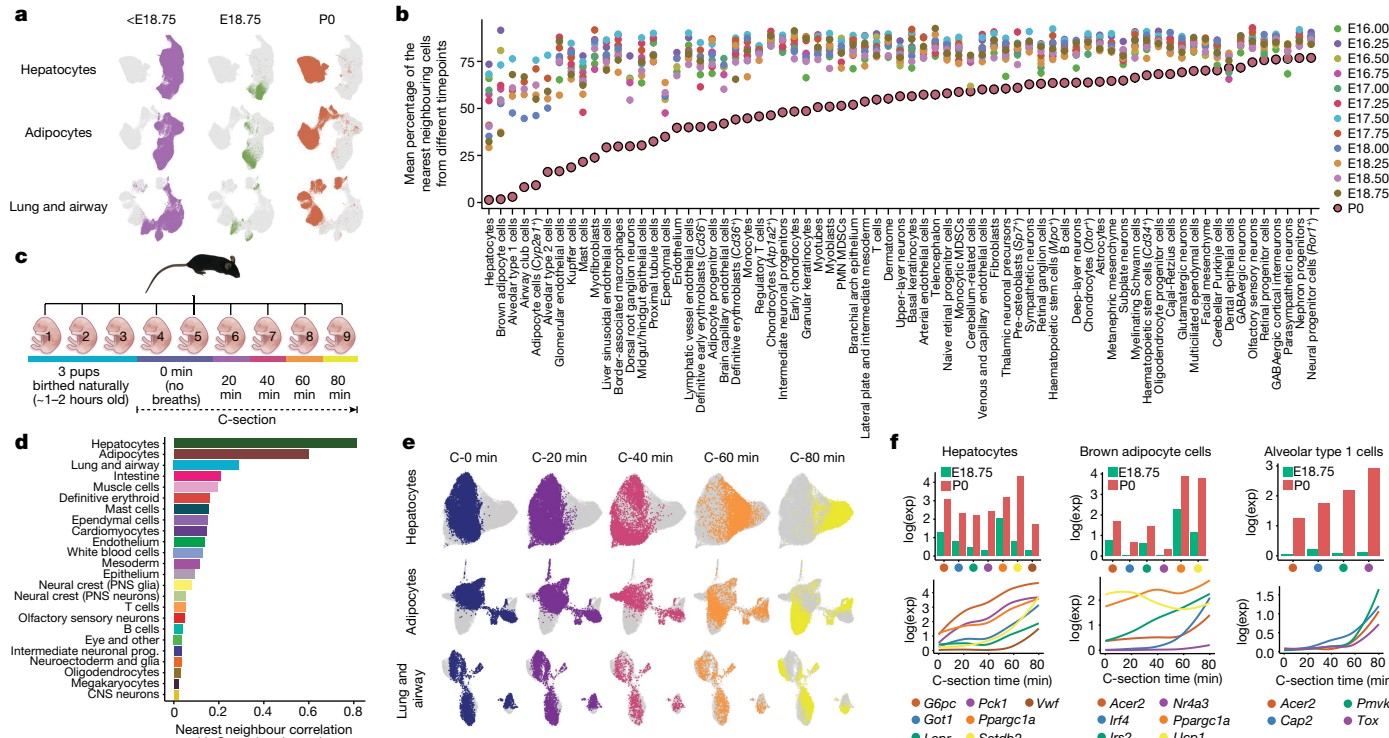

**Fig. 6 | Rapid shifts in transcriptional state occur in a restricted subset of cell types upon birth. a**, Re-embedded 2D UMAP of cells from hepatocytes, adipocytes, and lung and airway, with colours highlighting cells from pre-E18.75 stages (left), E18.75 (middle) or P0 (right) embryos. **b**, We identified cell types with abrupt transcriptional changes before versus after birth by combining cells from animals collected after E16, performing PCA and calculating the average proportion of nearest neighbour cells from a different timepoint for each cell type (Methods). A low proportion of neighbours from different timepoints corresponds to a relatively abrupt change in transcriptional state. P0 points are highlighted with a black boundary. Differentially expressed genes for the 20 most highly ranked cell types are shown in Supplementary Table 25. **c**, A new scRNA-seq dataset (birth series) was generated from nuclei of pups collected after delivery (three vaginal births, six C-sections with 20-min increments). **d**, For each cell cluster in the birth series dataset, we calculated a Pearson correlation between the timepoint of each cell and the average timepoints of its ten nearest neighbours. High correlations indicate rapid, synchronized changes in transcriptional state. **e**, Re-embedded 2D UMAP of cells from hepatocytes, adipocytes, and lung and airway, based on cells from six pups delivered by C-section, with colours highlighting cells from pups collected after different intervals after delivery. **f**, Average normalized gene expression of selected genes for E18.75 versus P0 in the original data (top) and normalized expression of the same genes as a function of C-section timepoints (bottom) for hepatocytes, brown adipocyte cells and alveolar type 1 cells. Gene expression is normalized to total UMIs per cell and plotted as the natural logarithm. The line of gene expression was plotted using the geom_smooth function in ggplot2.

Three broad concepts supported our ability to generate, analyse and integrate such a large dataset with a small team at a modest cost: First, multiplexing, which fundamentally underlies the exponential scalability of single-cell combinatorial indexing as well as that of massively parallel DNA sequencing. Second, open science, as we have taken abundant advantage of many freely released software packages for single-cell data analysis[14,20,47,63]. Third, our focus on mouse development, an eminently reproducible process through which we could access all mammalian cell types (or their predecessors) within a series of physically compact samples.

Our goal in this study was not to learn a specific piece of biology, but rather to advance the foundation for a comprehensive understanding of mammalian development. Although the dataset is a rich source of hypotheses (for example, to identify candidate transcription factor drivers of all prenatal cell types), the largest surprise was the discovery of rapid changes in transcriptional state in a restricted subset of cell types within 1 h immediately following birth. There is immense evolutionary pressure on the transition from placental to extrauterine life, which is arguably as fraught a moment as gastrulation in terms of physiological peril[64]. Some genes that are sharply upregulated in certain cell types can be attributed to specific adaptations. However, many more genes are dynamic in these and myriad other cell types shortly after birth. The adaptive functions served, as well as the mechanisms underlying their rapid induction, are ripe for further exploration.

Notably, human babies delivered by C-section versus vaginal routes have differences in long-term physiology and health outcomes[65]. It is plausible that aspects of these postnatal phenotypic differences have their roots in how the massive, abrupt, cell-type-specific changes documented here are influenced by the mode of delivery.

We only profiled only one embryo for most timepoints, such that we cannot systematically assess interindividual variation. However, such analyses may be better pursued through other datasets—for example, the recent profiling of 101 mutant or wild-type E13.5 embryos[66]. Although both sexes were profiled, generally alternating, we have yet to delve into sex differences, and this remains one of many avenues of investigation that we hope researchers in the field will pursue. The data may also be useful in ways that we did not originally anticipate—for example, for pre-training large language models of mammalian biology[67].

We recently proposed the concept of a consensus ontology of cell types, inclusive of lineage histories and molecular states, as a potential structure for a reference cell tree[68]. The cell-type tree constructed here, which spans mouse development from single-cell zygote to free-living pup, represents a further step in this direction. But just as Sulston reconstructed both the embryonic and post-embryonic lineages of *C. elegans*[69,70], mouse development does not end at P0. Extending this framework to postnatal timepoints may ultimately yield a single-cell time-lapse of the entire mammalian lifespan, from conception to death.

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

## Methods

### Data reporting

For newly generated mouse embryo data, no statistical methods were used to predetermine sample size. Embryos used in the experiments were randomized before sample preparation. Investigators were blinded to group allocation during sample collection and data generation and analysis. Embryo collection and sci-RNA-seq3 data generation were performed by different researchers in different locations.

### Mouse embryo collection and staging

All animal use at The Jackson Laboratory was done in accordance with the Animal Welfare Act and the AVMA Guidelines on Euthanasia, in compliance with the ILAR Guide for Care and Use of Laboratory Animals, and with prior approval from The Jackson Laboratory Animal Care and Use Committee under protocol AUS20028.

The details of collecting the 12 mouse embryos with somite counts ranging from 0 to 12 were described previously[8]. In brief, C57BL/6NJ (strain 005304) mice were obtained at The Jackson Laboratory and mice were maintained via standard husbandry procedures. Timed matings were set in the afternoon and plugs were checked the following morning. Noon of the day a plug was found was defined as E0.5. On the morning of E8.5, individual decidua were removed and placed in ice cold PBS during the collection. Individual embryos were dissected free of extraembryonic membranes, imaged, and the number of somites present were noted prior to snap freezing in liquid nitrogen (Extended Data Fig. 1a). A portion of yolk sac from each embryo was collected for sex based genotyping and samples were stored at −80 °C until further processing.

For newly processed mouse embryos, we used a combination of staging methodologies depending on gestational age of collection (Extended Data Fig. 1b–f). To maximize temporal coherence, resolution, and accuracy, we sought to stage individual embryos based on well-defined morphological criteria, rather than by gestational day alone. Embryos collected between E8.0–E10.0 were staged based upon the number of somites counted at the time of collection and further characterized by morphological features (Extended Data Fig. 1a). For E10.25–E14.75 embryos, developmental age was determined using the embryonic mouse ontogenetic staging system (eMOSS, https://limbstaging.embl.es/), which leverages dynamic changes in hindlimb bud morphology and landmark-free based morphometry to estimate the absolute developmental stage of a sample[71,72]. A modified staging tool, implemented in Python and exhibiting better performance on E14.0–E15.0 samples, was used to confirm staging of samples within this window (documentation and Python scripts available at https://github.com/marcomusy/welsh_embryo_stager). To distinguish samples staged via eMOSS, these samples are prefixed with 'mE' to indicate morphometric embryonic day (for example, mE13.5; Extended Data Fig. 1b–f). Due to the increased complexity of limb morphology at later stages automated staging beyond E15.0 is not possible. As a consequence, collections for all remaining embryonic samples (E15.0–E18.75) was performed precisely at 00:00, 06:00, 12:00 and 18:00 on the targeted day. From close inspection of limbs in this sample set we defined additional dynamics related to digit morphogenesis that allowed further binning of samples collected on days 15 and 16 (Extended Data Fig. 1b–f). Therefore, amongst samples profiled in this study, only the E17.0–E18.75 samples were staged solely by gestational age. Finally, P0 samples were collected from litters at noon of the day of birth (parturition for C57BL/6NJ occurs between E18.75 and E19.0).

### Collection of mouse pups immediately after birth

Samples for the validation experiment on periparturition transcriptional dynamics were collected from a plugged female that was monitored for signs of labour beginning at E18.75. Following the natural delivery of 3 pups the dam was euthanized, and following removal from the uterus and extraembryonic membranes, the remaining pups were either collected immediately or placed in a warming chamber to monitor respiratory response and collected at 20-min intervals. We collected nine new pups altogether. The first 3 pups were estimated to be between 1 h to 2 h old, although this was not precisely timed (samples 1–3 in Fig. 6c and Extended Data Fig. 12a). None of these pups had nursed at the time of collection. The next two pups were taken by C-section, decapitated and snap frozen immediately; no breaths were taken (samples 4 and 5 in Fig. 6c and Extended Data Fig. 12a). The next 4 pups were taken by C-section and used for a 'pink up' time course, collecting one pup every 20 min (that is, 20 min, 40 min, 60 min and 80 min; samples 6–9 in Fig. 6c and Extended Data Fig. 12a). During this time, all pups remained very active and working to establish a breathing rhythm. Pup 6 had not fully pinked up at time of collection, but pups 7–9 had. Pups 8 and 9 had visible lungs in their chest cavities at 60 min. The last pup collected at 80 min was fully pink with a reasonably stable breathing rhythm. No vocalization was heard from any pups during this collection. Of note, for additional quality control, we put nuclei from previously profiled E18.75 and P0 embryos into a small number of wells of the sci-RNA-seq3 experiment in which nuclei from this validation series were processed.

### Generating data using an optimized version of sci-RNA-seq3

Together with E8.5 data, which has been reported previously[8], a total of 15 sci-RNA-seq3 experiments were performed on a total of 75 mouse embryos. At least one sample was included for every 6-h interval from E8.0 to P0, and we also included embryos with as many specific somite counts as we could for the 0–34 somite range. Multiple samples were selected for a few timepoints (for example, two samples for E13.0) to boost cell numbers. Meanwhile, we tried to ensure that both male and female mice roughly alternated at adjacent timepoints (Extended Data Fig. 2j). A detailed summary and images of individual embryos can be found in Extended Data Fig. 1 and Supplementary Table. 1.

To generate the dataset, we used the optimized sci-RNA-seq3 protocol[3] as written, adjusting the volume and type of lysis buffer to the size of the embryos. In brief, frozen embryos were pulverized on dry ice and cells were lysed with a phosphate-based, hypotonic lysis buffer containing magnesium chloride, Igepal, diethyl pyrocarbonate as an RNase inhibitor, and either sucrose or bovine serum albumin (BSA). Lysate was passed over a 20-μm filter, and the nuclei-containing flow-through was fixed with a mixture of methanol and dithiobis (succinimidyl propionate) (DSP). Nuclei were rehydrated and washed in a sucrose/PBS/Triton X-100/magnesium chloride buffer (SPBSTM), then counted and distributed into 96-well plates for reverse transcription with indexed oligonucleotide-dT primers.

Age-specific adaptations were as follows. E10–E13 embryos use 5 ml BSA lysis buffer, E14 embryos use 10 ml BSA lysis buffer, E15–E18 embryos use 20 ml sucrose-based lysis buffer. Each of these samples were split over 48–96 wells for reverse transcription and the first round of indexing. A newborn P0 mouse requires 40 ml of sucrose-based lysis buffer, and the lysate is divided into 4 fractions for filtration and fixing because of the amount of tissue involved. The two P0 mice were each processed as an individual experiment and were each split over 384 wells for reverse transcription.

For the mouse samples E8.0–E9.75, we used the 'Tiny Sci' adaptation of the optimized sci-RNA-seq3[3]. Frozen embryos were gently resuspended in 100 μl lysis buffer to free the nuclei, then 400 μl of dithiobis (succinimidyl propionate)-methanol fixative was added. In the same tube, fixed nuclei were rehydrated, washed and then put directly into 8–32 wells for reverse transcription.

After reverse transcription, nuclei were pooled, washed, and redistributed into fresh 96-well plates to attach a second index sequence by ligation. Then the nuclei were pooled again, washed and redistributed into the final plates. There, the nuclei would undergo second-strand synthesis, extraction, tagmentation with Tn5 transposase and finally PCR

to add the final indexes. The PCR products were pooled, size-selected, and then the library was sequenced on an Illumina NovaSeq. For some experiments, a second NovaSeq run was necessary to capture the extent of the library complexity, so we would add more sequencing reads until the PCR duplication rate met a threshold of 50% or the median UMI count per cell went over 2,500. The validation dataset (Extended Data Fig. 4a–f) generated from 8–21-somite embryos was sequenced on an Illumina NextSeq.

## Processing of sci-RNA-seq3 sequencing reads

Data from each individual sci-RNA-seq3 experiment was processed independently. For each experiment, read alignment and gene count matrix generation was performed using the pipeline that we developed for sci-RNA-seq3[14] (https://github.com/JunyueC/sci-RNA-seq3_pipeline). In brief, base calls were converted to fastq format using Illumina's bcl2fastq v2.20 and demultiplexed based on PCR i5 and i7 barcodes using maximum likelihood demultiplexing package deML[73] with default settings. Demultiplexed reads were filtered based on the reverse transcription (RT) index and hairpin ligation adapter index (Levenshtein edit distance (ED) < 2, including insertions and deletions) and adapter-clipped using trim_galore v0.6.5 (https://github.com/FelixKrueger/TrimGalore) with default settings. Trimmed reads were mapped to the mouse reference genome (mm10) for mouse embryo nuclei using STAR v2.6.1d[74] with default settings and gene annotations (GENCODE VM12 for mouse). Uniquely mapping reads were extracted, and duplicates were removed using the UMI sequence, RT index, ligation index and read 2 end-coordinate (that is, reads with identical UMI, RT index, ligation index and tagmentation site were considered duplicates). Finally, mapped reads were split into constituent cellular indices by further demultiplexing reads using the RT index and ligation index. To generate digital expression matrices, we calculated the number of strand-specific UMIs for each cell mapping to the exonic and intronic regions of each gene with the Python v2.7.13 HTseq package[75]. For multi-mapping reads (that is, those mapping to multiple genes), the read were assigned to the gene for which the distance between the mapped location and the 3′ end of that gene was smallest, except in cases where the read mapped to within 100 bp of the 3′ end of more than one gene, in which case the read was discarded. For most analyses, we included both expected-strand intronic and exonic UMIs in per-gene single-cell expression matrices. After the single-cell gene count matrix was generated, cells with low quality (UMI < 200 or detected genes <100 or unmatched_rate (proportion of reads not mapping to any exon or intron) ≥ 0.4) were filtered out. Each cell was assigned to its originating mouse embryo on the basis of the reverse transcription barcode.

## Doublet removal

We performed three steps with the goal of exhaustively detecting and removing potential doublets. Of note, all these analyses were performed separately on data from each experiment.

First, we used Scrublet to detect doublets directly. In this step, we first randomly split the dataset into multiple subsets (six for most of the experiments) in order to reduce the time and memory requirements. We then applied the Scrublet v0.1 pipeline[76] to each subset with parameters (min_count = 3, min_cells = 3, vscore_percentile = 85, n_pc = 30, expected_doublet_rate = 0.06, sim_doublet_ratio = 2, n_neighbors = 30, scaling_method = 'log') for doublet score calculation. Cells with doublet scores over 0.2 were annotated as detected doublets.

Second, we performed two rounds of clustering and used the doublet annotations to identify subclusters that are enriched in doublets. The clustering was performed based on Scanpy v.1.6.0[20]. In brief, gene counts mapping to sex chromosomes were removed, and genes with zero counts were filtered out. Each cell was normalized by the total UMI count per cell, and the top 3,000 genes with the highest variance were selected, followed by renormalizing the gene expression matrix. The data was log-transformed after adding a pseudocount, and scaled

to unit variance and zero mean. The dimensionality of the data was reduced by PCA (30 components), followed by Louvain clustering with default parameters (resolution = 1). For the Louvain clustering, we first computed a neighbourhood graph using a local neighbourhood number of 50 using scanpy.pp.neighbors. We then clustered the cells into sub-groups using the Louvain algorithm implemented by the scanpy.tl.louvain function. For each cell cluster, we applied the same strategies to identify subclusters, except that we set resolution = 3 for Louvain clustering. Subclusters with a detected doublet ratio (by Scrublet) over 15% were annotated as doublet-derived subclusters. We then removed cells which are either labelled as doublets by Scrublet or that were included in doublet-derived subclusters. Altogether, 2.7% to 16.8% of cells in each experiment were removed by this procedure.

We found that the above Scrublet and iterative clustering-based approach has difficulty identifying doublets in clusters derived from rare cell types (for example, clusters comprising less than 1% of the total cell population), so we applied a third step to further detect and remove doublets. This step uses a different strategy to cluster and subcluster the data, and then looks for subclusters whose differentially expressed genes differ from those of their associated clusters. This step consists of a series of ten substeps. (1) We reduced each cell's expression vector to retain only protein-coding genes, long intergenic non-coding RNAs (lincRNAs) and pseudogenes. (2) Genes expressed in fewer than 10 cells and cells in which fewer than 100 genes were detected were further filtered out. (3) The dimensionality of the data was reduced by PCA (50 components) first on the top 5,000 most highly dispersed genes and then with UMAP (max_components = 2, n_neighbors = 50, min_dist = 0.1, metric = 'cosine') using Monocle 3-alpha[14]. (4) Cell clusters were identified in UMAP 2D space using the Louvain algorithm implemented in Monocle 3-alpha (resolution = $10^{-6}$). Cell partitions were detected using the partitionCells function implemented in Monocle 3-alpha. This function applies algorithms that automatically partition cells to learn disjoint or parallel trajectories based on concepts from 'approximate graph abstraction'[77]. (5) We took the cell partitions identified by Monocle 3-alpha (cell clusters were used instead for three experiments that profiled embryos before E10), downsampled each partition to 2,500 cells, and computed differentially expressed genes across cell partitions with the top_markers function of Monocle 3 (reference_cells = 1000). (6) We selected a gene set combining the top ten gene markers for each cell partition (filtering out genes with fraction_expressing <0.1 and then ordering by pseudo_R2). (7) Cells from each main cell partition were subjected to dimensionality reduction by PCA (10 components) on the selected set of top partition-specific gene markers. (8) Each cell partition was further reduced to 2D using UMAP (max_components = 2, n_neighbors = 50, min_dist = 0.1, metric = 'cosine'). (9) The cells within each partition were further sub-clustered using the Louvain algorithm implemented in Monocle 3-alpha (resolution = $10^{-4}$ for most clustering analysis). (10) Subclusters that expressed low levels of the genes that were found to be differentially expressed in step 5, had high levels of markers specific to a different partition, and had relatively high doublet scores, were labelled as doublet-derived subclusters and removed from the analysis. On average, this procedure eliminated 3.4% of cells from each experiment (range 0.5–13.2%) of the cells in each experiment (Extended Data Fig. 2a–e).

## Cell clustering and cell-type annotations

For data from individual experiments, after removing the potential doublets detected by the above three steps, we further filtered out the potential low-quality cells by investigating the numbers of UMIs and the proportion of reads mapping to the exonic regions per cell (Extended Data Fig. 2f). Then, we merged cells from individual experiments to generate the penultimate dataset, which included 15 sci-RNA-seq3 experiments and 21 runs of the Illumina NovaSeq instrument. In our early embeddings of this penultimate dataset, we noticed that one mouse embryo at E14.5 had a grossly reduced proportion of neuronal

cells. This particular sample had been divided during pulverization, and we suspect that specific anatomical portions of the frozen embryo did not make it into the experiment. We therefore removed cells from this E14.5 embryo, and we further filtered out cells from the whole dataset with doublet score (by Scrublet) > 0.15 (~0.3% of the whole dataset), as well as cells with either the percentage of reads mapping to ribosomal chromosome (Ribo%) > 5 or the percentage of reads mapping to mitochondrial chromosome (Mito%) >10 (~0.1% of the whole dataset). Finally, 11,441,407 cells from 74 embryos were retained, of which the median UMI count per cell is 2,700 and median gene count detected per cell is 1,574. For this final matrix, the number of cells recovered by each embryo and the basic quality information for cells from each sci-RNA-seq3 experiment is summarized in the Supplementary Tables 1 and 2. For sex separation and confirmation of embryos with or without sex genotyping, we counted reads mapping to a female-specific non-coding RNA (*Xist*) or chromosome Y genes (except *Erdr1* which is in both chromosome X and chromosome Y). Embryos were readily separated into females (more reads mapping to *Xist* than chromosome Y genes) and males (more reads mapping to chromosome Y genes than *Xist*).

We then applied Scanpy v.1.6.0[20] to this final dataset, performing conventional single-cell RNA-seq data processing: (1) retaining protein-coding genes, lincRNA, and pseudogenes for each cell and removing gene counts mapping to sex chromosomes; (2) normalizing the UMI counts by the total count per cell followed by log transformation; (3) selecting the 2,500 most highly variable genes and scaling the expression of each to zero mean and unit variance; (4) applying PCA and then using the top 30 principal components to calculate a neighbourhood graph (n_neighbors = 50), followed by Leiden clustering (resolution = 1); (4) performing UMAP visualization in 2D or 3D space (min.dist = 0.1). For cell clustering, we manually adjusted the resolution parameter towards modest overclustering, and then manually merged adjacent clusters if they had a limited number of DEGs relative to one another or if they both highly expressed the same literature-nominated marker genes. For each of the 26 major cell clusters identified by the global embedding, we further performed a sub-clustering with the similar strategies, except setting n_neighbors = 30 when calculating the neighbour graph and min_dist = 0.3 when performing the UMAP. Subsequently, we annotated individual cell clusters identified by the sub-clustering analysis using at least two literature-nominated marker genes per cell-type label (Supplementary Table 5).

To be clear, we have hierarchically nominated three levels of cell-type annotations in the manuscript. (1) In the global embedding involving all 11.4 M cells we identified 26 major cell clusters (Fig. 1b,c and Supplementary Table 4). (2) For individual major cell clusters, we performed sub-clustering, resulting in 190 cell types (Extended Data Fig. 3 and Supplementary Table 5). (3) For a handful of cell types, in specific parts of the manuscript, we performed further sub-clustering, to identify cell subtypes. For example: (i) we re-embedded 745,494 cells from the lateral plate and intermediate mesoderm derivatives, identifying 22 subtypes, most of which correspond to different types of mesenchymal cells (Fig. 3d and Supplementary Table 12). (ii) we re-embedded 296,020 cells (glutamatergic neurons, GABAergic neurons, spinal cord dorsal progenitors and spinal cord ventral progenitors) from stages <E13, identifying 18 different neuron subtypes (Fig. 4e and Supplementary Table 12).

Of note, we processed and analysed the birth series dataset (*n* = 962,697 nuclei after removing low-quality cells and potential doublets cells) and the early versus late somites data (*n* = 104,671 nuclei after removing low-quality cells and potential doublets cells) using exactly the same strategy, except without performing sub-clustering on each major cell cluster.

## Whole-mouse embryo analysis

Each cell was assigned to the mouse embryo from which it derived based on its reverse transcription barcode. For each of the 74 samples,

UMI counts mapping to the sample were aggregated to generate a pseudo-bulk RNA-seq profile for the sample. Each cell's counts were then normalized by dividing by its estimated size factor. The data were then log2-transformed after adding a pseudocount, and PCA was performed on the transformed data using the 3,000 most highly variable genes. The normalization and dimension reduction were performed using Monocle v3.

## Quantitatively estimating cell number for individual mouse embryo at any stage during organogenesis

To estimate the cell number of individual embryos, we selected a representative embryo from 12 timepoints at 1-day increments, from E8.5 to P0 (roughly considered as E19.5). Each embryo was digested with proteinase K overnight, and total genomic DNA was isolated with a Qiagen Puregene tissue kit (Qiagen 158063). DNA was quantified and cell number was estimated by taking the total ng of recovered DNA and assuming 2.5 billion base pairs per mouse genome (times two for a diploid cell), 650 g per mole of a base pair. Estimating cell number this way does not include any losses due to the DNA preparation, and does not count non-nucleated cells.

Based on the experimentally estimated cell numbers of those 12 embryos, we applied polynomial regression (degree = 3) to fix a curve across embryos between the embryonic day and $\log_2$-scaled cell number (adjusted $R^2 > 0.98$) (Extended Data Fig. 2l). P0 was treated as E19.5 in the model. Then, the total cell number of a whole mouse embryo at any day between E8.5 and P0 is predicted using the below formula:

$$\log_2(\text{cell number}) = 0.011369 \times \text{day}^3 - 0.583861 \times \text{day}^2 + 10.397036 \times \text{day} - 35.469755$$

To estimate the dynamic 'doubling time' of the total cell number in a whole mouse embryo, at a given timepoint (day), we took the derivative from the above formula as the $\log_2$-scaled proliferation rate $p(\text{day})$, and then calculated $24 \times 2/2^{p(\text{day})}$, resulting in a point estimate of the number of hours required for the mouse embryo to double its total cell number (Extended Data Fig. 2m).

## Characterizing transcriptional heterogeneity in the posterior embryo

We re-analysed 121,118 cells which were initially annotated as NMPs and spinal cord progenitors, mesodermal progenitors (*Tbx6*[+]), notochord, ciliated nodal cells, or gut, from embryos during the early somitogenesis (somite counts 0–34; E8–E10). Three clusters were identified, with cluster 1 dominated by NMPs and their derivatives (*n* = 98,545 cells), cluster 2 dominated by notochord and ciliated nodal cells (*n* = 3,949 cells), and cluster 3 dominated by gut cells (*n* = 18,624 cells).

To characterize transcriptional heterogeneity within each of the three cell clusters, we performed PCA on the 2,500 most highly variable genes in each cluster. Then, we calculated the Pearson correlation between the expression of the top highly variable genes and each of the top principal components within each of the three cell clusters. In brief, for each cell cluster, the top 2,500 highly variable genes were identified and their gene expression values were calculated from original UMI counts normalized to total UMIs per cell, followed by natural-log transformation and scaling. After performing Pearson correlation with the selected principal component, significant genes were identified if their correlation coefficients are less than mean − 1 × s.d. or greater than mean + 1 × s.d. of all the correlation coefficients, and false discovery rate < 0.05. In addition, we identified differentially expressed genes between early (*n* = 4,949 cells) and late (*n* = 3,910 cells) NMPs, using the FindMarkers function of Seurat v3[63], after filtering out genes that are detected in <10% of cells in both of the two populations. Significant genes were identified if their absolutely log-scaled fold changes >0.25, and adjusted *P* values < 0.05. Of note, here cells are labelled as NMPs if they are both strongly $T^+$ (raw count ≥5) and *Meis1*[−] (raw count = 0).

In Fig. 2k, the Pearson correlation coefficient between gene expression for the top highly variable genes and either PC1 of notochord (x axis) or PC1 of gut (y axis) are plotted. The overlapped genes between two cell clusters are shown as each dot, and the overlapped significant genes are highlighted in blue. The first quadrant corresponds to the inferred anterior aspect of each cluster, while the third quadrant corresponds to the inferred posterior aspect. In Fig. 2l, the log-scaled fold change of the average expression for the top highly variable genes between early versus late NMPs (x axis), and the Pearson correlation coefficient between gene expression for the top highly variable genes and PC2 of gut (y axis) are plotted. The first quadrant is associated with early somite counts for each cluster, while the third quadrant is associated with late somite counts. In the gene expression line plots in Fig. 2e, left and Fig. 2k,l, right, gene expression values were calculated from original UMI counts normalized to total UMIs per cell, followed by natural-log transformation. The line of gene expression was plotted by the geom_smooth function in ggplot2.

## Spatial mapping with Tangram

To infer the spatial origin of each lateral plate and intermediate mesoderm derivative, we used a public dataset called Mosta[46], which profiles spatial transcriptomes for 53 sections of mouse embryos spanning 8 timepoints from E9.5 to E16.5. We combined this data with our own data to perform spatial mapping analysis using Tangram[47]. In brief, for each timepoint of the Mosta data, we combined scRNA-seq data from three adjacent timepoints from our data (for example, E16.25, E16.5 and E16.75 from scRNA-seq versus E16.5 from Mosta data), and the total number of voxels within each section was randomly downsampled to 9,000 for computational efficiency. We used the Tangram with default parameters to estimate the spatial coordinates of cells from each cell type in the scRNA-seq data, and then visualized the results on the coordinates provided by Mosta. The Tangram model was trained in GPU mode using a NVIDIA A100 GPU. After applying Tangram, for each section, a cell-by-voxel matrix with mapping probabilities was returned. This matrix shows the probability that each cell originated from each voxel in the section. To reduce noise, we further smoothed the mapping probabilities for each voxel by averaging values of their k-nearest neighbouring voxels (k is calculated by natural-log-scaled total number of voxels on that section) followed by scaling it to 0 to 1 across voxels of each section. Although only selected results are presented in the paper, the mapping results for each Mosta section on which we performed this analysis are available at https://github.com/ChengxiangQiu/JAX_code/blob/main/spatial_mapping.tar.gz.

## Generating a cell-type tree for mouse development

We collected and combined scRNA-seq data from four published datasets, which consisted of 110,000 cells spanning E0 to E8.5, and the main dataset described in this paper, which consisted of 11.4 million cells spanning E8 to P0 (Supplementary Table 17). We generated the tree of cell types for mouse development via the following steps.

First, based on data source, developmental window and cell-type annotations, we split cells into fourteen subsystems which could be separately analysed and subsequently integrated. The first two subsystems correspond to the pre-gastrulation and gastrulation phases of development and are based on the external datasets[4–7]. The remaining 12 subsystems derive from the data reported here, and collectively encompass organogenesis and fetal development (Supplementary Tables 17 and 18).

Second, dimensionality reduction was performed separately on cells from each of the fourteen subsystems. Manual re-examination of each subsystem led to some corrections or refinements of cell-type annotations, ultimately resulting in 283 annotated cell-type nodes, some with only a handful of cells (for example, 60 ciliated nodal cells) and others with vastly more (for example, 650,000 fibroblasts)

(Supplementary Tables 19 and 20). Of note, each of these annotated cell-type nodes derives from one data source, such that there are some redundant annotations that facilitate 'bridging' between datasets (Extended Data Fig. 11d–h). In contrast to our previous strategy in which nodes were stage-specific[8], each cell-type node here is temporally asynchronous, and of course may also contain other kinds of heterogeneity (for example, spatial, differentiation, cell cycle and others).

Third, we sought to draw edges between nodes (Fig. 5a–f). Within each subsystem, we identified pairs of cells that were MNNs in 30-dimensional PCA space (k = 10 neighbours for pre-gastrulation and gastrulation subsystems, k = 15 for organogenesis and fetal development subsystems). Although the overwhelming majority of MNNs occurred within cell-type nodes, some MNNs spanned nodes and are presumably enriched for bona fide cell-type transitions. To approach this systematically, we calculated the total number of MNNs that spanned each possible pair of cell-type nodes within a given subsystem, normalized by the total number of possible MNNs between those nodes, and ranked all possible intra-subsystem edges based on this metric (Supplementary Table 21). Of note, due to its complexity, this was done in two stages for the 'Brain and spinal cord' subsystem, first applying the heuristic to the subset of cell types corresponding to the patterned neuroectoderm, and then again to identify edges between the patterned neuroectoderm and its derivatives (that is, neurons, glial cells and others).

Fourth, we manually reviewed the ranked list of 1,155 candidate edges for biological plausibility (those with a normalized MNN score > 1; Extended Data Fig. 11d), resulting in 452 edges which we manually annotated as more likely to correspond to either 'developmental progression' or 'spatial continuity' (Supplementary Table 22). Where nodes were connected to more than one other node, distinct subsets of cells were generally involved in each edge (Fig. 5a,b,d,e), and inter-node MNN pairs exhibited temporal coincidence (Fig. 5c,f). As only a handful of cells were profiled in the pre-gastrulation subsystem, those edges were added manually.

Finally, to bridge subsystems, we performed batch correction and co-embedding of selected timepoints from either the pre-gastrulation and gastrulation datasets, or the gastrulation and organogenesis and fetal development datasets, to identify equivalent cell-type nodes, resulting in a third category of 'dataset equivalence' edges (Extended Data Fig. 11e–h). For example, we performed anchor-based batch correction[63] followed by integration between cells from E6.5 to E8.5 generated on the 10x Genomics platform[7] (n = 108,857 cells) and the earliest 1% of this dataset (0–12 somite stage embryos) generated by sci-RNA-seq3 (n = 153,597 nuclei) (Extended Data Fig. 11e,f). This allowed us to identify 36 cell types from the integrated dataset, which we used to identify bridging edges between the gastrulation subsystem and the later subsystems (Extended Data Fig. 11g,h). Most of the 12 organogenesis and fetal development subsystems originate in cell-type nodes for which equivalent nodes are already present at gastrulation. The exceptions, presumably due to undersampling of this transition, were the 'blood' and 'PNS neuron' subsystems, for which we manually added edges to connect them with biologically plausible pseudo-ancestors. Altogether, we added 55 inter-subsystem edges.

In practice, a small number of nodes in the tree have more than one parent, so the 'tree' is formally a rooted, directed graph that represents mouse development from E0 to P0. The visualization shown in Fig. 5g was created using yFiles Hierarchical layout in Cytoscape v3.9.1. For presentation purposes, we removed most of the spatial continuity edges, except for those between spinal cord dorsal and ventral progenitors after E13.0 and GABAergic and glutamatergic neurons after E13.0. We also merged nodes with redundant labels derived from different datasets (that is, dataset equivalence edges). This resulted in a rooted graph with 262 cell-type nodes and 338 edges.

Our evaluation of the robustness of our approach to technical factors or parameter choices is provided in Extended Data Fig. 11a–c and Supplementary Note 2.

## Nominating key transcription factors and genes

The list of 1,636 mouse proteins that are putatively transcription factors was collated from AnimalTFDB v3 (http://bioinfo.life.hust.edu.cn/AnimalTFDB/)[78]. For each edge in the cell-type tree, we stratified each cell-type transition into four phases. Specifically, we identified the subset of cells within each node that were either 'inter-node' MNNs of the other cell-type or 'intra-node' MNNs of those cells. If A → B, this approach effectively models the transition as group 1 → 2 → 3 → 4 (Extended Data Fig. 11i,j). Next, we identified DETFs and genes (DEGs) across each portion of the modelled transition—that is, early (1 → 2), inter-node (2 → 3) and late (3 → 4)—by applying FindMarkers function in Seurat v3 with parameters (logfc.threshold = 0, min.pct = 0). This strategy highlights differences between cells that are most proximate to the cell-type transition itself.

After excluding dataset equivalence edges and the 'pre-gastrulation' subsystem, we nominated key transcription factors and genes that specify cell types for each of the 436 edges. Of note, the directionality of many of these edges was not immediately obvious (that is, those annotated as "spatial continuity" edges). In these cases, the orientation of the 'early' and 'late' phases is arbitrary. For edges with a relatively small number of MNN pairs, we expanded each group to at least 200 cells by iteratively including their MNNs within the same cell type, to increase statistical power.

## Identifying cell types with abrupt transcriptional changes before versus after birth

To systematically identify which cell types exhibit abrupt transcriptional changes before versus after birth, we performed the following steps.

- We focused on the 71 cell types with at least 200 cells from P0 and at least 200 cells from at least 5 timepoints prior to P0.
- We combined cells from animals collected subsequent to E16 and performed PCA based on the top 2,500 highly variable genes.
- Timepoints with at least 200 cells were selected and cells were downsampled from each timepoint to the median number of cells across those selected timepoints.
- The $k$-nearest neighbours ($k$ was adjusted for different cell types, by taking the $\log_2$-scaled median number of cells across the selected timepoints) were identified in PCA space ($n$ = 30 dimensions).
- We calculated the average proportion of nearest neighbour cells that were from a different timepoint for cells within each cell type. In this framing, a low proportion of neighbours from different timepoints corresponds to a relatively abrupt change in transcriptional state.

We subjected the birth-series dataset to a similar analysis. For each major cell cluster in the birth-series dataset, we took cells from the 6 pups delivered by C-section and calculated the Pearson correlation coefficient between the timepoint of each cell and the average timepoints of its 10 nearest neighbours identified from the global PCA embedding ($n$ = 30 dimensions). In this framing, a high correlation indicates that the cell and its nearest neighbours all underwent rapid, synchronized changes in transcriptional state.

## Reporting summary

Further information on research design is available in the Nature Portfolio Reporting Summary linked to this article.

## Data availability

The data generated in this study can be downloaded in raw and processed forms from the NCBI Gene Expression Omnibus (GEO) under accession numbers GSE186069 and GSE228590. The data are also available at https://omg.gs.washington.edu/, together with a browser that enables its visual exploration. The data are accessible for download and visualization on CELLxGENE. The published datasets analysed for this study were retrieved from either the GEO repository (GSE44183, GSE100597 and GSE109071), https://github.com/MarioniLab/EmbryoTimecourse2018 or https://db.cngb.org/stomics/mosta/ and re-processed. Published in situ hybridization images were obtained from the MGI website (https://www.informatics.jax.org/). Mouse reference genome (mm10) and gene annotations (GENCODE VM12) were used for read alignment and gene count matrix generation. Source data are provided with this paper.

## Code availability

The Python and R code used to analyse RNA-seq data is available at https://github.com/ChengxiangQiu/JAX_code.

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

**Acknowledgements** The authors thank the members of the Shendure laboratory and B. Hadland, for helpful discussions. This work was supported by the Brotman Baty Institute for Precision Medicine, a grant from Paul G. Allen Frontiers Group (Allen Discovery Center for Cell Lineage Tracing to J.S., A.F.S. and C.T.) and the National Institutes of Health (1UM1HG011586 to W.S.N., J.S and C.M.D.; R01HG010632 to J.S. and C.T.). I.C.W. and S.A.M. were supported by NIH grant UM1OD023222 and the JAX Director's Innovation Fund. E.K.N. is a Washington Research Foundation Postdoctoral Fellow. J.S. is an Investigator of the Howard Hughes Medical Institute.

**Author contributions** J.S., M.S. and C.Q. designed the research. I.C.W. collected and staged the mouse embryos. B.K.M. developed the optimized sci-RNA-seq3 protocol and generated the data (with assistance from R.M.D., T.,M.-L., E.K.N., M.L.T., O.F., D.R.O., A.R.G. and S.I.). C.Q. performed all computational analyses. X.H., S.S., W.S.N. and C.T. assisted with data analysis. I.C.W., E.K.N., X.D., C.M.D., N.H., J.C., C.B.M., D.K., A.F.S., M.S., S.A.M. and C.T. assisted with results interpretation. C.Q. and J.S. collaboratively explored and annotated the data and wrote the manuscript, except for sections corresponding to mouse collection, staging and data generation, which were written by I.C.W. and B.K.M., respectively. J.S. supervised the project.

**Competing interests** J.S. and C.T. are co-founders and scientific advisors to Scale Biosciences. J.S. is also a scientific advisory board member, consultant and/or co-founder of Cajal Neuroscience, Guardant Health, Maze Therapeutics, Camp4 Therapeutics, Phase Genomics, Adaptive Biotechnologies, Sixth Street Capital, Pacific Biosciences, and Prime Medicine. The other authors declare no competing interests.

**Additional information**
**Correspondence and requests for materials** should be addressed to Chengxiang Qiu or Jay Shendure.

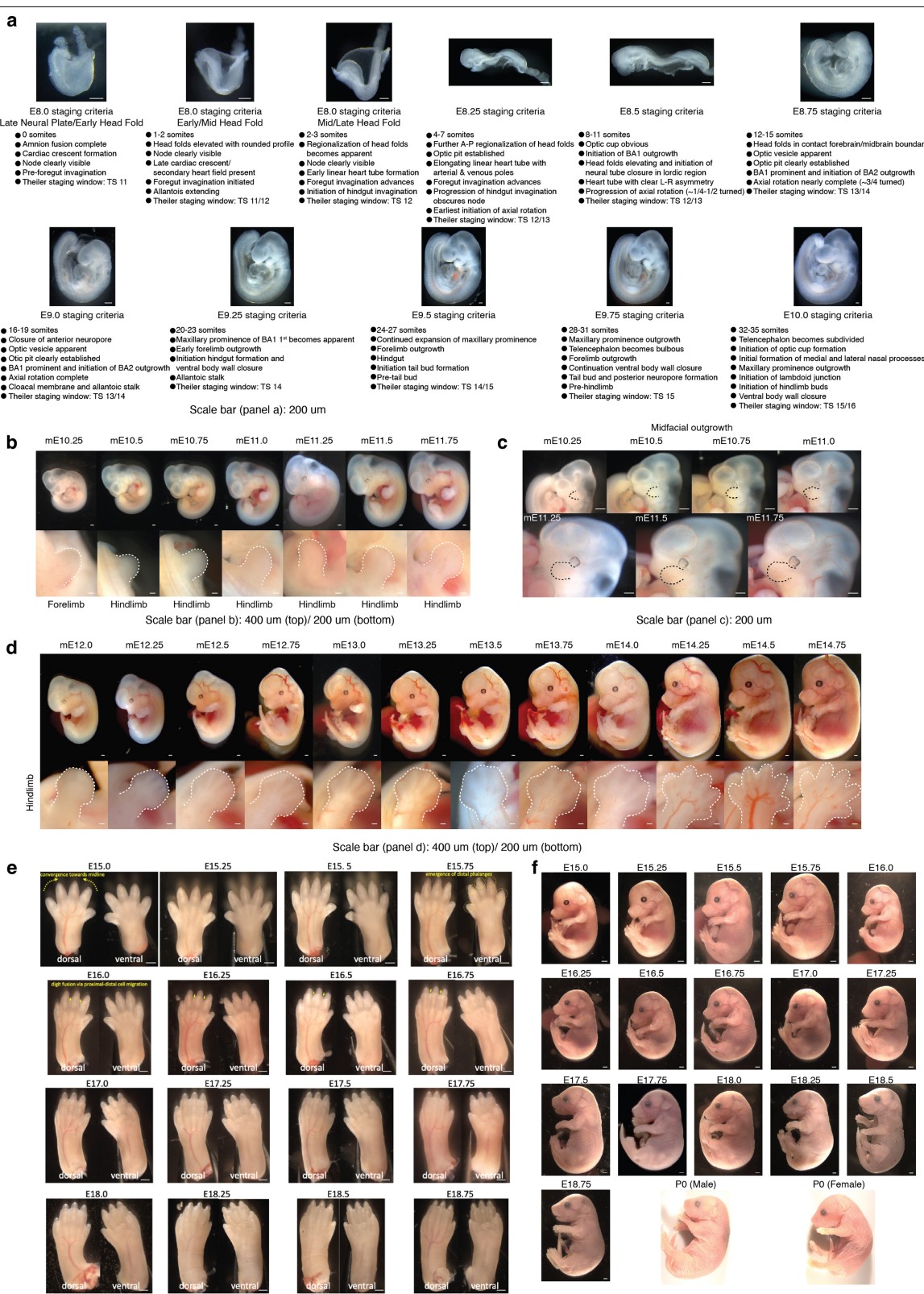

**a**

E8.0 staging criteria
Late Neural Plate/Early Head Fold
● 0 somites
● Amnion fusion complete
● Cardiac crescent formation
● Node clearly visible
● Pre-foregut invagination
● Theiler staging window: TS 11

E8.0 staging criteria
Early/Mid Head Fold
● 1-2 somites
● Head folds elevated with rounded profile
● Node clearly visible
● Late cardiac crescent/
  secondary heart field present
● Foregut invagination initiated
● Allantois extending
● Theiler staging window: TS 11/12

E8.0 staging criteria
Mid/Late Head Fold
● 2-3 somites
● Regionalization of head folds
  becomes apparent
● Node clearly visible
● Early linear heart tube formation
● Foregut invagination advances
● Initiation of hindgut invagination
● Theiler staging window: TS 12

E8.25 staging criteria
● 4-7 somites
● Further A-P regionalization of head folds
● Optic pit established
● Elongating linear heart tube with
  arterial & venous poles
● Foregut invagination advances
● Progression of hindgut invagination
  obscures node
● Earliest initiation of axial rotation
● Theiler staging window: TS 12/13

E8.5 staging criteria
● 8-11 somites
● Optic cup obvious
● Initiation of BA1 outgrowth
● Head folds elevating and initiation of
  neural tube closure in lordic region
● Heart tube with clear L-R asymmetry
● Progression of axial rotation (~1/4-1/2 turned)
● Theiler staging window: TS 12/13

E8.75 staging criteria
● 12-15 somites
● Head folds in contact forebrain/midbrain boundary
● Optic vesicle apparent
● Optic cup clearly established
● BA1 prominent and initiation of BA2 outgrowth
● Axial rotation nearly complete (~3/4 turned)
● Theiler staging window: TS 13/14

E9.0 staging criteria
● 16-19 somites
● Closure of anterior neuropore
● Optic vesicle apparent
● Otic pit clearly established
● BA1 prominent and initiation of BA2 outgrowth
● Axial rotation complete
● Cloacal membrane and allantoic stalk
● Theiler staging window: TS 13/14

E9.25 staging criteria
● 20-23 somites
● Maxillary prominence of BA1 1st becomes apparent
● Early forelimb outgrowth
● Initiation hindgut formation and
  ventral body wall closure
● Allantoic stalk
● Theiler staging window: TS 14

E9.5 staging criteria
● 24-27 somites
● Continued expansion of maxillary prominence
● Forelimb outgrowth
● Hindgut
● Initiation tail bud formation
● Pre-tail bud
● Theiler staging window: TS 14/15

E9.75 staging criteria
● 28-31 somites
● Maxillary prominence outgrowth
● Telencephalon becomes bulbous
● Forelimb outgrowth
● Continuation ventral body wall closure
● Tail bud and posterior neuropore formation
● Pre-hindlimb
● Theiler staging window: TS 15

E10.0 staging criteria
● 32-35 somites
● Telencephalon becomes subdivided
● Initiation of optic cup formation
● Initial formation of medial and lateral nasal processes
● Maxillary prominence outgrowth
● Initiation of lambdoid junction
● Initiation of hindlimb buds
● Ventral body wall closure
● Theiler staging window: TS 15/16

Scale bar (panel a): 200 um

**b**
mE10.25  mE10.5  mE10.75  mE11.0  mE11.25  mE11.5  mE11.75

Forelimb  Hindlimb  Hindlimb  Hindlimb  Hindlimb  Hindlimb  Hindlimb

Scale bar (panel b): 400 um (top)/ 200 um (bottom)

**c**
Midfacial outgrowth
mE10.25  mE10.5  mE10.75  mE11.0
mE11.25  mE11.5  mE11.75

Scale bar (panel c): 200 um

**d**
mE12.0  mE12.25  mE12.5  mE12.75  mE13.0  mE13.25  mE13.5  mE13.75  mE14.0  mE14.25  mE14.5  mE14.75

Hindlimb

Scale bar (panel d): 400 um (top)/ 200 um (bottom)

**e**
E15.0  E15.25  E15.5  E15.75
convergence towards midline    emergence of distal phalanges
dorsal  ventral

E16.0  E16.25  E16.5  E16.75
digit fusion via proximal-distal cell migration
dorsal  ventral

E17.0  E17.25  E17.5  E17.75
dorsal  ventral

E18.0  E18.25  E18.5  E18.75
dorsal  ventral

Scale bar (panel e): 200 um

**f**
E15.0  E15.25  E15.5  E15.75  E16.0
E16.25  E16.5  E16.75  E17.0  E17.25
E17.5  E17.75  E18.0  E18.25  E18.5
E18.75  P0 (Male)  P0 (Female)

Scale bar (panel f): 1 mm (except for P0 embryos)

**Extended Data Fig. 1** | See next page for caption.

**Extended Data Fig. 1 | Embryos were collected and staged based on morphological features, including somite number and limb bud geometry.**
**a**, Embryos harvested between E8 and E10 were precisely staged based upon somite counting. Harvested embryos were grouped into bins based on somite counting and further characterized based upon morphological features. Stage-representative images are shown with details of the main staging criteria for each coarse temporal bin listed. The approximately overlapping Theiler Stage (TS) is also noted for reference. Scale bar: 200 um. **b**, After E10, embryos were precisely staged based on morphological features. This was mainly done using the embryonic mouse ontogenetic staging system (eMOSS), an automated process that leverages limb bud geometry to infer developmental stage[71,72]. Staging results derived from eMOSS are designated with "mE" for morphometric embryonic day. Specifically, for each temporal bin at 6-hr increments from E10.25-E11.75, an image of a stage-representative embryo is shown in the top row. Images of each embryo's limb bud (white dashed outline) used for staging are shown in the bottom row. Scale bar: 400 um (top)/200 um (bottom). **c**, View of the craniofacial region of embryos shown in panel b demonstrates that limb bud staging also recreates the ordered ontogenetic progression of craniofacial morphogenesis, including development of the brain, eye, and outgrowth of facial prominences (black dashed line highlights maxillary process). Scale bar: 200 um. **d**, For each temporal bin at 6-hr increments from E12.0-E14.25, an image of a randomly selected embryo is shown in the top row. The subview of its hindlimb is shown in the bottom row. Scale bar: 400 um (top)/200 um (bottom). **e**, eMOSS is able to stage E10.25-E14.75, after which limb morphology becomes too complex. We defined additional dynamics related to digit formation to stage E15.0-E16.75 embryos. However, the remaining timepoints (E17.0-E18.75) were staged based upon gestational age. For each temporal bin at 6-hr increments from E15.0-E18.75, an image of the hindlimbs of a randomly selected embryo is shown. Scale bar: 200 um. **f**, For each temporal bin at 6-hr increments from E15.0-P0, an image of a stage-representative embryo is shown. Scale bar: 1 mm (except for P0 embryos).

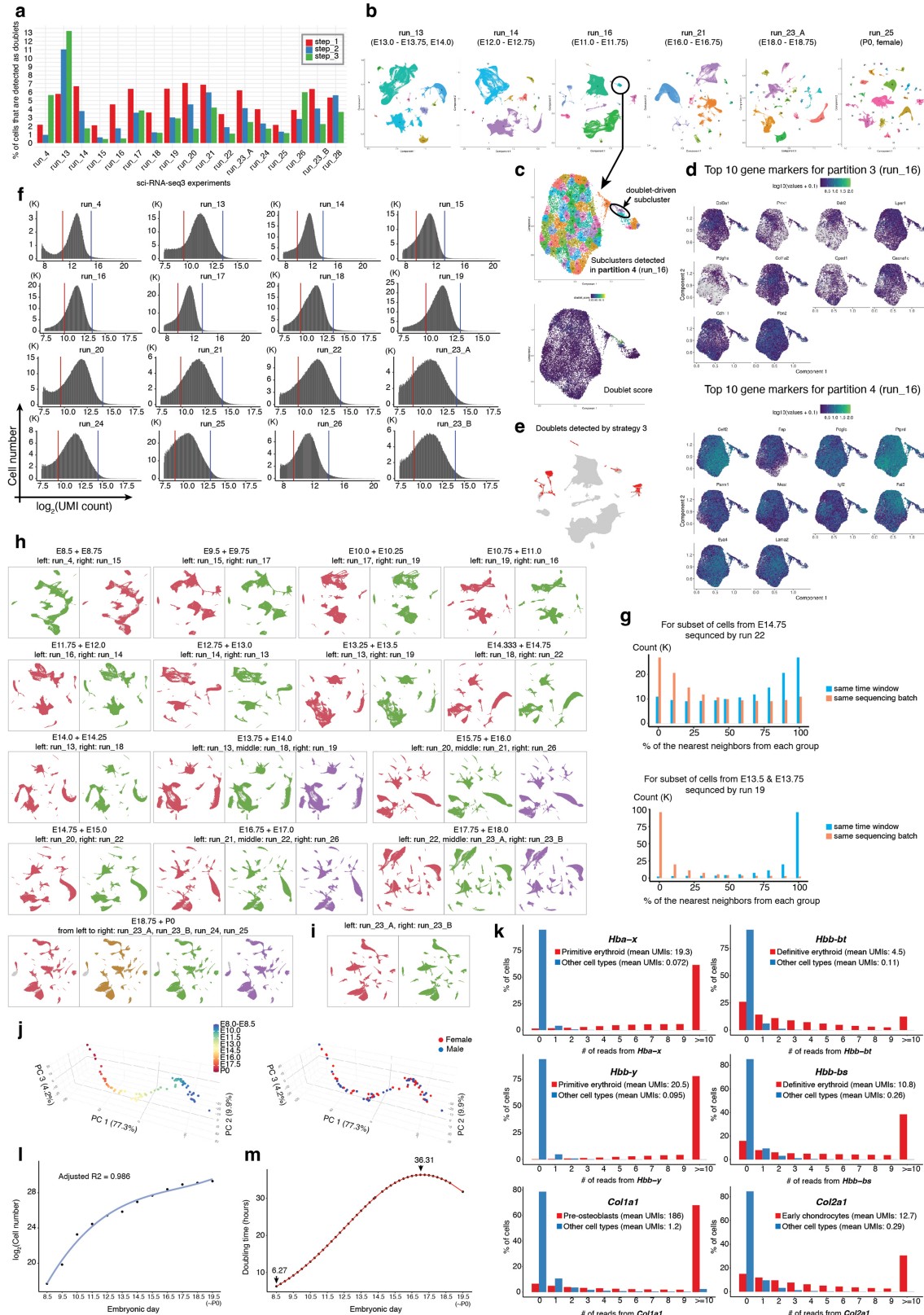

**Extended Data Fig. 2** | See next page for caption.

**Extended Data Fig. 2 | Quality control on sci-RNA-seq3 experiments. a**, We performed three steps to detect and remove potential doublets from each single sci-RNA-seq3 experiment. First, we used *Scrublet* to calculate a doublet score for each cell. Cells with a doublet score over 0.2 were annotated as detected doublets. Second, we clustered and subclustered the entire dataset. Subclusters with a detected doublet ratio over 15% were annotated as doublet-derived subclusters. Third, after removing doublets detected by the first two steps, we performed clustering again to identify the major cell partitions (*i.e.* disjoint trajectories). Three experiments (runs 4, 15, and 17) that profiled embryos before E10 used cell clusters instead of cell partitions. We then generated a union gene list by combining the top 10 differentially expressed genes from each cell partition. This gene list was used to perform subclustering on each cell partition. Subclusters that showed low expression of target cell partition-specific markers and enriched expression of non-target cell cluster-partition markers were identified as doublet-driven clusters. More details are provided in the **Methods**. The percentage of cells detected and removed as doublets by each of the three steps in individual sci-RNA-seq3 experiments is shown. **b**, The labeled cell partitions for each of six selected experiments are shown, after removing doublets from the first two steps. **c**, Example of detection of doublet-driven subclusters via step 3. Re-embedded 2D UMAP of cells from partition 4 of experiment run_16, with cells colored by subclusters. The same UMAP is shown below, with cells colored by doublet score calculated by *Scrublet*. **d**, The same UMAP as in panel c, colored by the normalized gene expression of the top 10 differentially expressed genes in either partition 3 (top) or partition 4 (bottom). **e**, The same UMAP as experiment run_16 in panel b, highlighted by doublets detected in step 3 (red). **f**, Histograms of $\log_2$(UMI count) per single nucleus for each of 15 sci-RNA-seq3 experiments. For the 14 newly performed experiments (run_13 to run_26), upper (blue line) and lower (red line) thresholds used for quality filtering correspond to the mean plus 2 standard deviations and mean minus 1 standard deviation of log2-scaled values, respectively, after excluding cells with >85% of reads mapping to exonic regions (except for the lower bound of 500, which was manually assigned for run_25), are shown with vertical lines. The data of run_4, which was reported previously[8], was subjected to the same thresholds used in the original study, *i.e.* the mean +/− 2 standard deviations of log2-scaled values (blue and red vertical lines, respectively), after excluding cells with >85% of reads mapping to exonic regions. Run_23_A & B were from the same sci-RNA-seq3 experiment, but with nuclei which were sequenced separately. **g**, Although most of the embryos from the same approximate stage (*e.g.* E14.0-E14.75) were included in the same sci-RNA-seq3 experiment (Supplementary Table 1), we profiled extra nuclei in some experiments for a handful of timepoints to ensure sufficient coverage.

Here we sought to leverage those instances to check for potential batch effects across experiments. For this, on the embedding learned from all of the data, we asked whether these cells' profiles are more similar to cells from the same experiment or, alternatively, cells from the same time window. Top: for a random subset of cells from E14.75 which were profiled in experiment run_22 (primarily E17.0-E17.75), we performed a *k*-nearest neighbors (*k*NN, *k* = 10) approach in the global 3D UMAP to find the nearest neighboring cells either from the same experiment (red) or the same time window (E14.0-E14.75) but different experiment (blue). The percentages of the nearest neighboring cells from the two groups for individual cells are presented in the histogram. Bottom: a similar analysis was performed for a random subset of cells from E13.5 & E13.75 which were profiled in experiment run_19 (primarily E10.5-E11). In both examples, we observe that nearest neighbors are overwhelmingly cells from a different experiment (but the same time window), rather than cells from the same experiment (but a different time window). **h**, Cells processed in different experiments are well-integrated without batch correction. To further check for potential batch effects, we generated co-embeddings of samples processed from adjacent timepoints in different experiments, without batch correction. **i**, We also generated a co-embedding of cells from run_23_A (red) and run_23_B (green), which derived from the same sci-RNA-seq3 experiment but were sequenced on different NovaSeq runs. **j**, Embeddings of pseudo-bulk RNA-seq profiles of 74 mouse embryos in PCA space with visualization of top three PCs. Embryos are colored by either developmental stage (left) or data-inferred sex (right). **k**, Ambient noise (*e.g.* as might be due to transcript leakage) was assessed by examining hemoglobin and collagen transcripts. The distribution of the number of reads mapping to each selected hemoglobin or collagen gene across cells, for the cell type that is expected to express that gene at high levels (red) vs. all other cell types (blue). The mean UMI counts of cells in each group are also reported. The overall levels of ambient noise as assessed by these transcripts was low, *e.g.* the mean number of UMIs for *Hbb-bs* was 10.8 in definitive erythroid cells and 0.26 in all other cells, and for *Col1a1* was 186 in pre-osteoblasts vs. 1.23 in all other cells. **l**, Quantitatively estimating cell number for individual mouse embryos as a function of developmental stage. Based on the experimentally estimated cell numbers of the 12 embryos (ranging from E8.5 to P0), we applied polynomial regression (degree = 3) to fix a curve across embryos between the embryonic day and log2-scaled cell number. P0 was treated as E19.5 in the model. **m**, The estimated "doubling time" of the total cell number in a whole mouse embryo are plotted as a function of timepoints. The timepoints with the longest (E17.0) and shortest (E8.5) estimated "doubling times" are highlighted.

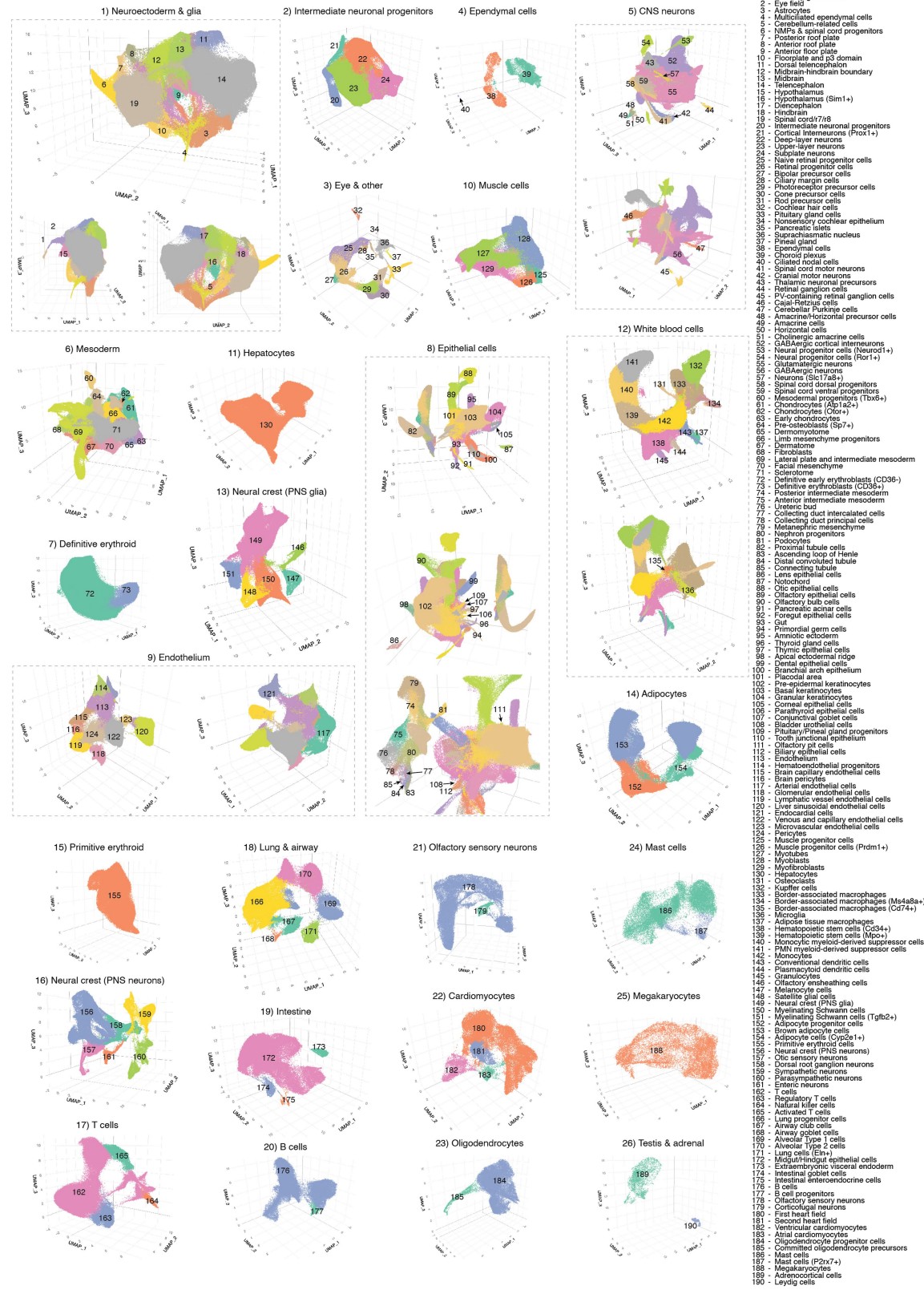

**Extended Data Fig. 3 | Cell type annotations.** For each of the 26 major cell clusters, we performed subclustering and then annotated each of 190 subclusters using at least two literature-nominated marker genes per cell type label (Supplementary Table 5).

1 - Retinal pigment cells
2 - Eye field
3 - Astrocytes
4 - Multiciliated ependymal cells
5 - Cerebellum-related cells
6 - NMPs & spinal cord progenitors
7 - Posterior roof plate
8 - Anterior roof plate
9 - Anterior floor plate
10 - Floorplate and p3 domain
11 - Dorsal telencephalon
12 - Midbrain-hindbrain boundary
13 - Midbrain
14 - Telencephalon
15 - Hypothalamus
16 - Hypothalamus (Sim1+)
17 - Diencephalon
18 - Hindbrain
19 - Spinal cord/r7/r8
20 - Intermediate neuronal progenitors
21 - Cortical interneurons (Prox1+)
22 - Deep-layer neurons
23 - Upper-layer neurons
24 - Subplate neurons
25 - Naive retinal progenitor cells
26 - Retinal progenitor cells
27 - Bipolar precursor cells
28 - Ciliary margin cells
29 - Photoreceptor precursor cells
30 - Cone precursor cells
31 - Rod precursor cells
32 - Cochlear hair cells
33 - Pituitary gland cells
34 - Nonsensory cochlear epithelium
35 - Pancreatic islets
36 - Suprachiasmatic nucleus
37 - Pineal gland
38 - Ependymal cells
39 - Choroid plexus
40 - Ciliated nodal cells
41 - Spinal cord motor neurons
42 - Cranial motor neurons
43 - Thalamic neuronal precursors
44 - Retinal ganglion cells
45 - PV-containing retinal ganglion cells
46 - Cajal-Retzius cells
47 - Cerebellar Purkinje cells
48 - Amacrine/Horizontal precursor cells
49 - Amacrine cells
50 - Horizontal cells
51 - Cholinergic amacrine cells
52 - GABAergic cortical interneurons
53 - Neural progenitor cells (Neurod1+)
54 - Neural progenitor cells (Ror1+)
55 - Glutamatergic neurons
56 - GABAergic neurons
57 - Neurons (Slc17a8+)
58 - Spinal cord dorsal progenitors
59 - Spinal cord ventral progenitors
60 - Mesodermal progenitors (Tbx6+)
61 - Chondrocytes (Atp1a2+)
62 - Chondrocytes (Otor+)
63 - Early chondrocytes
64 - Pre-osteoblasts (Sp7+)
65 - Dermomyotome
66 - Limb mesenchyme progenitors
67 - Dermatome
68 - Fibroblasts
69 - Lateral plate and intermediate mesoderm
70 - Facial mesenchyme
71 - Sclerotome
72 - Definitive early erythroblasts (CD36-)
73 - Definitive erythroblasts (CD36+)
74 - Posterior intermediate mesoderm
75 - Anterior intermediate mesoderm
76 - Ureteric bud
77 - Collecting duct intercalated cells
78 - Collecting duct principal cells
79 - Metanephric mesenchyme
80 - Nephron progenitors
81 - Podocytes
82 - Proximal tubule cells
83 - Ascending loop of Henle
84 - Distal convoluted tubule
85 - Connecting tubule
86 - Lens epithelial cells
87 - Notochord
88 - Otic epithelial cells
89 - Olfactory epithelial cells
90 - Olfactory bulb cells
91 - Pancreatic acinar cells
92 - Foregut epithelial cells
93 - Gut
94 - Primordial germ cells
95 - Amniotic ectoderm
96 - Thyroid gland cells
97 - Thymic epithelial cells
98 - Apical ectodermal ridge
99 - Dental epithelial cells
100 - Branchial arch epithelium
101 - Placodal area
102 - Pre-epidermal keratinocytes
103 - Basal keratinocytes
104 - Granular keratinocytes
105 - Corneal epithelial cells
106 - Parathyroid epithelial cells
107 - Conjunctival goblet cells
108 - Bladder urothelial cells
109 - Pituitary/Pineal gland progenitors
110 - Tooth junctional epithelium
111 - Olfactory pit cells
112 - Biliary epithelial cells
113 - Endothelium
114 - Hematoendothelial progenitors
115 - Brain capillary endothelial cells
116 - Brain pericytes
117 - Arterial endothelial cells
118 - Glomerular endothelial cells
119 - Lymphatic vessel endothelial cells
120 - Liver sinusoidal endothelial cells
121 - Endocardial cells
122 - Venous and capillary endothelial cells
123 - Microvascular endothelial cells
124 - Pericytes
125 - Muscle progenitor cells
126 - Muscle progenitor cells (Prdm1+)
127 - Myotubes
128 - Myoblasts
129 - Myofibroblasts
130 - Hepatocytes
131 - Kupffer cells
132 - Border-associated macrophages
133 - Border-associated macrophages (Ms4a8a+)
134 - Border-associated macrophages (Cd74++)
135 - Microglia
136 - Adipose tissue macrophages
137 - Hematopoietic stem cells (Cd34+)
138 - Hematopoietic stem cells (Mpo+)
139 - Monocyte myeloid-derived suppressor cells
140 - Monocyte myeloid-derived suppressor cells
141 - PMN myeloid-derived suppressor cells
142 - Monocytes
143 - Conventional dendritic cells
144 - Plasmacytoid dendritic cells
145 - Granulocytes
146 - Olfactory ensheathing cells
147 - Melanocyte cells
148 - Satellite glial cells
149 - Neural crest (PNS glia)
150 - Myelinating Schwann cells
151 - Myelinating Schwann cells (Tgfb2+)
152 - Adipocyte progenitor cells
153 - Brown adipocyte cells
154 - Adipocyte cells (Cyp2e1+)
155 - Primitive erythroid cells
156 - Neural crest (PNS neurons)
157 - Otic sensory neurons
158 - Dorsal root ganglion neurons
159 - Sympathetic neurons
160 - Parasympathetic neurons
161 - Enteric neurons
162 - T cells
163 - Regulatory T cells
164 - Natural killer cells
165 - Activated T cells
166 - Lung progenitor cells
167 - Airway club cells
168 - Airway goblet cells
169 - Alveolar Type 1 cells
170 - Alveolar Type 2 cells
171 - Lung cells (Eln+)
172 - Midgut/Hindgut epithelial cells
173 - Extraembryonic visceral endoderm
174 - Intestinal goblet cells
175 - Intestinal enteroendocrine cells
176 - B cells
177 - B cell progenitors
178 - Olfactory sensory neurons
179 - Corticofugal neurons
180 - First heart field
181 - Second heart field
182 - Ventricular cardiomyocytes
183 - Atrial cardiomyocytes
184 - Oligodendrocyte progenitor cells
185 - Committed oligodendrocyte precursors
186 - Mast cells
187 - Mast cells (P2rx7+)
188 - Megakaryocytes
189 - Adrenocortical cells
190 - Leydig cells

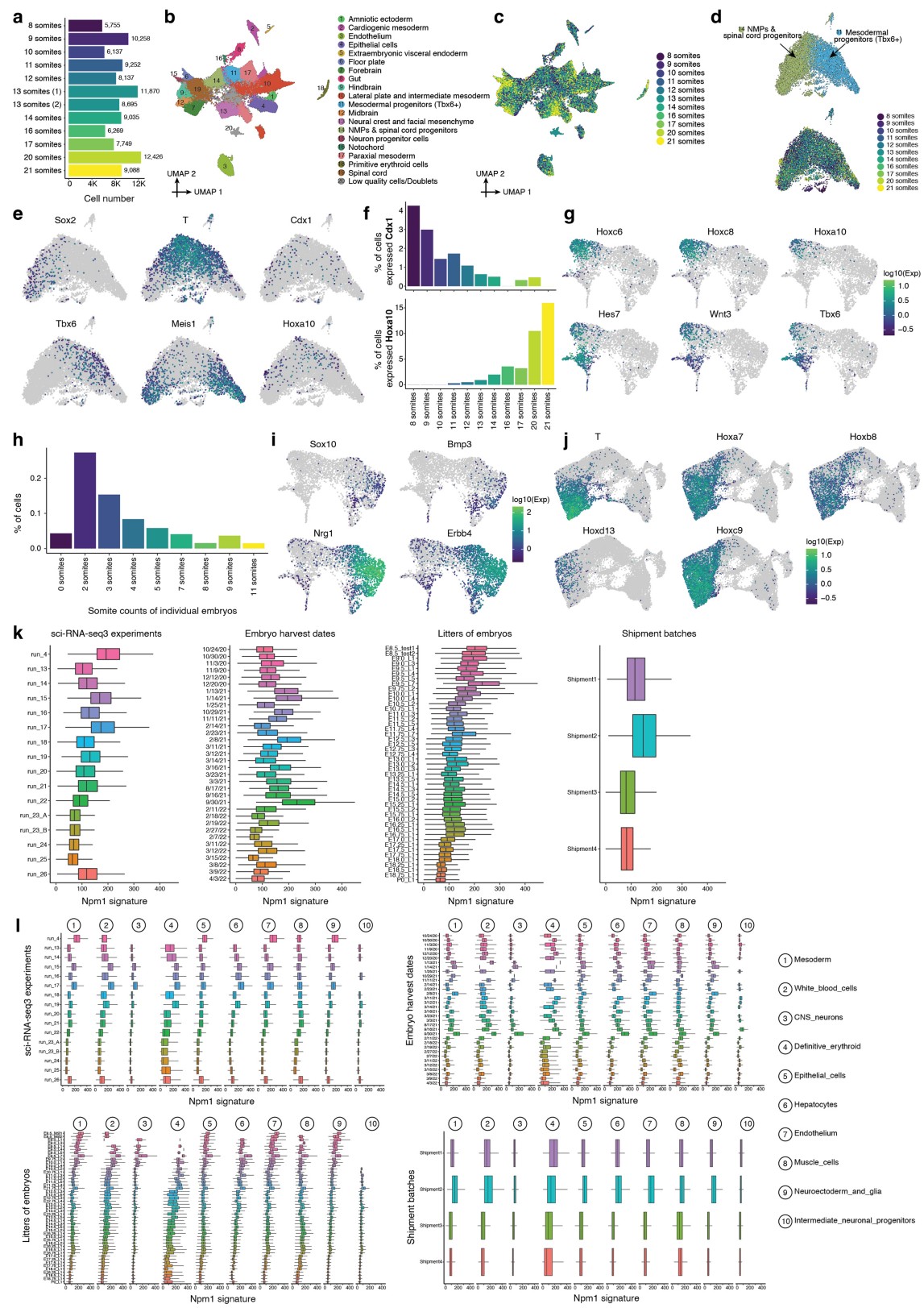

**Extended Data Fig. 4 |** See next page for caption.

**Extended Data Fig. 4 | Transcriptional heterogeneity in the posterior embryo during early somitogenesis. a**, A validation sci-RNA-seq3 dataset of mouse embryos from somites 8 to 21. To validate findings related to differences between embryos staged with early vs. late somite counts, particularly in NMPs, we profiled another 12 precisely staged mouse embryos, ranging from 8 to 21 somites, in an independent sci-RNA-seq3 experiment. The resulting library was sequenced on an Illumina NextSeq 2000, resulting in 104,671 cells in total, with a median UMI count of 513 and a median gene count of 446 per cell. The number of cells profiled from each embryo. **b**, 2D UMAP visualization of the validation dataset (all cell types). **c**, The same UMAP as in panel b, with cells colored by somite count of the originating embryo. **d**, Re-embedded 2D UMAP of 9,686 cells from NMPs & spinal cord progenitors (cluster 11) and mesodermal progenitors (*Tbx6* +) (cluster 14) in panel b. Cells are colored by either the original annotation (top) or somite count (bottom). **e**, The same UMAP as in panel d, colored by gene expression of marker genes which appear specific to different subpopulations of NMPs: column 1: differences between neuroectodermal (*Sox2* +) vs. mesodermal (*Tbx6* +) fates; column 2: the differentiation of bipotential NMPs (*T* +, *Meis1*-) towards either fate; column 3: earlier (*Cdx1* +) vs. later (*Hoxa10* +) NMPs. References for marker genes are provided in Supplementary Table 12. **f**, Within the cells shown in panel d, the proportion of cells (y-axis) which express either *Cdx1* (top) or *Hoxa10* (bottom) are plotted as a function of somite count of the originating embryo. **g**, Transcriptional heterogeneity in the posterior embryo during the early somitogenesis. The same UMAP as in Fig. 2g, colored by gene expression of marker genes which appear specific to the subpopulation of notochord cluster that is *Noto* +, including posterior *Hox* genes (*Hoxc6*, *Hoxc8*, *Hoxa10*), and genes involved in Notch signaling (*Hes7*), Wnt signaling (*Wnt3*) and mesodermal differentiation (*Tbx6*). **h**, Cell proportions falling into the ciliated nodal cell cluster for embryos with different somite counts. **i**, The same UMAP as in Fig. 2g, colored by gene expression of marker genes which appear specific to the subpopulation of the notochord *Noto*- and more strongly *Shh* +, including *Sox10*, *Bmp3*, *Nrg1*, and *Erbb4*. **j**, The same UMAP as in Fig. 2i, colored by gene expression of marker genes which appear specific to the posterior gut endoderm, including *T*, *Hoxa7*, *Hoxb8*, *Hoxd13*, and *Hoxc9*. **k**, Checking the consistency of *Npm1* signatures across different batches. First, we downsampled the dataset to ~1 M cells using *geosketch*[79] and performed k-means clustering to ensure that each cluster contained roughly 500 cells. Second, we aggregated UMI counts for cells within each cluster to generate 2,289 meta-cells, and normalized the UMI counts for each meta-cell followed by log2-transformation. Third, we performed Pearson correlation between each protein-coding gene and *Npm1*, and selected genes with correlation coefficients > 0.6 (738 genes, ~3% of the total protein coding genes). A gene set enrichment analysis suggests that the module is associated with RNP complexes (corrected p-value = 1.4e-105), cytoplasmic translation (corrected p-value = 2.8e-90), and ribosomal proteins (corrected p-value = 7.4e-71). Finally, we summed the normalized UMI counts of these genes to calculate a *Npm1* signature for individual cells. The resulting *Npm1* signatures are subsetted in four plots, from left to right: by sci-RNA-seq3 experiment, embryo harvest date, litter of embryos, or shipment batch. **l**, Same as panel k, but further stratified by the top 10 abundant major cell clusters. Boxplots, in panel k (n = 1,144,141 cells) and l (n = 299,725 cells in Mesoderm, n = 127,150 cells in White blood cells, n = 104,205 cells in CNS neurons, n = 73,005 cells in Definitive erythroid, n = 66,772 cells in Epithelial cells, n = 64,845 cells in Hepatocytes, n = 62,951 cells in Endothelium, n = 61,249 cells in Muscle cells, n = 52,748 cells in Neuroectoderm and glia, n = 45,940 cells in Intermediate neuronal progenitors), represent IQR (25th, 50th, 75th percentile) with whiskers representing 1.5× IQR.

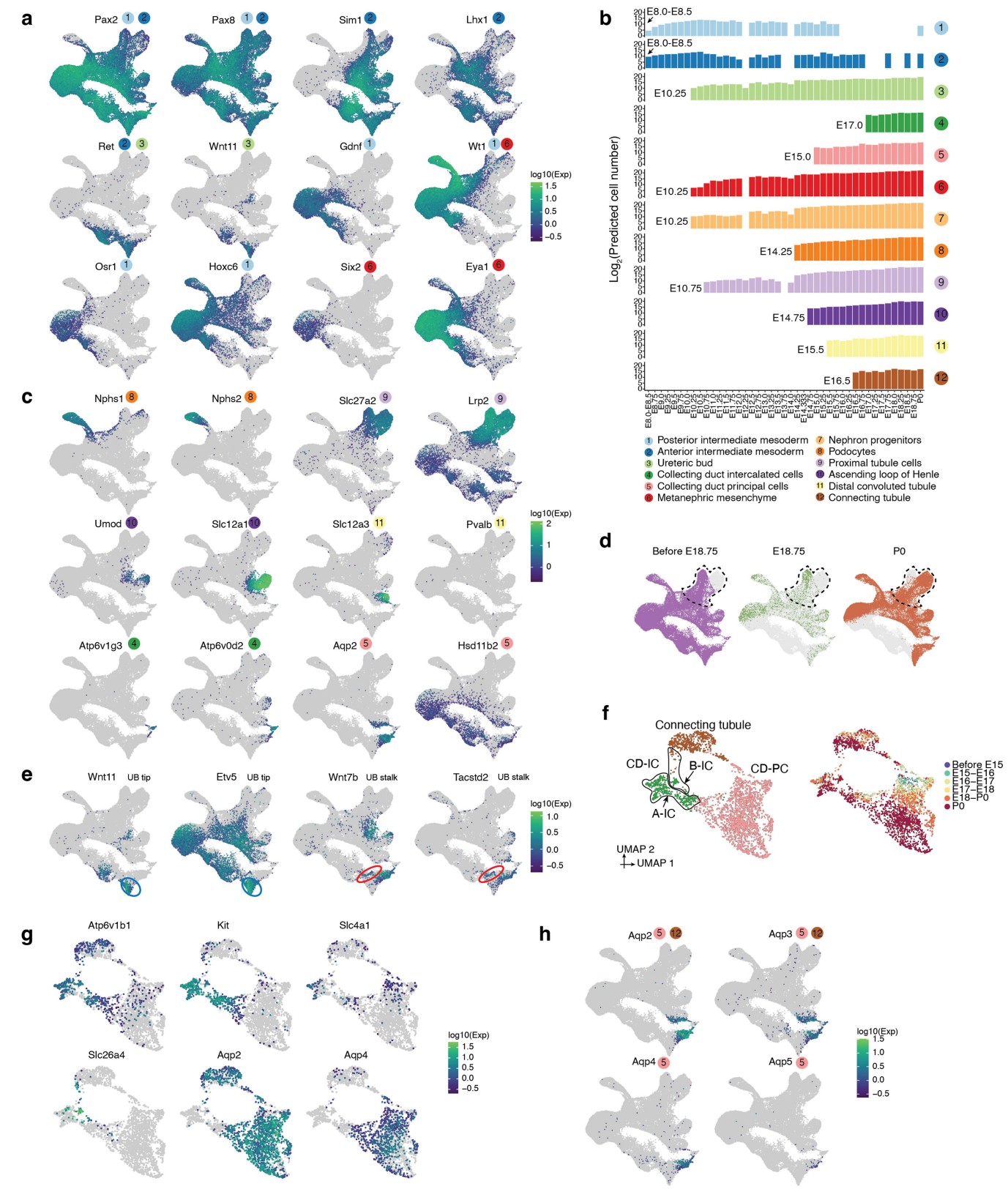

**Extended Data Fig. 5** | See next page for caption.

**Extended Data Fig. 5 | Transcriptional heterogeneity in renal development.**
**a**, The same UMAP as in Fig. 3a, colored by expression of marker genes which appear specific to anterior intermediate mesoderm (*Pax2*+, *Pax8*+, *Sim1*+, *Lhx1*+, *Ret*+), posterior intermediate mesoderm (*Pax2*+, *Pax8*+, *Gdnf1*+, *Wt1*+, *Osr1*+, *Hoxc6*+), ureteric bud (*Ret*+, *Wnt11*+) or metanephric mesenchyme (*Wt1*+, *Six2*+, *Eya1*+). References for marker genes are provided in Supplementary Table 5. **b**, The predicted absolute number (log2 scale) of cells of each renal cell type at each timepoint. The predicted absolute number was calculated by the product of its sampling fraction in the overall embryo and the predicted total number of cells in the whole embryo at the corresponding timepoint (Fig. 1b). For each row, the first timepoint with at least 10 cells assigned that cell type annotation is labeled, and all observations prior to that timepoint are discarded. **c**, The same UMAP as in Fig. 3a, colored by expression of marker genes which appear specific to podocytes (*Nphs1*+, *Nphs2*+), proximal tubule cells (*Slc27a2*+, *Lrp2*+), ascending loop of Henle (*Umod*+, *Slc12a1*+), distal convoluted tubule (*Slc12a3*+, *Pvalb*+), collecting duct intercalated cells (*Atp6v1g3*+, *Atp6v0d2*+) or collecting duct principal cells (*Aqp2*+, *Hsd11b2*+). References for marker genes are provided in Supplementary Table 5. **d**, The same UMAP as Fig. 3a is shown three times, with colors highlighting cells from before E18.75 (left), E18.75 (middle), or P0 (right). Dotted cycles highlight cells which appear to correspond to the proximal tubule. **e**, The same UMAP as in Fig. 3a, colored by expression of marker genes which appear specific to the ureteric bud tip (*Wnt11*+, *Ret*+, *Etv5*+) or stalk (*Wnt7b*+, *Tacstd2*+)[40]. Ureteric bud tip and stalk are highlighted by blue and red circles, respectively. **f**, Re-embedded 2D UMAP of 2,894 cells from connecting tubule cells, collecting duct principal cells (CD-PC), and collecting duct intercalated cells (CD-IC). Cells are colored by either their initial annotations (top) or timepoint (bottom). Black circles highlight the cells which appear to be either type A (A-IC) or type B (B-IC) intercalated cells. **g**, The same UMAP as in panel f, colored by expression of marker genes specific to CD-IC (*Atp6v1b1*+), A-IC (*Kit*+, *Slc4a1*+), B-IC (*Slc26a4*+), CD-PC (*Aqp2*+, *Aqp4*+), and connecting tubule (*Aqp2*+, *Aqp4*-) (Supplementary Table 12). **h**, The same UMAP as in Fig. 3a, colored by expression of marker genes which appear specific to connecting tubule cells (*Aqp2*+, *Aqp3*+, *Aqp4*-) or collecting duct cells (*Aqp2*+, *Aqp3*+, *Aqp4*+, *Aqp5*+)[80].

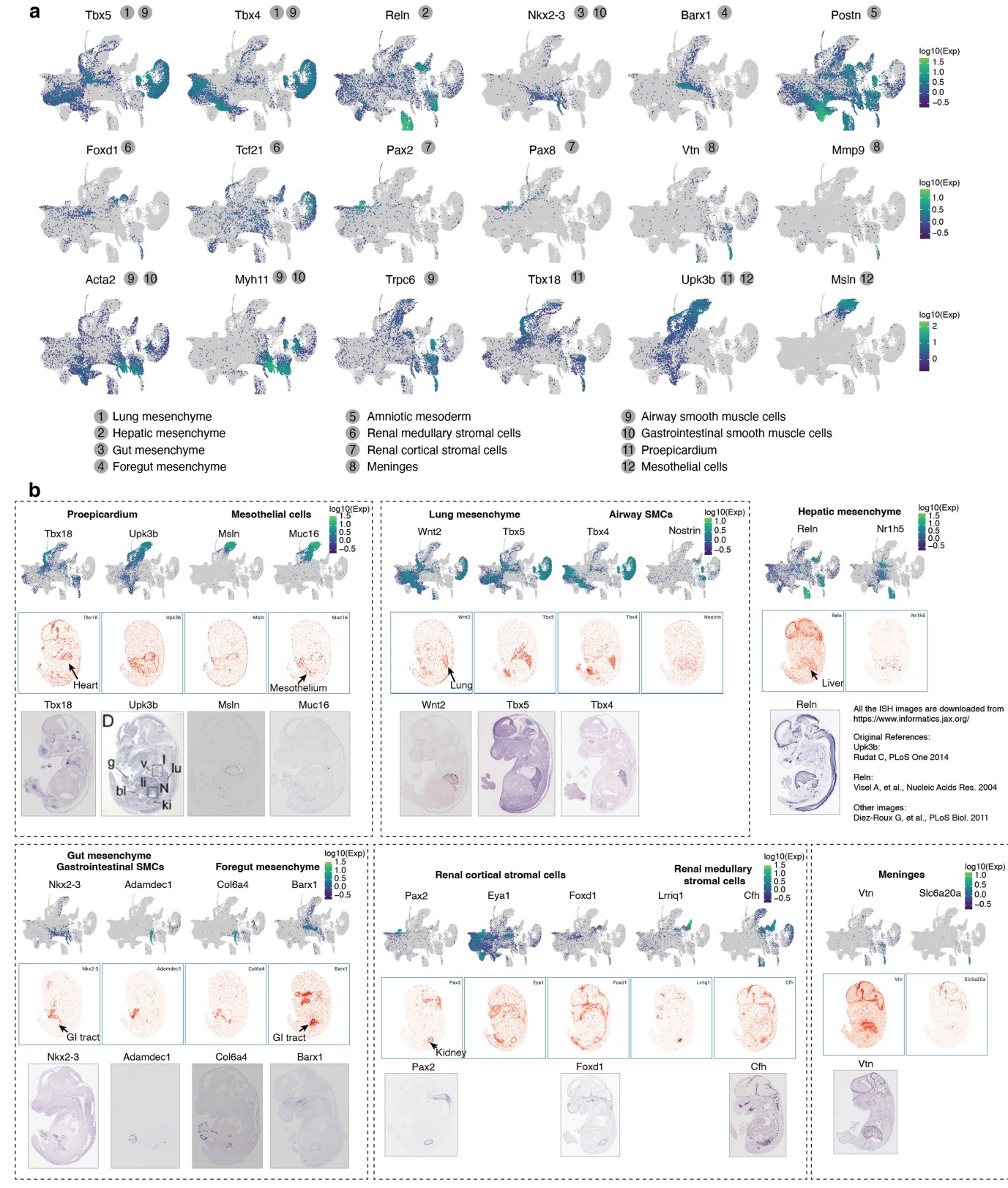

**Extended Data Fig. 6** | See next page for caption.

**Extended Data Fig. 6 | Transcriptional heterogeneity in mesenchyme.**
**a**, The same UMAP as in Fig. 3d, colored by expression of marker genes which appear specific to lung mesenchyme (*Tbx5* +, *Tbx4* +), hepatic mesenchyme (*Reln* +), gut mesenchyme (*Nkx2-3* +), foregut mesenchyme (*Barx1* +), amniotic mesoderm (*Postn* +), renal medullary stromal cells (*Foxd1* +, *Tcf21* +), renal cortical stromal cells (*Pax2* +, *Pax8* +), meninges (*Vtn* +), airway smooth muscle cells (*Trpc6* +, *Tbx5* +), gastrointestinal smooth muscle cells (*Nkx2-3* +), proepicardium or mesothelium (*Msln* +). References for marker genes are provided in Supplementary Table 12. **b**, Published in situ hybridization (ISH) images support our annotations of lateral plate and intermediate mesoderm derivatives. In each subpanel (defined by dotted rectangles), three rows are shown for one or two lateral plate and intermediate mesoderm derivative cell types. Notably, each of these cell types was annotated based on spatial mapping analysis, as shown in Fig. 3e. Top: The same UMAP as in Fig. 3d, colored by gene expression of marker genes which appear specific to the given cell type. Middle: Virtual in situ hybridization (ISH) images of individual genes from one selected section (E1S1) from E14.5 of the *Mosta* data (https://db.cngb.org/stomics/mosta/). Bottom: In situ hybridization (ISH) images of individual genes were obtained from the Jackson Laboratory Mouse Genome Informatics (MGI) website (https://www.informatics.jax.org/). The original reference for these ISH images are[81–83].

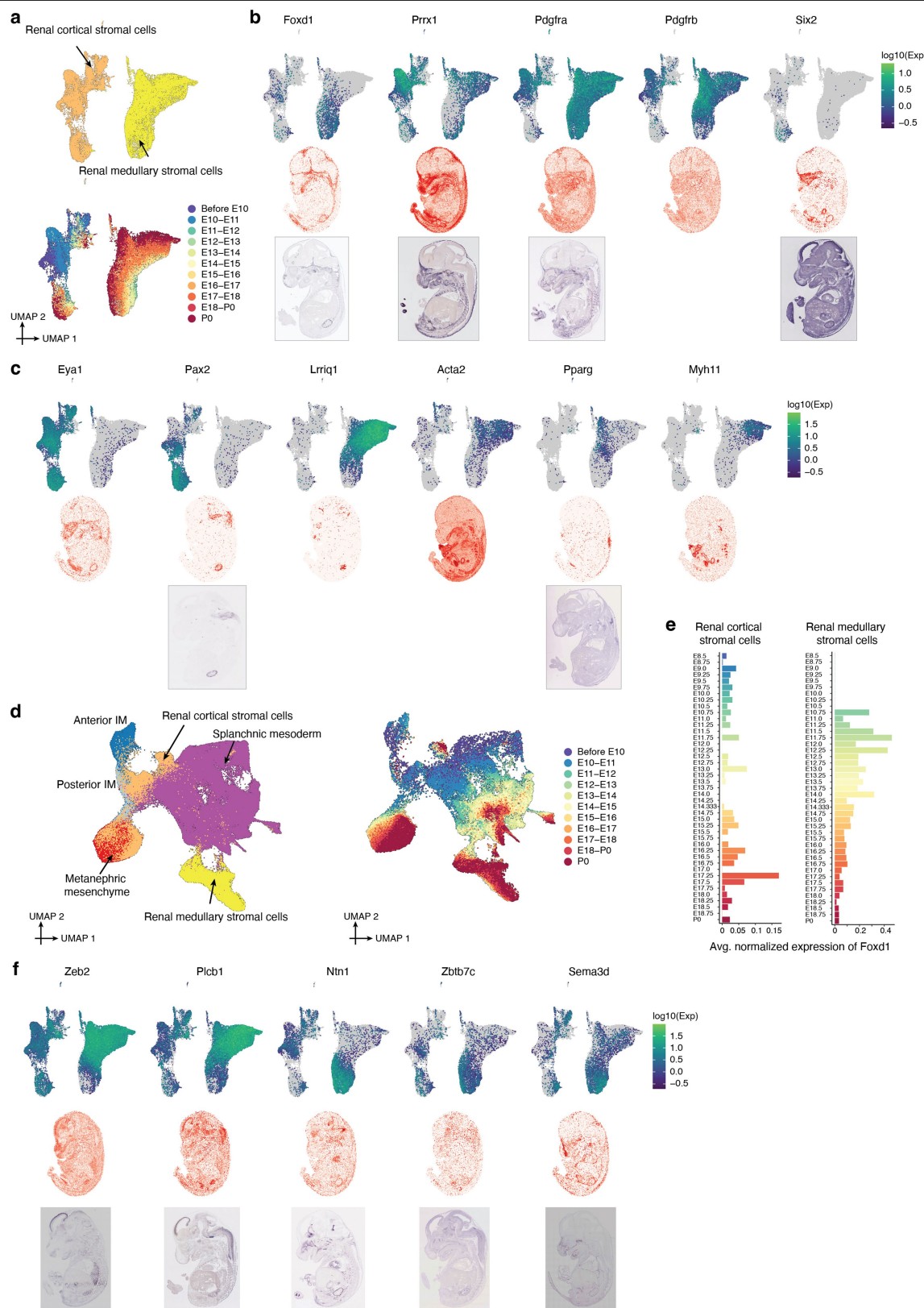

**Extended Data Fig. 7 |** See next page for caption.

**Extended Data Fig. 7 | Assessing the potential origins of LPM subsets annotated as renal cortical & medullary stromal cells. a**, Re-embedded 2D UMAP of 39,468 cells from renal cortical & medullary stromal cells. Cells are colored by either annotation (top) or timepoint (bottom, after downsampling to a uniform number of cells per time window). **b**, Top: The same UMAP as in panel a, colored by gene expression of marker genes which appear specific to renal cortical & medullary stromal cells. Both cell types express *Foxd1*, *Prrx1*, *Pdgfra*, and *Pdgfrb*, but only renal cortical stromal cells express *Six2*. Middle: Virtual in situ hybridization (ISH) images of individual genes. Bottom: ISH images of individual genes. **c**, Top: The same UMAP as in panel a, colored by gene expression of marker genes which appear specific to renal cortical stromal cells (*Eya1*+, *Pax2*+), and renal medullary stromal cells (*Lrriq1*+, *Acta2*+, *Pparg*+, *Myh11*+). Middle: Virtual ISH images of individual genes. Bottom: ISH images of individual genes. **d**, Re-embedded 2D UMAP of 206,908 cells from renal cortical & medullary cells, anterior intermediate mesoderm, posterior intermediate mesoderm, metanephric mesenchyme, and splanchnic mesoderm. Cells are colored by either their initial annotations (left) or timepoint (right, after downsampling to a uniform number of cells per time window). **e**, The average normalized expression of *Foxd1* over time is shown for renal cortical stromal cells (left) and renal medullary stromal cells (right). Gene expression was normalized by the size factor estimated by Monocle/3. **f**, Top: The same UMAP as in panel a, colored by gene expression of marker genes which appear specific to two subsets of renal stromal cells: medullary renal stromal cells (*Zeb2*+, *Plcb1*+) and cortical renal stromal cells (*Ntn1*+, *Zbtb7c*+, *Sema3d*+), respectively. Middle: Virtual ISH images of individual genes. Bottom: ISH images of individual genes. In panel b, c, and f, virtual ISH images of individual genes were obtained from one selected section (E1S1) from E14.5 of the *Mosta* data (https://db.cngb.org/stomics/mosta/). ISH images were obtained from the Jackson Laboratory Mouse Genome Informatics (MGI) website (https://www.informatics.jax.org/). The original reference for these ISH images are[81,82,84].

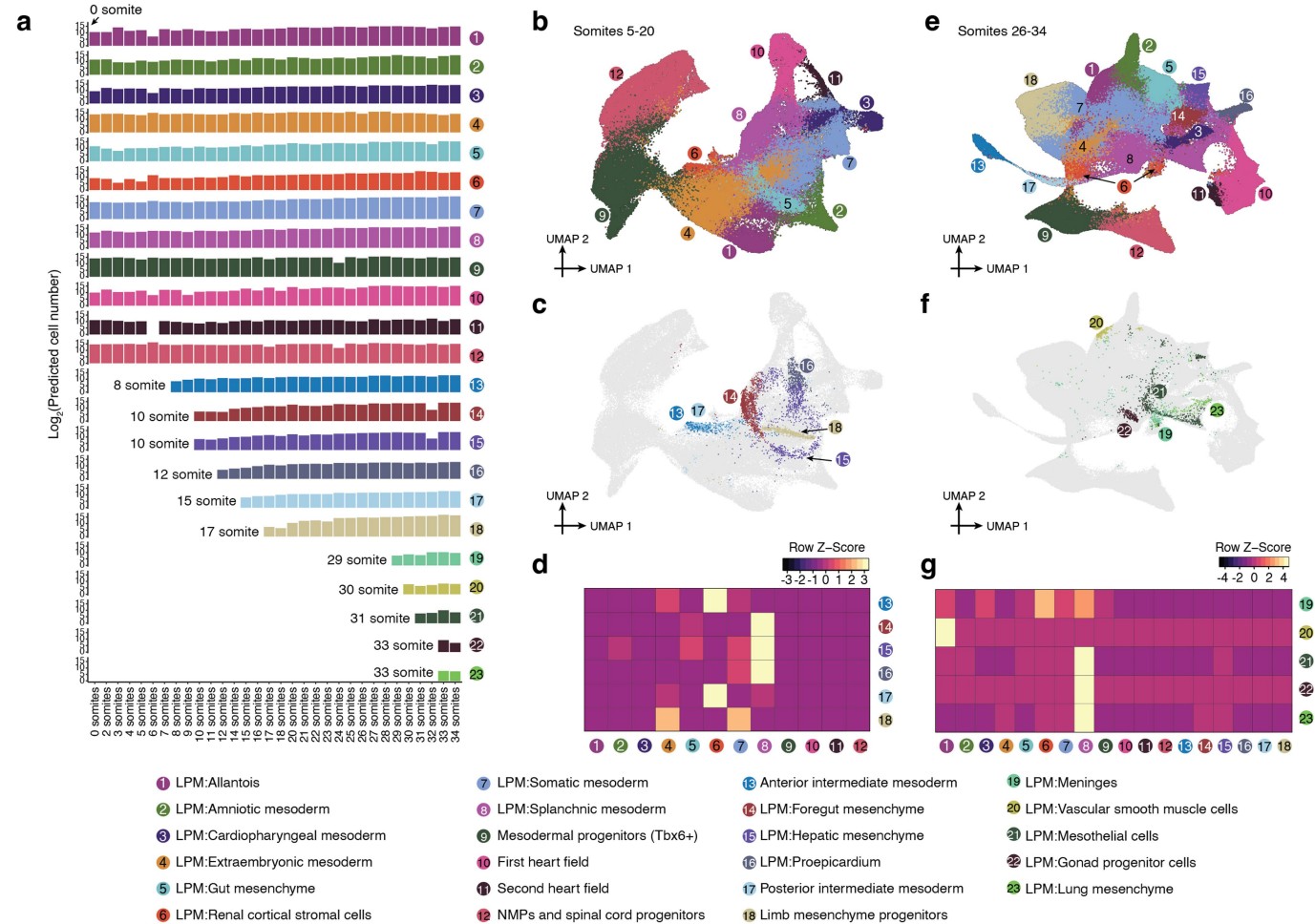

**Extended Data Fig. 8 | The emergence of mesenchymal subtypes from the patterned mesoderm. a**, The predicted absolute number (log2 scale) of cells of each mesoderm cell type at each somite count. The predicted absolute number was calculated by the product of its sampling fraction in the overall embryo and the predicted total number of cells in the whole embryo at the corresponding timepoint. Because cell numbers were only predicted for the broader bins (Fig. 1b), rather than individual somite counts, these were used for roughly corresponding sets (0-12 somite stage: E8.5; 14-15 somite stage: E8.75; 16-18 somite stage: E9.0; 20-23 somite stage: E9.25; 24-26 somite stage: E9.5; 27-31 somite stage: E9.75; 32-34 somite stage: E10.0). For each row, the first somite count with at least 10 cells assigned that cell type annotation is labeled, and all observations prior to that somite count are discarded. **b**, Re-embedded 2D UMAP of 110,753 cells from the selected cell types of mesoderm (clusters 1-12 as listed in panel a) from 5-20 somite stage embryos. **c**, The same UMAP as in panel b, but with inferred progenitor cells colored by derivative cell type with the highest mutual nearest neighbors (MNN) pairing score. **d**, Normalized MNN

pairing score between mesodermal territories (column) and their inferred derivative cell types (row) from 5-20 somite stage embryos. The selected cell populations are first embedded into 30 dimensional PCA space, and then for individual derivative cell types, MNN pairs (k = 10 used for k-NN) between their earliest 500 cells (in absolute time) and cells from mesodermal territories are identified. **e**, Re-embedded 2D UMAP of 275,000 cells from the selected cell types of mesoderm (clusters 1-18 as listed in panel a) from 26-34 somite stage embryos. **f**, The same UMAP as in panel e, but with inferred progenitor cells colored by derivative cell type with the highest MNN pairing score. **g**, Normalized MNN pairing score between mesodermal territories (column) and their inferred derivative cell types (row) from 26-34 somite stage embryos. The selected cell populations are first embedded into 30 dimensional PCA space, and then for individual derivative cell types, MNN pairs (k = 10 used for k-NN) between their earliest 500 cells (in absolute time) and cells from mesodermal territories are identified.

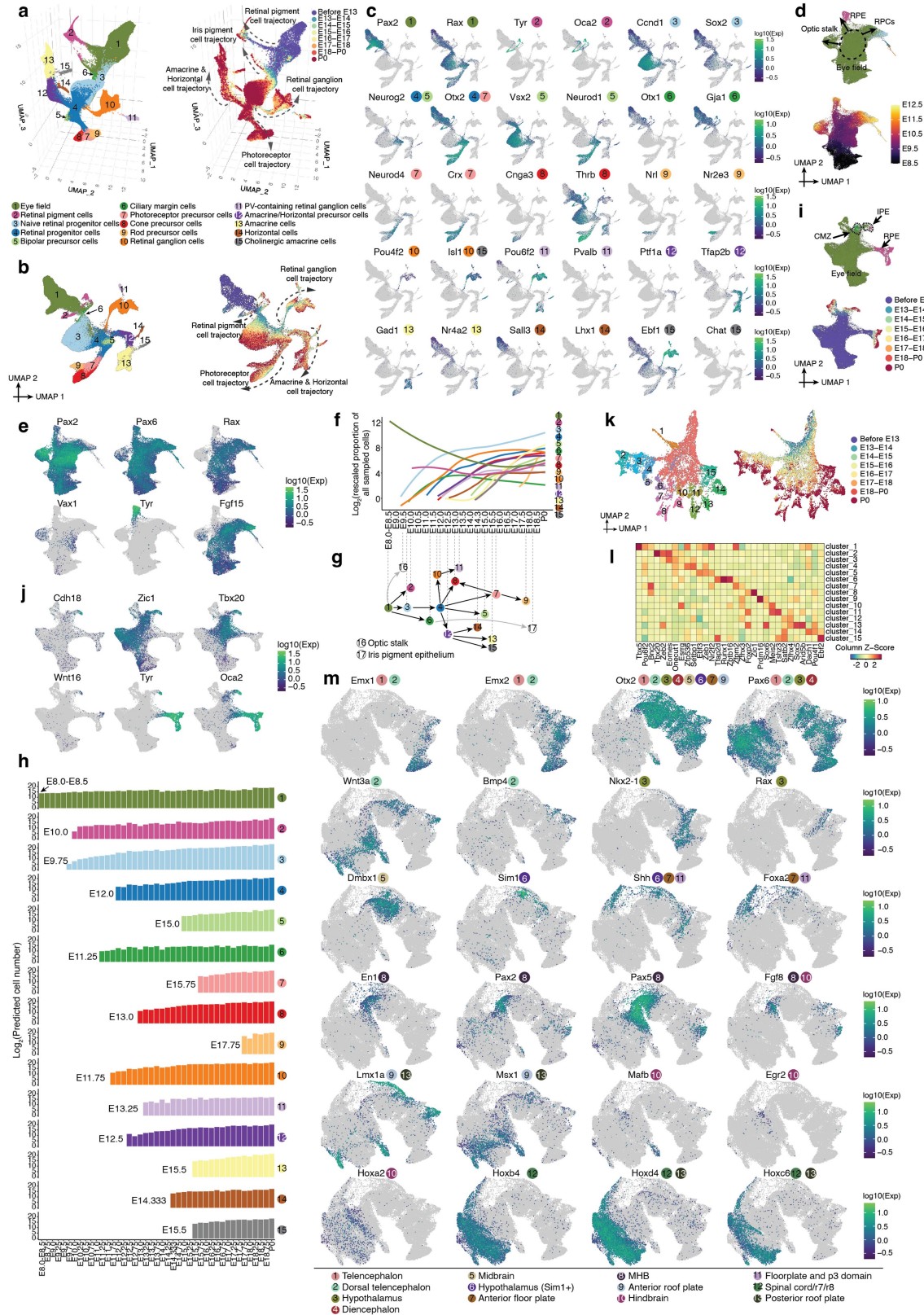

**Extended Data Fig. 9** | See next page for caption.

**Extended Data Fig. 9 | The timing and trajectories of retinal development, and marker gene expression for different neuroectodermal territories.**
**a**, Re-embedded 3D UMAP of 160,834 cells corresponding to the retinal development from E8 to P0. Cells are colored by either their initial annotations (left) or timepoint (right, after downsampling to a uniform number of cells per time window). Arrows highlight five of the main trajectories observed.
**b**, Re-embedded 2D UMAP of 160,834 cells corresponding to the retinal development from E8 to P0. The same UMAP as in panel a, except 2D instead of 3D projection. **c**, The same UMAP as in panel b, colored by gene expression of marker genes for each annotated retinal cell type. References for marker genes are provided in Supplementary Table 5. **d**, Re-embedded 2D UMAP of the subset of cells in panel a from stages <= E12.5. Cells are colored by either their initial annotations (top) or timepoint (bottom). **e**, The same UMAP as in panel d, colored by gene expression of markers of retinal progenitor cells RPCs (*Pax2* +, *Pax6* +, *Rax* +, *Fgf15* +), RPE (*Tyr* +), and the optic stalk (*Pax2* +, *Vax1* +, *Rax*−). References for marker genes are provided in Supplementary Table 12. **f**, Rescaled proportion of profiled cells (log2; y-axis) for each cell type shown in panel a, as a function of developmental time (x-axis). For rescaling, the % of profiled cells in the entire embryo assigned a given annotation was multiplied by 100,000, prior to taking the log2. Line plotted with *geom_smooth* function in ggplot2.
**g**, Schematic of retinal cell types emphasizing the timing at which they first appear and their inferred developmental relationships from E8-P0, based on manual review of the trajectories. The gray lines indicate subsets of the eye field and RPE subsequently annotated as the optic stalk (label 16) and iris pigment epithelium (label 17), respectively. Cell types are positioned along the x-axis at the timepoint at which they are first observed (as shown in panel h).
**h**, The predicted absolute number (log2 scale) of cells of each retinal cell type at each timepoint. The predicted absolute number was calculated by the product of its sampling fraction in the overall embryo and the predicted total number of cells in the whole embryo at the corresponding timepoint (Fig. 1b). For each row, the first timepoint with at least 10 cells assigned that cell type annotation is labeled, and all observations prior to that timepoint are discarded. **i**, Re-embedded 2D UMAP of a subset of cells in panel a corresponding to eye field, RPE and CMZ. Cells are colored by either their initial annotations (top) or timepoint (bottom). **j**, The same UMAP as in panel i, colored by gene expression of marker genes for IPE or pigment epithelium more generally (*Tyr* & *Oca2*). RPE: retinal pigment epithelium. CMZ: ciliary marginal zone. RPCs: retinal progenitor cells. IPE: iris pigment epithelium. References for marker genes are provided in Supplementary Table 12. **k**, Re-embedded 2D UMAP of retinal ganglion cells. Cells are colored by either clusters (left; Leiden clustering followed by downselection to late-appearing clusters) or timepoint (right). **l**, The top 3 TF markers of the 15 clusters shown in panel k. Marker TFs were identified using the *FindAllMarkers* function of Seurat/v3[63]. Their mean gene expression values in each cluster are represented in the heatmap, calculated from original UMI counts normalized to total UMIs per cell, followed by natural-log transformation. The full list of significant TFs is provided in Supplementary Table 14. **m**, Marker gene expression for different neuroectodermal territories. The same UMAP as in Fig. 4a, colored by gene expression of marker genes for each neuroectodermal territory. References for marker genes are provided in Supplementary Table 5.

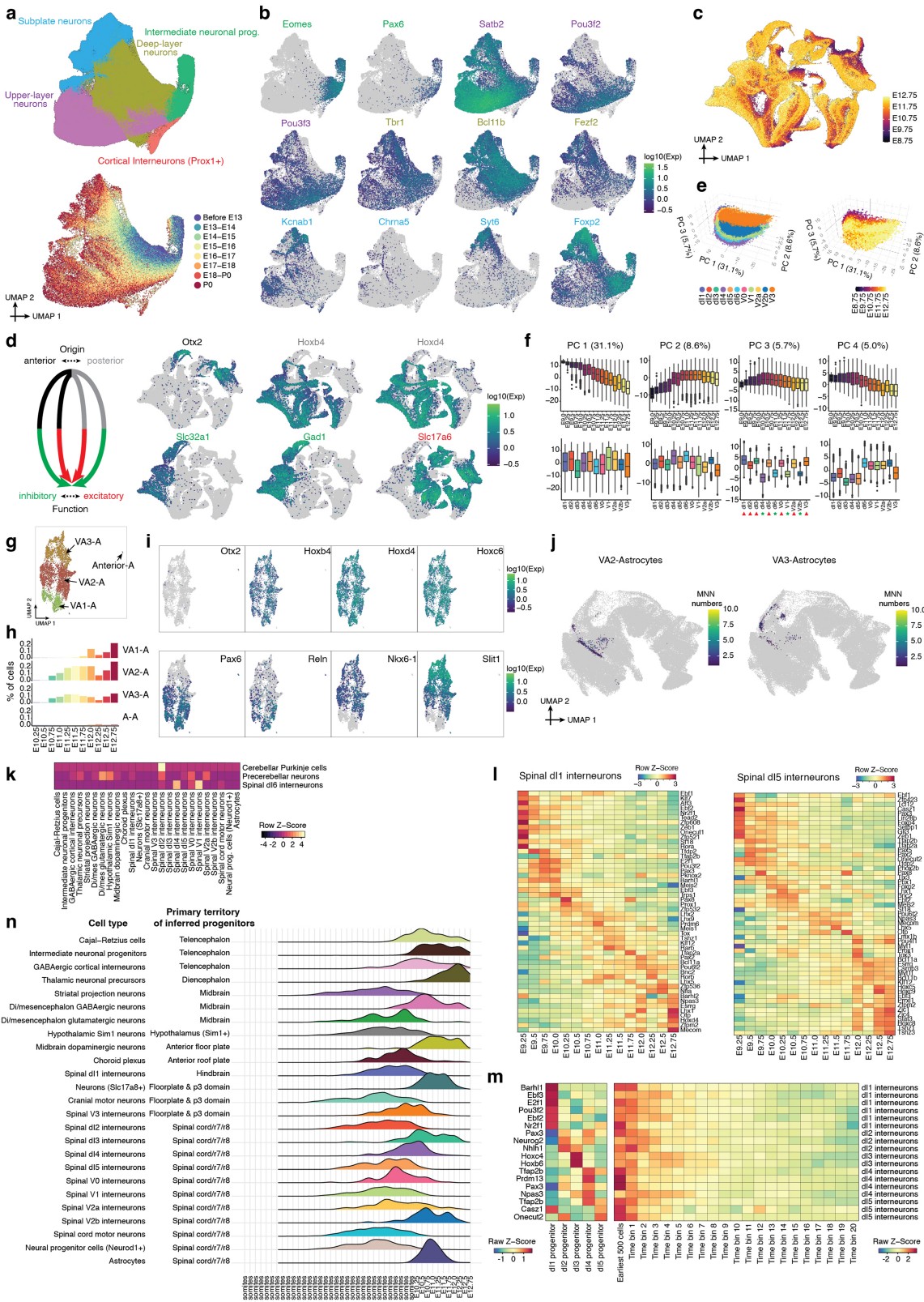

**Extended Data Fig. 10** | See next page for caption.

**Extended Data Fig. 10 | Subtypes of intermediate neuronal progenitors, glutamatergic & GABAergic neurons, and early astrocytes, and the timing of neuronal subtype differentiation from the patterned neuroectoderm.**
**a**, Re-embedded 2D UMAP of 628,251 cells within the intermediate neuronal progenitors major cell cluster, colored by either cell type (top) or developmental stage (bottom, after downsampling to a uniform number of cells per time window). **b**, The same UMAP as in panel a, colored by gene expression of marker genes which appear specific to intermediate neuronal progenitors (*Eomes* +, *Pax6* +), upper-layer neurons (*Satb2* +, *Pou3f2* +, *Pou3f3* +), deep-layer neurons (*Tbr1* +, *Bcl11b* +, *Fezf2* +), or subplate neurons (*Kcnab1* +, *Chrna5* +, *Syt6* +, *Foxp2* +). References for marker genes are provided in Supplementary Table 5. **c**, The same UMAP as in Fig. 4e, with cells colored by timepoints. **d**, Left: Neuronal subtypes shown in Fig. 4e originate from anterior vs. posterior of neuroectoderm, and then subsequently display inhibitory vs. excitatory functions after differentiation. Right: The same UMAP as in Fig. 4e, colored by gene expression of marker genes which appear specific to anterior (*Otx2* +) vs. posterior (*Hoxb4* +, *Hoxd4* +) origins, or inhibitory (*Slc32a1* +, *Gad1* +) vs. excitatory (*Slc17a6* +) functions. References for marker genes are provided in Supplementary Table 12. **e**, 3D visualization of gene expression variation in 11 spinal interneurons, colored by cell type (left) or timepoint (right). **f**, Correlations between top four PCs and timepoints (top row) or cell types (bottom row). Boxplots (*n* = 97,842 cells) represent IQR (25th, 50th, 75th percentile) with whiskers representing 1.5× IQR. Red triangles and green stars highlight glutamatergic and GABAergic spinal cord interneurons, respectively. **g**, Subtypes of early astrocytes and their inferred progenitors. Re-embedded 2D UMAP of 5,928 cells within the astrocytes from stages <E13. **h**, Composition of embryos from each 6-hr bin by different subpopulations of astrocytes. **i**, The same UMAP as in panel g, colored by gene expression of marker genes which

appear specific to anterior (*Otx2* +) or posterior (*Hoxb4* +, *Hoxd4* +, *Hoxc6* +) astrocytes, VA1-astrocytes (*Pax6* +, *Reln* +), VA2-astrocytes (*Pax6* +, *Reln* +, *Nkx6-1* +, *Slit1* +), and VA3-astrocytes (*Nkx6-1* +, *Slit1* +). References for marker genes are provided in Supplementary Table 12. **j**, The same UMAP of the patterned neuroectoderm as in Fig. 4a, with inferred progenitor cells of astrocytes colored by the frequency that has been identified as a MNN with either VA2-astrocytes (left) or VA3-astrocytes (right). **k**, For those three cell types (cerebellar Purkinje cells, precerebellar neurons, spinal dI6 interneurons) which were excluded from the analyses represented in Fig. 4g, h due to having fewer than 50 MNN pairs, we performed a recursive mapping to identify whether they might share progenitors with another derived cell type, essentially repeating the analysis but attempting to map the earliest cells of these cell types to other derivative cell types rather than the patterned neuroectoderm. The heatmap shows the number of MNN pairs between pairwise cell types. In brief, this analysis suggests that spinal dI2 interneurons and cerebellar astrocytes share progenitors, while the progenitors of the other two re-analyzed cell types remain ambiguous. **l**, Gene expression across timepoints, for the specific TF markers of spinal dI1 (left) or spinal dI5 (right) interneurons. **m**, Left: gene expression for 18 selected TFs, across progenitor cells of dI1-5 from the neuroectodermal territories. Right: gene expression for 18 selected TFs across 21 time bins for dI1-5 spinal interneurons in which the TF has been nominated as marker TF. For individual spinal interneurons (each row), the first time bin involves the earliest 500 cells, then the left cells break into 20 bins ordered by their timepoints and with the same number of cells in each bin. Only cells from stages <E13 are included. **n**, For each neuronal subtype in Fig. 4g, h, we selected the annotation in the patterned neuroectoderm to which the most inferred progenitors had been assigned, and plotted the distribution of timepoints for that subset of inferred progenitors.

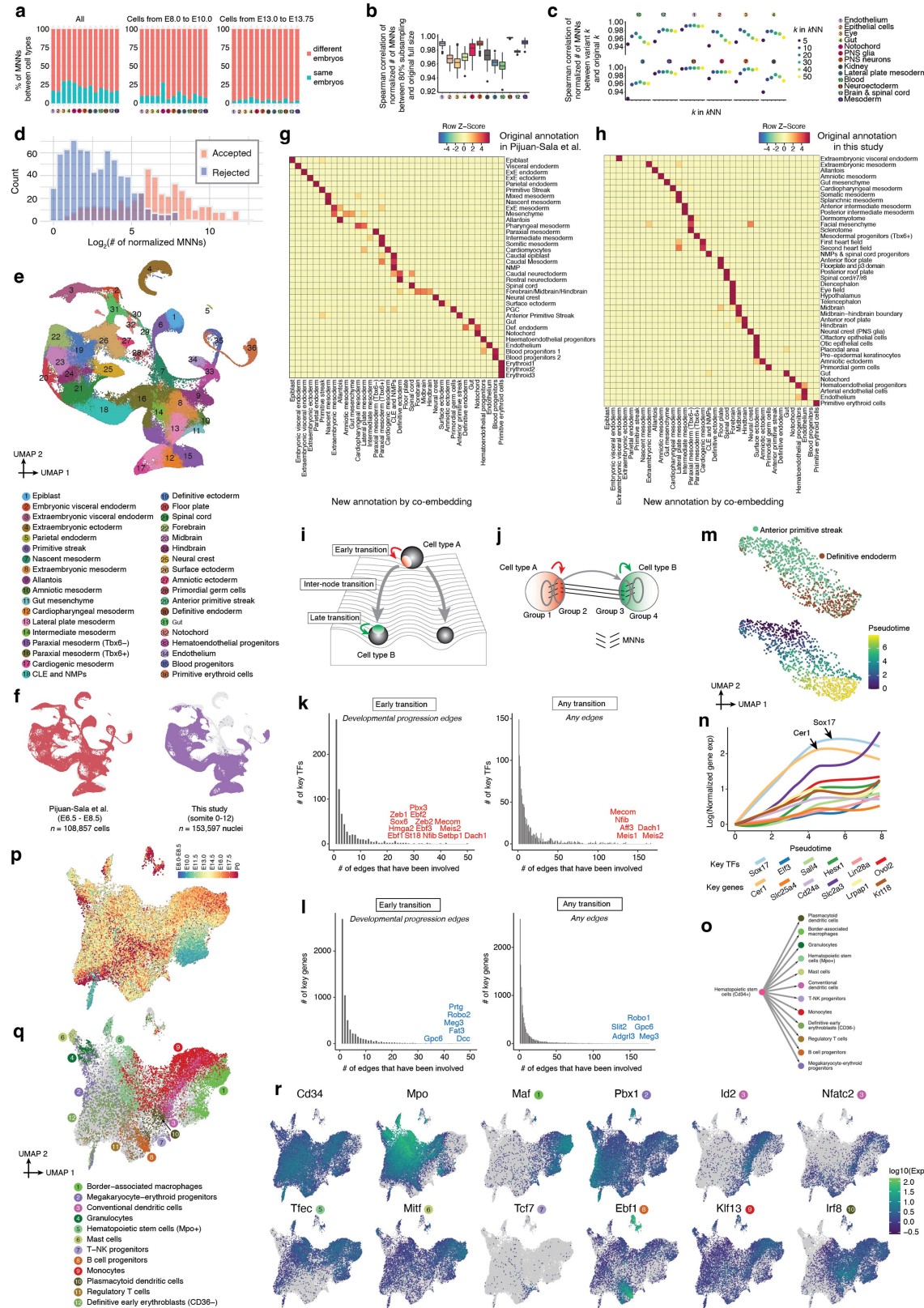

**Extended Data Fig. 11** | See next page for caption.

**Extended Data Fig. 11 | Identifying equivalent cell type nodes across datasets, and systematically nominating TFs and other genes for cell type specification. a**, The MNN approach used for graph construction is robust to subsampling and choice of the k parameter. The percentage of MNNs between different cell types, from the same embryo (blue) or from different embryos (red), is shown for each developmental system during organogenesis & fetal development, for all cells (left), cells from E8.0 to E10.0 (middle), or cells from E13.0 to E13.75 (right). **b**, The Spearman correlation coefficients of the normalized number of MNNs between cell types, comparing random subsampling of 80% of the cells to the full set of cells. The subsampling was repeated 100 times. The number of MNNs between cell types were normalized by the total number of possible MNNs between them. Boxplot ($n = 1,200$ correlation coefficients) represents IQR (25th, 50th, 75th percentile) with whiskers representing $1.5\times$ IQR. Outliers are shown as the dots outside the whiskers. **c**, The Spearman correlation coefficients of the normalized number of MNNs between cell types, comparing various choices for k parameter (k = 5, 10, 20, 30, 40, 50) and the choice of k parameter (k = 15) when applying kNN to the developmental systems during organogenesis & fetal development. The number of MNNs between cell types were normalized by the total number of possible MNNs between them. Colors and numbers in panels a-c correspond to each developmental system annotations listed at the top right. **d**, 1,155 edges with the number of normalized MNNs > 1 were manually reviewed for biological plausibility. Histogram of edges that were accepted or rejected as a function of normalized MNN score. **e**, Integration of scRNA-seq profiles from gastrulation and early somitogenesis to identify equivalent cell type nodes across datasets generated by distinct technologies. 2D UMAP visualization of co-embedded cells, derived both from a gastrulation dataset based on cells from E6.5 to E8.5 generated on the 10x Genomics platform[7] ($n = 108,857$ cells) and the earliest ~1% of this dataset (0-12 somite stage embryos) generated by sci-RNA-seq3 ($n = 153,597$ nuclei), after batch correction[63]. This is essentially an updated version of an analysis that we have done previously[8]. We performed clustering and cell type annotation on the integrated co-embedding, as shown. **f**, The same UMAP as in panel e is shown twice, with colors highlighting cells/nuclei from Pijuan-Sala's dataset[7] (left) or early somitogenesis[8] (right). **g**, For cells from the original Pijuan-Sala's dataset[7], we quantify and display the overlap between the original annotations and the new annotations shown in panel e. For each row, the proportions of cells that are distributed across each column are transformed to z-score. **h**, For nuclei from the early somitogenesis embryos[8], we quantify and display the overlap between the original annotations and the new annotations shown in panel e. These mappings were the basis for dataset equivalence edges between the "gastrulation" and 12 "organogenesis & fetal development" subsystems. For each row, the proportions of cells that are distributed across each column are transformed to z-score. CLE: Caudal lateral epiblast. NMPs: Neuromesodermal progenitors. **i**, A Waddington landscape cartoon illustrating how a cell type transition might be broken into three phases. **j**, Given a directional edge between two nodes, A → B, we identified the subset of cells within each node that were either MNNs of the other cell type (inter-node; groups 2 & 3) or MNNs of those cells (intra-node; groups 1 & 4). If A → B, this effectively models the transition as group 1 → 2 → 3 → 4. **k**, Histograms of the number of edges in which TFs are differentially expressed. The left histogram counts only genes when they are differentially expressed across the early phase of an developmental progression edge, while the right histogram counts genes when they are differentially expressed in any phase of all edges. **l**, Same as panel k, but for all genes rather than only TFs. **m**, Re-embedded 2D UMAP of 988 cells participating in groups 1-4 of the transition from anterior primitive streak → definitive endoderm. Cells are colored by either cell type annotations (top) or estimated pseudotime (bottom) using *Monocle*3[14]. **n**, For cells in panel m, normalized gene expression of selected genes are plotted as a function of estimated pseudotime. Gene expression values were calculated from original UMI counts normalized to total UMIs per cell, followed by natural-log transformation. The line of gene expression was plotted by the *geom_smooth* function in ggplot2. We manually added an offset based on their expression at pseudotime = 0 to the y-axis for individual genes. **o**, A sub-graph of Fig. 5g, including hematopoietic stem cells (Cd34 + ) and 12 cell type nodes which appear derived from it. **p**, Re-embedded 2D UMAP of 37,750 cells from hematopoietic stem cells (Cd34 + ), colored by developmental stage (after downsampling to a uniform number of cells per stage). **q**, The same UMAP as in panel p, but with inferred progenitor cells (the cells participating in the MNNs that support the edges) colored by derivative cell type with the most frequent MNN pairs. **r**, The same UMAP as in panel p, colored by gene expression of selected top key TFs which were upregulated during the "early transition" for each derivative.

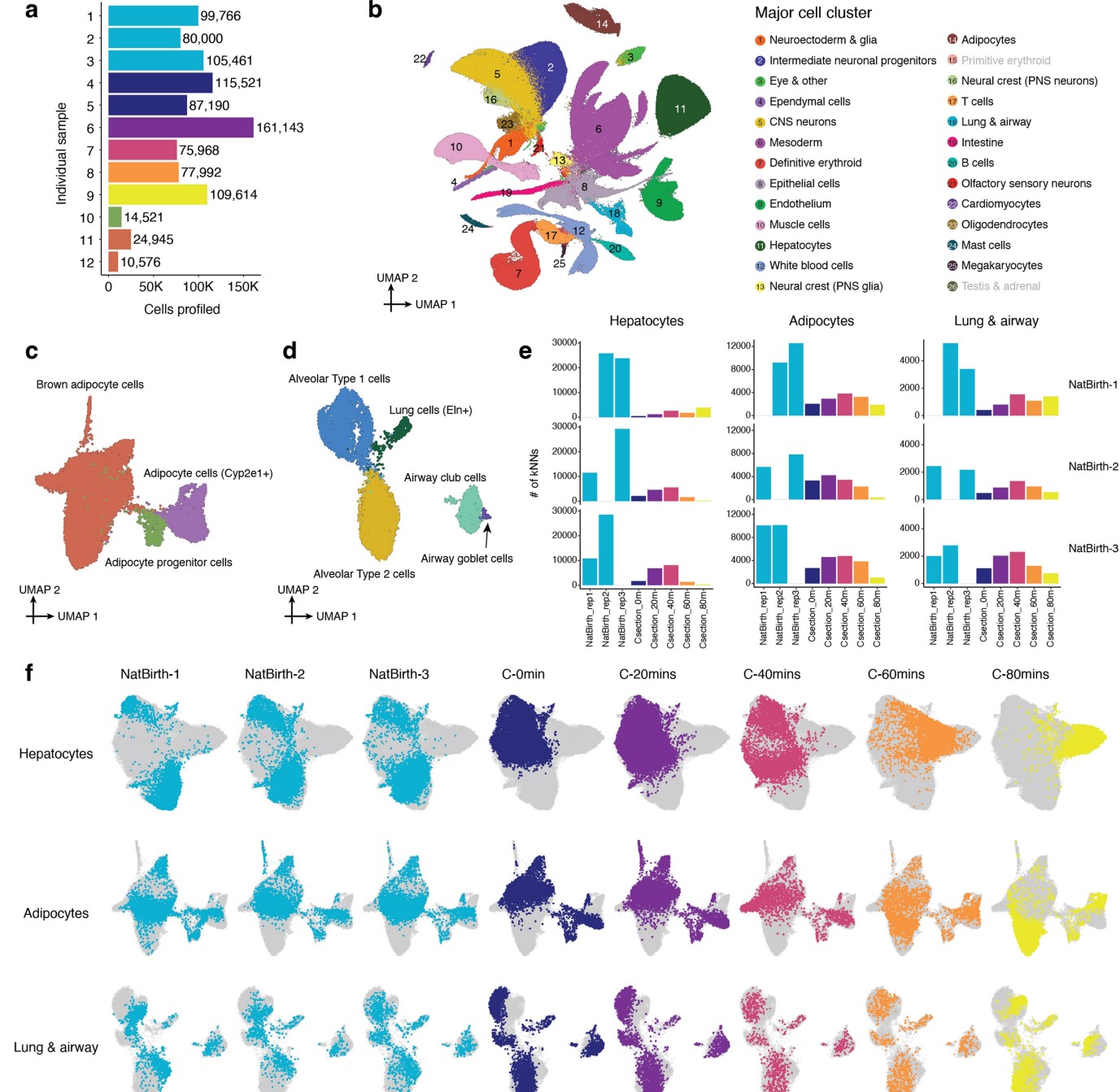

**Extended Data Fig. 12 | Rapid shifts in transcriptional state occur in a restricted subset of cell types upon birth, and differ between vaginally and C-section delivered pups. a**, The number of nuclei profiled for each animal shown in Fig. 6c. A small number of nuclei from additional fetal samples from the original set of experiments were also profiled for quality control (samples 10-12). **b**, 2D UMAP visualization of the birth-series dataset (*n* = 962,697 cells). Colors correspond to 26 major cell cluster annotations (Fig. 1b, c). Two major cell clusters (the primitive erythroid and testis & adrenal major cell clusters) shown in the original dataset but missed here are highlighted in gray. Primitive erythroid cells are not present at these timepoints and testis & adrenal cells are collapsed to the epithelial cells major cell cluster due to their low numbers. **c**, Re-embedded 2D UMAP of 19,696 cells of the adipocyte major cell cluster.

**d**, Re-embedded 2D UMAP of 7,986 cells of the lung & airway major cell cluster. **e**, For these three major cell clusters, we co-embedded cells from three vaginally delivered pups (samples 1-3 in Fig. 6c) and six pups delivered by C-section (samples 4-9 in Fig. 6c), followed by subsetting a uniform number of cells per sample. For cells from each of the three vaginally delivered pups, we calculated the number of their 10 nearest neighbors in the PCA embedding (*n* = 30 dimensions) from other samples. **f**, Re-embedded 2D UMAP of cells from these three major cell clusters, based on cells from three vaginally delivered pups and six pups delivered by C-section. For each row, the same UMAP is shown multiple times, with colors highlighting cells from individual pups (or two pups, in the case of the 0-min C-section timepoint).

# nature research

| | |
|---|---|

# Reporting Summary

Nature Research wishes to improve the reproducibility of the work that we publish. This form provides structure for consistency and transparency in reporting. For further information on Nature Research policies, see our Editorial Policies and the Editorial Policy Checklist.

## Statistics

For all statistical analyses, confirm that the following items are present in the figure legend, table legend, main text, or Methods section.

| n/a | Confirmed | |
|---|---|---|
| ☐ | ☒ | The exact sample size ($n$) for each experimental group/condition, given as a discrete number and unit of measurement |
| ☐ | ☒ | A statement on whether measurements were taken from distinct samples or whether the same sample was measured repeatedly |
| ☐ | ☒ | The statistical test(s) used AND whether they are one- or two-sided<br>*Only common tests should be described solely by name; describe more complex techniques in the Methods section.* |
| ☐ | ☒ | A description of all covariates tested |
| ☐ | ☒ | A description of any assumptions or corrections, such as tests of normality and adjustment for multiple comparisons |
| ☐ | ☒ | A full description of the statistical parameters including central tendency (e.g. means) or other basic estimates (e.g. regression coefficient) AND variation (e.g. standard deviation) or associated estimates of uncertainty (e.g. confidence intervals) |
| ☐ | ☒ | For null hypothesis testing, the test statistic (e.g. $F$, $t$, $r$) with confidence intervals, effect sizes, degrees of freedom and $P$ value noted<br>*Give P values as exact values whenever suitable.* |
| ☒ | ☐ | For Bayesian analysis, information on the choice of priors and Markov chain Monte Carlo settings |
| ☒ | ☐ | For hierarchical and complex designs, identification of the appropriate level for tests and full reporting of outcomes |
| ☐ | ☒ | Estimates of effect sizes (e.g. Cohen's $d$, Pearson's $r$), indicating how they were calculated |

*Our web collection on statistics for biologists contains articles on many of the points above.*

## Software and code

Policy information about availability of computer code

| | |
|---|---|
| Data collection | No software was used except Illumina RTA basecalling at this stage. |
| Data analysis | The Python (version 3.10.10) and R (version 3.6.3 and 4.2.3) codes used to analyze the RNA-seq data are available at https://github.com/ChengxiangQiu/JAX_code. The following common, freely available data analysis software were used to analyze data: bcl2fastq version 2.20 (https://support.illumina.com), deML version 1.1.3 (https://github.com/grenaud/deML), HTseq version 0.6.1 (https://github.com/htseq/htseq), trim_galore version 0.6.5 (https://github.com/FelixKrueger/TrimGalore), STAR version 2.6.1d (https://github.com/alexdobin/STAR), scrublet version 0.1 (https://github.com/swolock/scrublet), Scanpy version 1.6.0 (https://github.com/theislab/scanpy), Monocle version 3, and 3-alpha (https://cole-trapnell-lab.github.io/monocle3), Seurat version 3 (https://github.com/satijalab/seurat), Tangram version 1.0.3 (https://github.com/broadinstitute/Tangram), Cytoscape version 3.9.1 (https://cytoscape.org/), geosketch version 1.2 (https://github.com/brianhie/geosketch). |

For manuscripts utilizing custom algorithms or software that are central to the research but not yet described in published literature, software must be made available to editors and reviewers. We strongly encourage code deposition in a community repository (e.g. GitHub). See the Nature Research guidelines for submitting code & software for further information.

## Data

Policy information about availability of data

All manuscripts must include a data availability statement. This statement should provide the following information, where applicable:
- Accession codes, unique identifiers, or web links for publicly available datasets
- A list of figures that have associated raw data
- A description of any restrictions on data availability

The data generated in this study can be downloaded in raw and processed forms from the NCBI Gene Expression Omnibus under accession number GSE186069 and

April 2020

# Field-specific reporting

Please select the one below that is the best fit for your research. If you are not sure, read the appropriate sections before making your selection.

☒ Life sciences  ☐ Behavioural & social sciences  ☐ Ecological, evolutionary & environmental sciences

For a reference copy of the document with all sections, see nature.com/documents/nr-reporting-summary-flat.pdf

# Life sciences study design

All studies must disclose on these points even when the disclosure is negative.

| Sample size | No statistical methods were used to predetermine sample size.<br>Our previous study (Qiu et al., 2022), which profiled 154,313 cells from 12 mouse embryos at early somitogenesis stage, successfully identified the same 30 cell types as those identified in E8.5 data by Pijuan-Sala et al. (2019). The extensive data, along with the separate processing of individual somite-resolved embryos, enabled the detection of significant substructures, such as A-P floor plates and various hindbrain segmentations. In this study, we applied the same technology to profile single nuclei from mouse embryos, identifying over 200 distinct cell types and focusing on several specific tissues and organs. This comprehensive analysis suggests that our sample size is adequate for investigating cell states and developmental trajectories during mouse organogenesis. In addition, we experimentally quantified the total DNA of staged embryos and estimated that the embryo grows 3,000-fold from E8.5 to P0. Therefore, despite the large number of nuclei profiled, our cellular coverage remains limited, ranging from 0.5-fold for early stages (summing 6 embryos, somite counts 7-12) to 0.002-fold immediately before birth (summing 6 embryos, E17.5-E18.75). |
|---|---|
| Data exclusions | When we took a first round of cell-embedding, we noticed that one mouse embryo at E14.5 had a grossly reduced proportion of neuronal cells. This particular sample had been divided during pulverization, and we suspect that large portions of the frozen embryo did not make it into the experiment. We removed cells from this E14.5 embryo. |
| Replication | Firstly, we performed 15 sci-RNA-seq3 experiments, and the data from each experiment overlapped well, demonstrating high replicability. We have employed various methods to confirm the data quality. Secondly, to validate our findings regarding posterior embryos, we generated an independent validation dataset comprising somites 8-21, and the findings were validated. Thirdly, to validate our observations of abrupt transcriptional changes before and after birth, we generated a new "birth-series" dataset, and the findings were validated. Finally, for the spatial mapping analysis, we utilized publicly available ISH images to verify our cell-type annotations within the lateral plate mesoderm. |
| Randomization | From a total of 523 embryos staged at the Jackson Laboratory, we selected 75 for whole embryo scRNA-seq, targeting one embryo for every somite count from 0 to 34 (2-hr increments), and one embryo for every 6-hr bin from E10 to P0. Embryos used in experiments were randomly selected from each timepoint before sample preparation. |
| Blinding | In this study, investigators were blinded to group allocation during sample collection and data generation/analysis: embryo collection and sci-RNA-seq3 data generation/analysis were performed by different researchers in different locations. |

# Reporting for specific materials, systems and methods

We require information from authors about some types of materials, experimental systems and methods used in many studies. Here, indicate whether each material, system or method listed is relevant to your study. If you are not sure if a list item applies to your research, read the appropriate section before selecting a response.

## Materials & experimental systems

| n/a | Involved in the study |
|---|---|
| ☒ | Antibodies |
| ☒ | Eukaryotic cell lines |
| ☒ | Palaeontology and archaeology |
| ☐ | ☒ Animals and other organisms |
| ☒ | Human research participants |
| ☒ | Clinical data |
| ☒ | Dual use research of concern |

## Methods

| n/a | Involved in the study |
|---|---|
| ☒ | ChIP-seq |
| ☒ | Flow cytometry |
| ☒ | MRI-based neuroimaging |

# Animals and other organisms

Policy information about studies involving animals; ARRIVE guidelines recommended for reporting animal research

| | |
|---|---|
| Laboratory animals | 83 precisely staged C57BL/6NJ (strain# 005304) mice were obtained at The Jackson Laboratory. Mice of both sexes were included in the study, with a roughly equal number of males and females. |
| Wild animals | Study did not involve wild animals |
| Field-collected samples | Study did not involve field-collected samples |
| Ethics oversight | All animal use at The Jackson Laboratory was done in accordance with the Animal Welfare Act and the AVMA Guidelines on Euthanasia, in compliance with the ILAR Guide for Care and Use of Laboratory Animals, and with prior approval from The Jackson Laboratory animal care and use committee (ACUC) under protocol AUS20028. |

Note that full information on the approval of the study protocol must also be provided in the manuscript.

