## [Peer Review File · Nature]

Manuscript Title: A single cell timelapse of mouse prenatal development, from gastrula to birth

Reviewer Comments & Author Rebuttals

Reviewer Reports on the Initial Version:

Referees' comments:

Referee #1 (Remarks to the Author):

The authors have generated the most comprehensive set of single nuclear data on the developing mouse to date and subjected these data to a number of insightful analyzes. I particularly enjoyed the clarity of the writing and the authors pointing out limits to the data or conclusions throughout. The care in the attention to staging collections, the significant efforts to reduce batch variability in the indexing approach and the decent depth of gene reads/nucleus will make this a lasting resource for the community. I expect that "domain" experts will rapidly enhance the analyzes here and I have erred away from nit-picking in favor of encouraging a timely publication. My only significant comment relates to community accessibility and a pet-peeve that not enough is done beyond making data accessible to enable viewing of the data. Given the computational expertise and resources of the authors, it is a reasonable expectation for the authors to enable accessible data viewing portals that would greatly facilitate rapid querying these powerful datasets. Shiny apps have been a simple way several groups have democratized data to those without "r" expertise.

A few specific minor comments readily addressable:

- 1) The authors note their analysis suggest Brachyury is within posterior gut cells. Indeed, Schifferer et al (2021) Development Figure 1C" shows this nicely.
- 2) The statement here is a little confusing: "specification of these posterior and anterior trajectories in late gastrulation is initiated by interactions between Gdnf and Ret". Gdnf and Ret demarcate different populations but there is no evidence this signaling axis specifies cell types?
- 3) The authors write "Of note, we observe "convergence" of the posterior and anterior trajectories in collecting duct intercalated cells (cluster 4 in Fig. 3a-b). More detailed investigation suggests that the posterior intermediate mesoderm may also contribute to the collecting duct, although lineage analysis would be necessary to confirm this (Fig. 3d-e; Supplementary Fig. 7f)". Ransick et al., (2019) Dev Cell demonstrated with lineage tracing the dual origin of intercalated cell types from the distal nephron and ureteric lineages – data visible at Kidney Cell Explorer (<https://cello.shinyapps.io/kidneycellexplorer/>).
- 4) The authors write "We can also distinguish two subsets of LPM-derivatives mapping to the kidney, one to the inside and the other to the surface, which may correspond to renal stroma and the renal pericytes and mesangial cells, respectively (Fig. 3g)". It is likely that the Foxd1 population (cluster 4 labelled renal stroma) contains much of the stromal cell diversity. Fate mapping studies (Kobayashi et al 2014 Stem Cell Reports) indicate that that Foxd1+ cells give rise to the renal pericytes and mesangial cells. May be good to clarify that the observation should not be interpreted as two different origins. Parenthetically, I am surprised that cluster 14 separates so clearly from cluster 4. Perhaps there is an alternative ID for these cells?
- 5) The authors write "The apparent bifurcation of the proximal tubule corresponds to major

differences in the transcriptional state of cells from embryos obtained before birth (E18.75 or earlier) vs. after birth (P0) (cluster 9 in Fig. 3a-b; Supplementary Fig. 7d). We return to this observation in the final section of the manuscript.” However, there is not much more presented in the later discussion. If the authors are to single out proximal tubule cells, I suggest adding the differential gene expression data to the relevant supplementary tables so the data can be viewed.

Referee #2 (Remarks to the Author):

The paper by Qiu, Martin, Welch et al is presenting a remarkable new dataset including 11.4M QC positive single cell profiles acquired from 74 mouse embryo sampled carefully over the entire developmental time axis starting at E8 and post birth. This is a dream dataset, representing the culmination of a remarkable effort of technology development from the Shendure lab and their collaborators. This data is collected, controlled and processed in a standardized and well documented fashion, and is now providing an unprecedentedly deep and precise basis for understanding mammalian embryonic development.

The procedures for embryo staging and selection, single cell profiling and QC are at a very high standard. The authors are doing an excellent job in describing how the new resource was constructed. In that respect, the expert staging and selection, and then the quantification of overall embryo size and growth rate is very important. It is understandable that the bulk of the results presented in the text are highlighting specific lineages and stages, rather than aiming at a hopeless attempt for comprehensive detailed description of the process. The authors are also refraining from unneeded overly theoretical statements or arguments on principles, and I commend them on avoiding this and focusing on the data and how to use it. I would advise against requests for more validation experiments on the data discussed in Figure 2-7 – and while I have some practical analysis/bias concerns below, it is clear that all can be addressed with textual changes, simple additional analysis, or, in the worst case, omission of some panels for which evidence is not conclusive. This resource can change the way by which we approach embryonic development of specific lineages and tissues given a much-needed global context. This is possible even given the present form of the data, and will be intensified after much more analysis, tools, integration with spatial data and epigenomics profiles will be added on top of it by the authors (in future studies) and the community.

Points for consideration:

1) Batch effects 1: disassociation stress and more. The authors are well aware of potential biases in their study design, including both batch effects due to differences in sampling times, protocol or physical constraints (e.g. embryo size), and effects related to the their highly multiplexed protocols. There are some experiments aiming to control for these effects, for example resampling of 8-21 somite embryos (Fig S5) or usage of extra (kind of spiked-in controls) nulcei (Fig S3b). But these controls are used to support a conclusion or prove there is no significant batch effect, rather than just quantifying the effect and examine the impact of data interpretation. The most troublesome example I noticed is the analysis in Fig 2 (for example fig 2m). Here, some of the most problematic transcripts that are known to be volatile to stress, acquisition variance and handling are shown as the major markers of early vs late MMP. Genes such as HSP90aa, NPM1 (and in many cases a specific subset of ribosomal genes) can be observed at increased level following single cell disassociation, with variable induction intensity due to the protocol, time on ice etc. These genes are looking irrelevant (or at least not directly relevant) to the biology of the MMP differentiation process, and are not really associated with the interesting and most likely valid transition from caudal (Cdx1), toward Hoxa10 regulation.

I would suggest an open approach to this problem. These effects can be normalized or mitigated using a variety of techniques (including those used by the authors), as long as users of the data

are aware of them. I would like to see the Npm1 gene cluster/module defined (e.g. by co-variance analysis of the count matrix) such that an Npm1 signature (total expression from the genes in the cluster) can be computed for each cell. Then, the distribution of this signature across embryos, batches (and if needed stratified over broad cell types) can be assessed and shown in a simple boxplot. If indeed it turns out this signature is highly batch-prone, it will be advisable to try and eliminate the relevant genes from the analysis – for example in Fig 2. Then additional gene clusters that are strongly correlated or anticorrelated with the Npm1 signature can also be characterized and excluded. If the authors want to argue a signature such as Nmp1 is regulated during the developmental process (in MMPs or other lineages), a great way to support this is to show how the signature is changing in single embryo resolution over time, demonstrating the batch “boundaries” are not introducing a “gap” in the signature observed activity.

2) Batch effects 2: background noise. It is exciting to see that the overall quality and depth of the authors multiplexing approach improved substantially compared to previous versions. It will be however very helpful to add details on the possibility of batch effects resulting from potential mixing of transcripts across cells in the same experiment. I don't suggest the authors should be required to normalized such effects completely – but reporting on them will be really helpful. There is vast literature on ambient noise characterization in droplet-based technologies, and while the technology here is inherently different, the basic principle of detecting such bias can be used. The simplest metric is typically the levels (possible almost 0) or highly type-specific genes with very high expression (e.g. Hbb-y, Hba-x, some collagens) in cell types that are not supposed to express these genes and can be observed in at least two batches. Adding a supplementary figures and either clearing this point or raising awareness to this potential bias will be important.

3) Embryo sampling variation. There is much interest, both technical and biological, in the universality of the cell type composition across embryos over time. The authors can easily add this information and discuss it briefly. How technically robust is whole-embryo sampling? Are there specific lineages/types that can be more prone to under-sampling given tissue handling variation? The authors are showing embryos projected in PC space, but much more informative will be the cell type distribution, color-coded (and possibly, shown for both clusters and select subclusters). In the discussion, some comments on inter-individual differences should be added. I think it is acceptable that this cannot become an analysis goal as so much can be done in this front, but the distinction between universal developmental process and individualized differences is something the authors should mention. Regarding sex-linked differences – I also believe it is acceptable not to analyze this in this paper. But the authors should add a comment in the discussion prompting others to take such effect into account when using the new data.

Referee #3 (Remarks to the Author):

Qiu et al., A single-cell transcriptional timelapse of mouse embryonic development, from gastrula to pup

In this study, the authors focus on using single-nuclei RNA-sequencing to profile mouse development from early organogenesis through to early postpartum. The study represents a rich dataset that will no doubt be of high use to the wider scientific community – albeit, at present, this use is hindered by the lack of a high-quality website that allows the data to be easily accessible. Overall, the paper is well written and easy to follow. The biological insights are relatively high level and consistent with prior expectations – as the authors state, their aim was to provide a snapshot of different aspects of the data. Nevertheless, the lack of depth in some sections or an attempt to validate, more comprehensively, some of the novel findings (e.g., around the cell type lineage) using external data is a little disappointing. Additionally, for such a data rich paper, the computational methods employed were often highly heuristic, with a lack of robust quantification

of results – several statements / insights (e.g., trajectories / comment 15 below) really do require some statistical quantification. Similarly, I was disappointed at the terseness of the Methods section – much more information about how specific analyses were performed is needed. I strongly feel the authors need to do a more rigorous and careful job on this front. Similarly, it was extremely disappointing that, although a link to the code was provided, for half of the paper (from Figure 4 onwards) the GitHub repository simply states 'TBD'.

Despite these concerns, I am certain that this dataset will be of value to the community but would, in particular, strongly urge the authors to put the computational analyses on a firmer footing during the revision process.

1. Limits of profiling nuclei vs whole cells? Is this a particular challenge for neuronal tissues?
2. For the postpartum samples, could some of the changes be driven by the environment? If you alter the environment, do the results differ?
3. Variability between embryos within a stage? Only profiling a single embryo could lead to additional noise? How were the 75 selected embryos chosen?
4. One embryo failed entirely?
5. SF3 – label for panel b should be red and green – or better, change the green to blue in the plot for colour-blind readers. Also, why do you see a rather different profile in the LH and RH plots in SF3b? Was the quality in run 22 lower than in run 19?
6. Is the harvesting of the postpartum pups potentially a traumatic event that could influence the environment / change the transcriptome? Can this be controlled for? The number of replicates is relatively small for this crucial window (if I understand the methods correctly) – does this give any cause for concern?
7. The third doublet removal strategy is interesting, albeit rather heuristic. Moreover, it filters out a large fraction of cells (up to 13.2% in some experiments). I have several questions about this approach: i) were cell cycle genes removed prior to performing the analysis?; ii) in step 10, how were 'subclusters showing low expression of the differentially expressed genes identified in step 5' identified? Was this analysis performed in the normalized count space? Iii) For clusters of cells that fall along a developmental trajectory, does this approach remove intermediate cells? Depending upon the granularity of the clustering, I could imagine this being a severe problem for continuous cell trajectories. More generally, I would like to see more information about this approach and more detail on the type of cells excluded.
8. When the data were pooled computationally (pp33-34) was any batch correction strategy applied? From the text, it read as if samples were merged without any additional correction. I do not find the analysis presented in Fig 1d compelling evidence for the lack of a batch effect: I would prefer to see full plots of the embedding for samples processed from adjacent time points in different experiments. Additionally, it would be illustrative to consider the data that were processed in multiple runs.
9. Methods p34: you mean '12 timepoints at 1 day increments' not '1 hour increments' I think.
10. The resolution of clusters in SF4 is relatively low given the number of cells. While I understand the argument of the authors that this will be refined further moving forward, I think a somewhat deeper clustering and annotation would have been helpful for the community and would substantially increase the utility of the dataset.
11. Were the authors able to explore the intercalation of cells (e.g., work from Kat Hadjantonakis and others) from the extra-embryonic endoderm when forming the gut-tube (Figure 2)? Some such cells must contribute to the posterior portion of the gut tube.
12. The result in Fig 2l does not seem especially surprising – AP patterning is expected to be conserved across different developmental trajectories, so I am not sure why the authors find this 'striking'. I would suggest toning down the language in the text both here and when referring to Fig 2m. This reuse of regulatory programs is well known – it is a nice demonstration but not especially surprising and should be rephrased accordingly.
13. Are the manually inferred relationships shown In Figure 3c consistent with those obtained from application of computational approaches?
14. Some experimental validation of the tangram inferred location of cell types (Figure 3g) would

be important – this is a relatively straightforward experiment, given the marker genes that are available and would support the computational inference. For example, can you validate the expression of markers associated with the two LPM-subsets mapping to the inside and outside of the kidney?

15. How were the solid black lines drawn in Fig 4c? The legend talks about inference, but this feels like a very manual connection, which is challenging given such a complex series of cell states. Can the authors put these relationships on a more quantitative footing?

16. Fig 5c: the authors state that the 'extensive heterogeneity of the patterned neuroectoderm progressively diminishes as differentiating neurons become more similar with respect to their transcriptional states' – this is hard to observe from the figure – can the authors quantify this observation?

17. The description of the cell type tree felt like it belonged as a more detailed supplementary note, with more discussion being warranted around a number of factors in the heuristic algorithm. For example, are the MNNs between cell types enriched for cells that are collected from the same embryo? If so, this might suggest a technical factor could be driving a particular edge. More generally, a discussion on parameter choices and robustness of the approach would be welcome; also a discussion around known complex transitions would provide confidence in the result; in particular, in cases where there is cell type intercalation (gut tube) is this observed in the mapping? It would provide confidence in the results.

18. As an alternative to the cell type analysis presented in Fig 7b, perhaps the authors could consider connecting neighbourhoods of cells? Could the annotation of cell types across this interval be more challenging than across other parts of the developmental trajectory?

19. The lack of a user-friendly interface for exploring the data is a major hindrance and will substantially limit its utility. This needs to be resolved.

20. The GitHub repository is incomplete (e.g., TBD for Fig 4 etc) – this needs to be resolved.

Author Rebuttals to Initial Comments:

Response to Reviewers

We thank the three reviewers for their constructive feedback on the submitted version of the manuscript. A point-by-point response, which includes summaries of changes made in response to these comments, is provided below. The original reviewer comments are replicated in full in blue text, while our responses are in black text.

Response to Referee #1: pages 1-7

Response to Referee #2: pages 7-14

Response to Referee #3: pages 15-35

Referee #1 (Remarks to the Author):

The authors have generated the most comprehensive set of single nuclear data on the developing mouse to date and subjected these data to a number of insightful analyzes. I particularly enjoyed the clarity of the writing and the authors pointing out limits to the data or conclusions throughout. The care in the attention to staging collections, the significant efforts to reduce batch variability in the indexing approach and the decent depth of gene reads/nucleus will make this a lasting resource for the community. I expect that “domain” experts will rapidly enhance the analyzes here and I have erred away from nit-picking in favor of encouraging a timely publication.

We thank the reviewer for these positive comments on both the work and the writing, as well as for their advocacy of its timely publication.

My only significant comment relates to community accessibility and a pet-peeve that not enough is done beyond making data accessible to enable viewing of the data. Given the computational expertise and resources of the authors, it is a reasonable expectation for the authors to enable accessible data viewing portals that would greatly facilitate rapid querying these powerful datasets. Shiny apps have been a simple way several groups have democratized data to those without “r” expertise.

We agree in full, and apologize for not hitting the mark in terms of accessibility. In the course of our revisions, we have overhauled the website. Please take a look at the links below:

<https://atlas.gs.washington.edu/jax/>

<https://atlas.gs.washington.edu/jax/public/index.html>

In particular, we now provide interactive 3D visualizations not only for the whole dataset, but also for various subsets (e.g. major cell clusters, the specific subsets that are presented in main figures, etc.). The side-by-side views are synchronized in terms of the viewpoint on the 3D UMAP, and each of the two views can be labeled by cell type, timepoint, or the raw or normalized expression level of any gene of interest. Of note, random downsampling was used for any subset with more than 100,000 cells, in order to facilitate fast response times for the interactive aspect of the portal. However, each full subset (and/or its metadata) can be easily downloaded for focused analyses by users via links at the bottom of the page that are updated as different subsets are selected.

We plan to continue improving this interactive website, and welcome any suggestions. We are also in discussions with CZ about how to make these data available at the CZ “CELL by GENE” data portal.

A few specific minor comments readily addressable:

1) The authors note their analysis suggest Brachyury is within posterior gut cells. Indeed, Schifferl et al (2021) Development Figure 1C” shows this nicely.

Thank you for pointing this out these beautiful images in the Shifferl et al. (2021) paper, which we were not aware of. We have revised the text and have added a reference to the suggested paper.

Original text: “As *T* expression is classically associated with the notochord and posterior mesoderm in the mouse literature, we were initially surprised to see strong *T* expression in the inferred posterior hindgut, coincident with the expression of posterior *Hox* genes. However, this expression pattern is consistent with the ancestral role of *T* in the closing of the blastopore and hindgut defects in *Drosophila brachyenteron* and *Caenorhabditis elegans mab-9* mutants.”

Revised text: “As *T* expression is classically associated with the notochord and posterior mesoderm in the mouse literature, we were initially surprised to see strong *T* expression in the inferred posterior hindgut, coincident with the expression of posterior *Hox* genes. **To our knowledge, this expression pattern was only recently documented** (Schifferl et al. 2021), and is consistent with the ancestral role of *T* in the closing of the blastopore as well as hindgut defects in *Drosophila brachyenteron* and *Caenorhabditis elegans mab-9* mutants.”

2) The statement here is a little confusing: “specification of these posterior and anterior trajectories in late gastrulation is initiated by interactions between *Gdnf* and *Ret*”. *Gdnf* and *Ret* demarcate different populations but there is no evidence this signaling axis specifies cell types?

We apologize for any confusion. The statement in question was mainly based on previous literature (Majumdar et al. 2003), rather than our own data. In that paper, the authors propose that: “*Wnt11* and *Ret/Gdnf* cooperate in a positive autoregulatory feedback loop to coordinate ureteric branching by maintaining an appropriate balance of *Wnt11*-expressing ureteric epithelium and *Gdnf*-expressing mesenchyme”. On further review of the literature, what is presented in this paper appears to be a model rather than textbook knowledge. However, the importance of these genes for kidney development is well established via genetics (*i.e.* they are not merely markers). We have updated the text accordingly.

Original text: “The specification of these posterior and anterior trajectories in late gastrulation is initiated by interactions between *Gdnf* and *Ret* (Majumdar et al. 2003), followed by their progression to metanephric mesenchyme and the ureteric bud, respectively, around E10.25, and then to specific functional components of the nephron.”

Revised text: “The posterior and anterior trajectories in late gastrulation are marked by *Gdnf* and *Ret* expression, respectively, critical genes for normal kidney development (Costantini and Shakya 2006; Majumdar et al. 2003). These trajectories then progress to the metanephric mesenchyme and ureteric bud, respectively, around E10.25, and then to specific functional components of the nephron.”

3) The authors write “Of note, we observe “convergence” of the posterior and anterior trajectories in collecting duct intercalated cells (cluster 4 in Fig. 3a-b). More detailed investigation suggests that the posterior intermediate mesoderm may also contribute to the collecting duct, although lineage analysis

would be necessary to confirm this (Fig. 3d-e; Supplementary Fig. 7f)". Ransick et al., (2019) Dev Cell demonstrated with lineage tracing the dual origin of intercalated cell types from the distal nephron and ureteric lineages – data visible at Kidney Cell Explorer (<https://cello.shinyapps.io/kidneycellexplorer/>).

Thank you for bringing this to our attention. We have revised the text and added the reference.

Original text: "More detailed investigation suggests that the posterior intermediate mesoderm may also contribute to the collecting duct, although lineage analysis would be necessary to confirm this."

Revised text: "More detailed investigation suggests that the posterior intermediate mesoderm may also contribute to the collecting duct, consistent with lineage tracing experiments demonstrating the dual origin of intercalated cell types from the distal nephron and ureteric lineages (Ransick et al. 2019)."

4) The authors write "We can also distinguish two subsets of LPM-derivatives mapping to the kidney, one to the inside and the other to the surface, which may correspond to renal stroma and the renal pericytes and mesangial cells, respectively (Fig. 3g)". It is likely that the *Foxd1* population (cluster 4 labelled renal stroma) contains much of the stromal cell diversity. Fate mapping studies (Kobayashi et al 2014 Stem Cell Reports) indicate that that *Foxd1*⁺ cells give rise to the renal pericytes and mesangial cells. May be good to clarify that the observation should not be interpreted as two different origins. Parenthetically, I am surprised that cluster 14 separates so clearly from cluster 4. Perhaps there is an alternative ID for these cells?

Thank you for these insightful suggestions. In retrospect, we worried that we relied too heavily on spatial mapping to annotate these two LPM-derived subsets, and your comment led us to revisit the data using more specific marker genes.

First, as shown in the figure below (co-embedding of cells annotated as renal stroma or renal pericytes & mesangial cells), we found that the subset of cells that we annotated as renal pericytes & mesangial cells (**Fig. R1a**) express classic kidney marker genes, *Eya1* and *Pax2* (**Fig. R1b**), while spatial transcriptomics data shows these genes are expressed cortically in the nephrogenic zone (**Fig. R1b**). These genes are not expressed in the cluster that we annotated as renal stromal cells.

Second, we found that *Lrriq1* and *Cfh* genes are specifically expressed in the cells that we annotated as renal stromal cells (**Fig. R1a**), while spatial transcriptomics map the expression of these genes to the medulla of the kidney (**Fig. R1b**). These genes are not expressed in the cluster that we annotated as renal pericytes & mesangial cells.

Foxd1 gene expression has a distinct pattern, expressed in subsets of both the renal stromal and renal pericytes & mesangial cell clusters. This is despite the fact that like *Eya1* and *Pax2*, it maps to cortically in spatial transcriptomics data (see third row of panels, which for *Foxd1* include both virtual and real ISH images, in **Fig. R1b**). This incongruence is not easily explained by temporal factors, as *Foxd1* appears to be expressed in renal stromal cells at E14.5, which is when these MOSTA data were collected (**Fig. R1d**).

It is interesting to note that while *Cfh* appears uniformly expressed in renal stromal cells, *Lrriq1* and *Foxd1* exhibit anticorrelated expression patterns (lower three rows of **Fig. R1b**). Once again, these patterns do not appear correlated with time (lower panel of **Fig. R1a**; **Fig R1d**). Turning to the *in situ*

images, *Foxd1* appears cortical, *Lrriq1* appears medullary, and *Cfh1* more diffuse (lower three rows of **Fig. R1b**). Thus, a simple explanation may be spatial heterogeneity within renal stromal cells, with cortical renal stromal cells expressing *Foxd1* (coincident with renal pericytes & mesangial cells, also cortically located), medullary renal stromal cells expressing *Lrriq1*, and all renal stromal cells expressing *Cfh*. After forming this hypothesis, we examined the spatial expression patterns of several other genes that were heterogeneously expressed in renal stromal cells. The results of these analyses were consistent with our hypothesis, as genes whose expression patterns matched *Foxd1* within renal stromal cells (e.g. *Ntn1*, *Zbtb7c*, *Sema3d*) appeared cortical in the *in situs*, while genes whose expression patterns matched *Lrriq1* within renal stromal cells (e.g. *Zeb2*, *Plcb1*) appeared medullary in the *in situs* (**Fig. R2**).

Returning to the potential origin(s) of these cell populations, we supplemented the co-embedding with intermediate mesoderm, metanephric mesenchyme and splanchnic mesoderm (**Fig. R1c**). We observe subsets of renal pericytes & mesangial cells whose transcriptional profiles form a continuum with posterior intermediate mesoderm, metanephric mesenchyme, and splanchnic mesoderm, suggesting multiple origins. Of note, not all of these subsets persist until P0.

Renal stromal cells, on the other hand, do not appear to be closely associated with the intermediate mesoderm or metanephric mesenchyme (consistent with their lack of expression of *Eya1* or *Pax2*), and instead appear transcriptionally continuous with the splanchnic mesoderm (**Fig. R1c**).

Based on these observations, although subsets of both express *Foxd1*, we hypothesize that renal pericytes & mesangial cells and renal stromal cells have different origins. Lineage tracing is necessary to definitively test this hypothesis, but would be beyond the scope of the current manuscript. Therefore, we have decided to leave this question open for future research.

We have revised the text as follows (but welcome any suggestions for modifying how we are framing our observations):

Original text: “We can also distinguish two subsets of LPM-derivatives mapping to the kidney, one to the cortex and the other more heterogeneously, which may correspond to renal stroma and the renal pericytes and mesangial cells, respectively (**Fig. 3g**).”

Revised text: “We can also distinguish two subpopulations of LPM-derivatives mapping to the kidney, one to the cortex and the other more heterogeneously distributed within the renal mesenchyme, which we believe correspond to renal pericytes & mesangial cells and renal stromal cells, respectively. Although both subpopulations express *Foxd1*, supporting their assignment to the kidney, focused analyses are consistent with their having distinct origins (**Supplementary Fig. 13**). However, lineage tracing experiments would be necessary to test this hypothesis. Of note, renal stromal cells exhibited gene expression heterogeneity along what may be the cortical-medullary spatial axis, of genes including *Foxd1* (cortical), *Netrin-1* (cortical) and *Zeb2* (medullary) (**Supplementary Fig. 14**).”

Reviewer Figure 1 (Supplementary Figure 13 in the revised manuscript). Assessing the potential origins of LPM subsets annotated as renal pericytes & mesangial cells and renal stromal cells. a, Re-embedded 2D UMAP of 39,468 cells from renal pericytes & mesangial cells and renal stromal cells. Cells are colored by either annotation (top) or timepoint (bottom, after downsampling to a uniform number of cells per time window). **b**, Left: The same UMAP as in panel a, colored by gene expression of marker genes which appear specific to renal pericytes & mesangial cells (*Pax2*⁺, *Eya1*⁺) or renal stromal cells (*Lrriq1*⁺, *Cfh*⁺). *Foxd1* is expressed in a subset of both cell types. Middle: Virtual *in situ* hybridization (ISH) images of individual genes from one selected section (E1S1) from E14.5 of the *Mosta* data (<https://db.cngb.org/stomics/mosta/>). Right: *In situ* hybridization (ISH) images of individual genes were obtained from the Jackson Laboratory Mouse Genome Informatics (MGI) website (<https://www.informatics.jax.org/>). The original reference for these images is (Diez-Roux et al. 2011). **c**, Re-embedded 2D UMAP of 206,908 cells from renal pericytes & mesangial cells, renal stromal cells, anterior intermediate mesoderm, posterior intermediate mesoderm, metanephric mesenchyme, and splanchnic mesoderm. Cells are colored by either their initial annotations (top) or timepoint (bottom, after downsampling to a uniform number of cells per time window). **d**, The average normalized expression of *Foxd1* over time is shown for renal pericytes & mesangial cells (top) and renal stromal cells (bottom). Gene expression was normalized by the size factor estimated by Monocle/3.

Reviewer Figure 2 (Supplementary Figure 14 in the revised manuscript). Spatial heterogeneity within the renal stromal cells. Left: The same UMAP as in Fig. R1a, colored by gene expression of marker genes which appear specific to two subsets of renal stromal cells: medullary renal stromal cells (*Zeb2*⁺, *Plcb1*⁺) and cortical renal stromal cells (*Ntn1*⁺, *Zbtb7c*⁺, *Sema3d*⁺), respectively. Middle: Virtual *in situ* hybridization (ISH) images of individual genes from one selected section (E1S1) from E14.5 of the *Mosta* data (<https://db.cngb.org/stomics/mosta/>). Right: *In situ* hybridization (ISH) images of individual genes were obtained from the Jackson Laboratory Mouse Genome Informatics (MGI) website (<https://www.informatics.jax.org/>). The original reference for these images is (Diez-Roux et al. 2011).

5) The authors write “The apparent bifurcation of the proximal tubule corresponds to major differences in the transcriptional state of cells from embryos obtained before birth (E18.75 or earlier) vs. after birth (P0) (cluster 9 in Fig. 3a-b; Supplementary Fig. 7d). We return to this observation in the final section of the manuscript.” However, there is not much more presented in the later discussion. If the authors are to single out proximal tubule cells, I suggest adding the differential gene expression data to the relevant supplementary tables so the data can be viewed.

We do mention the proximal tubule cells in introducing final section:

“In the course of our analyses of this time-lapse, we anecdotally noted that for certain cell types, cells derived from P0 pups appeared very well separated from their fetal pseudoancestors, in sharp contrast with other cell types across the same temporal interval as well as with even these same cell types at all prior temporal intervals. The proximal tubule is one example of this phenomenon, discussed briefly above (cluster 9 in **Fig. 3a-b**; **Supplementary Fig. 10d**). However, a similar pattern was also noted for hepatocytes, adipocytes, and various cell types of the lungs and airways (**Fig. 7a**).”

However, as noted, we do not provide any details. We have addressed this as suggested, by adding a new table (**Supplementary Table 28**) that provides the differentially expressed genes between E18.75 and P0 for each of the top 20 cell types listed in **Fig. 7b**, which are ranked by the degree of transcriptional disjunction between stages immediately before vs. after birth. Proximal tubule cells are one of these 20 most highly ranked cell types. A new sentence referencing this table, which is part of the **Fig. 7b** legend, reads as follows:

New sentence: “Differentially expressed genes for the 20 most highly ranked cell types in this analysis are shown in **Supplementary Table 28**.”

Referee #2 (Remarks to the Author):

The paper by Qiu, Martin, Welch et al is presenting a remarkable new dataset including 11.4M QC positive single cell profiles acquired from 74 mouse embryo sampled carefully over the entire developmental time axis starting at E8 and post birth. This is a dream dataset, representing the culmination of a remarkable effort of technology development from the Shendure lab and their collaborators. This data is collected, controlled and processed in a standardized and well documented fashion, and is now providing an unprecedentedly deep and precise basis for understanding mammalian embryonic development.

The procedures for embryo staging and selection, single cell profiling and QC are at a very high standard. The authors are doing an excellent job in describing how the new resource was constructed. In that respect, the expert staging and selection, and then the quantification of overall embryo size and growth rate is very important. It is understandable that the bulk of the results presented in the text are highlighting specific lineages and stages, rather than aiming at a hopeless attempt for comprehensive detailed description of the process. The authors are also refraining from unneeded overly theoretical statements or arguments on principles, and I commend them on avoiding this and focusing on the data and how to use it. I would advise against requests for more validation experiments on the data discussed in Figure 2-7 – and while I have some practical analysis/bias concerns below, it is clear that

all can be addressed with textual changes, simple additional analysis, or, in the worst case, omission of some panels for which evidence is not conclusive. This resource can change the way by which we approach embryonic development of specific lineages and tissues given a much-needed global context. This is possible even given the present form of the data, and will be intensified after much more analysis, tools, integration with spatial data and epigenomics profiles will be added on top of it by the authors (in future studies) and the community.

We are grateful to the reviewer for these positive comments on the experimental design, the data itself, and our conservative approach to its presentation. We also appreciate their advocacy of its timely publication.

Points for consideration:

1) Batch effects 1: disassociation stress and more. The authors are well aware of potential biases in their study design, including both batch effects due to differences in sampling times, protocol or physical constraints (e.g. embryo size), and effects related to their highly multiplexed protocols. There are some experiments aiming to control for these effects, for example resampling of 8-21 somite embryos (Fig S5) or usage of extra (kind of spiked-in controls) nuclei (Fig S3b). But these controls are used to support a conclusion or prove there is no significant batch effect, rather than just quantifying the effect and examine the impact of data interpretation. The most troublesome example I noticed is the analysis in Fig 2 (for example Fig 2m). Here, some of the most problematic transcripts that are known to be volatile to stress, acquisition variance and handling are shown as the major markers of early vs late MMP. Genes such as HSP90aa, NPM1 (and in many cases a specific subset of ribosomal genes) can be observed at increased level following single cell disassociation, with variable induction intensity due to the protocol, time on ice etc. These genes are looking irrelevant (or at least not directly relevant) to the biology of the MMP differentiation process, and are not really associated with the interesting and most likely valid transition from caudal (Cdx1), toward Hoxa10 regulation.

I would suggest an open approach to this problem. These effects can be normalized or mitigated using a variety of techniques (including those used by the authors), as long as users of the data are aware of them. I would like to see the Npm1 gene cluster/module defined (e.g. by co-variance analysis of the count matrix) such that an Npm1 signature (total expression from the genes in the cluster) can be computed for each cell. Then, the distribution of this signature across embryos, batches (and if needed stratified over broad cell types) can be assessed and shown in a simple boxplot. If indeed it turns out this signature is highly batch-prone, it will be advisable to try and eliminate the relevant genes from the analysis – for example in Fig 2. Then additional gene clusters that are strongly correlated or anticorrelated with the Npm1 signature can also be characterized and excluded. If the authors want to argue a signature such as Npm1 is regulated during the developmental process (in MMPs or other lineages), a great way to support this is to show how the signature is changing in single embryo resolution over time, demonstrating the batch “boundaries” are not introducing a “gap” in the signature observed activity.

We thank the reviewer for advocating that we push harder to identify, as well as to make more transparent to the reader, potential technical artifacts related to embryo preservation and handling that might be bleeding into the results. Please note we are NOT dissociating cells in the typical manner required by other protocols such as that of the 10X Genomics kit. Thus, some of the cited technical variables (e.g. time on ice) are not relevant. Rather, embryos were flash-frozen immediately after being

harvested and photographed. They were subsequently shipped to Seattle, where subsets were selected for lysis without dissociation (which is a major benefit of analyzing nuclei rather than cells). Even during powdering and lysis, they were kept frozen, until they were in the lysis buffer.

However, given that the cited genes are not obviously related to the underlying biology, and that there may be technical artifacts associated with our procedures to which we are blind, we proceeded with the recommended analysis as follows: 1) We downsampled the dataset to ~1M cells using *geosketch* (Hie et al. 2019), and then performed k-means clustering to ensure that each cluster contained roughly 500 cells. 2) We aggregated UMI counts for cells within each cluster to generate 2,289 meta-cells, and then normalized the UMI counts for each meta-cell followed by log₂-transformation. 3) We performed Pearson correlation between *Npm1* and each protein-coding gene. We selected the genes with correlation coefficients > 0.6 (738 genes, ~3% of the total protein coding genes; *Hsp90aa1* was one of the genes passing this threshold with a correlation coefficient of 0.75). A brief gene set enrichment analysis suggests that the module is associated with RNP complexes (corrected p-value = 1.4e-105), cytoplasmic translation (corrected p-value = 2.8e-90), and ribosomal proteins (corrected p-value = 7.4e-71). 4) We summed the normalized UMI counts for these genes to calculate a *Npm1* signature for individual cells. The distribution of *Npm1* signatures are shown below for different sci-RNA-seq3 experiments, embryo harvest dates, litters of embryos, and shipment batches (**Fig. R3a**).

The results are notable for a decrease in this signature over developmental time. The pattern is consistent across cell types, albeit more pronounced for some cell types than others (e.g. mesoderm > neuronal; **Fig. R3b**). Although there are differences in the usage of this module with different harvest dates, litters or shipment batches, there is no consistent trend, and the variation is likely simply due to the fact that different biological stages are represented in different batches. In summary, the usage of the *Npm1*-defined module declines with biological age in the overall dataset. This is consistent with the analyses represented in **Fig. 2m**, where we show that the expression of *Npm1* and transcripts encoding Hsp90 isoforms decline between early and late somite stages.

If the reviewer would like to check the early vs. late stage variable against the batch information shown in **Fig. R3**, early NMPs derived from embryos harvested on 2/8/21 and were processed in run 4, while late NMPs derived from embryos harvested on on multiple dates (1/13/21, 1/14/21, 10/29/21, 11/11/21, 11/3/20, 3/16/21, 3/3/21, 8/17/21, and 9/30/21) and were processed in runs 15 and 17. Both early and late NMPs were derived from shipment batch 2.

The text has been revised as follows to reflect the new analysis:

Original text: “Genes reproducibly associated with early somite counts in both NMPs and the gut were strongly enriched for Myc targets, and also included *Lin28a*, a deeply conserved regulator of developmental timing, and multiple isoforms of Hsp90 (**Fig. 2m; Supplementary Table 11**), possibly reflecting greater proliferation.”

Revised text: “Genes reproducibly associated with early somite counts in both NMPs and the gut were strongly enriched for Myc targets, and also included *Lin28a*, a deeply conserved regulator of developmental timing (**Fig. 2m; Supplementary Table 11**). Other genes such as *Npm1* and *Hsp90* isoforms are plausibly associated with batch effects. However, analysis of a module of genes correlated with *Npm1* found it to be declining with developmental time across the entire time series, rather than correlated with batch variables (**Supplementary Fig 9**).”

Reviewer Figure 3 (Supplementary Figure 9 in the revised manuscript). Checking the consistency of *Npm1* signatures across different batches. **a**, First, we downsampled the dataset to ~1M cells using *geosketch* (Hie et al. 2019) and performed k-means clustering to ensure that each cluster contained roughly 500 cells. Second, we aggregated UMI counts for cells within each cluster to generate 2,289 meta-cells, and normalized the UMI counts for each meta-cell followed by log₂-transformation. Third, we performed Pearson correlation between each protein-coding gene and *Npm1*, and selected genes with correlation coefficients > 0.6 (738 genes, ~3% of the total protein coding genes). A gene set enrichment analysis suggests that the module is associated with RNP complexes (corrected p-value = 1.4e-105), cytoplasmic translation (corrected p-value = 2.8e-90), and ribosomal proteins (corrected p-value = 7.4e-71). Finally, we summed the normalized UMI counts of these genes to calculate a *Npm1* signature for individual cells. The resulting *Npm1* signatures are subsetted in four plots, from left to right: by sci-RNA-seq3 experiment, embryo harvest date, litter of embryos, or shipment batch. **b**, Same as panel a, but further stratified by the top 10 abundant major cell clusters.

2) Batch effects 2: background noise. It is exciting to see that the overall quality and depth of the authors multiplexing approach improved substantially compared to previous versions. It will be however very helpful to add details on the possibility of batch effects resulting from potential mixing of transcripts across cells in the same experiment. I don't suggest the authors should be required to normalized such effects completely – but reporting on them will be really helpful. There is vast literature on ambient noise characterization in droplet-based technologies, and while the technology here is inherently different, the basic principle of detecting such bias can be used. The simplest metric is typically the levels (possible almost 0) or highly type-specific genes with very high expression (e.g. *Hbb-y*, *Hba-x*, some collagens) in cell types that are not supposed to express these genes and can be observed in at least two batches. Adding a supplementary figures and either clearing this point or raising awareness to this potential bias will be important.

This is a great point and we thank the reviewer for advocating that we make this issue transparent to the reader. To address this comment, we examined the expression of four hemoglobin genes (*Hbb-y*, *Hba-x*, *Hbb-bt*, and *Hbb-bs*) and 2 collagen genes (*Col1a1* and *Col2a1*) across 26 major cell clusters. These genes are specific and highly expressed in primitive erythroid cells (*Hbb-y*, *Hba-x*), definitive erythroid cells (*Hbb-bt*, *Hbb-bs*), pre-osteoblasts (*Col1a1*) or early chondrocytes (*Col2a1*). As shown in **Fig. R4**, we do observe some ambient noise, as an appreciable fraction of “other cell type” cells exhibit a handful of read counts for these genes. We have added the following text (and included this figure in the supplement) to raise awareness of this issue:

New text: “We also checked for ambient noise (e.g. as might be due to transcript leakage) by examining highly abundant, highly cell-type-specific genes such as hemoglobins and collagens, and found it present at low levels, e.g. the mean number of UMIs for *Hbb-bs* was 10.8 in definitive erythroid cells and 0.26 in all other cells, and for *Col1a1* was 186 in pre-osteoblasts vs. 1.23 in all other cells (**Supplementary Fig. 5**).”

Reviewer Figure 4 (Supplementary Figure 5 in the revised manuscript). Ambient noise (e.g. as might be due to transcript leakage) was assessed by examining hemoglobin and collagen transcripts. The distribution of the number of reads mapping to each selected hemoglobin or collagen gene across cells, for the cell type that is expected to express that gene at high levels (red) vs. all other cell types (blue). The mean UMI counts of cells in each group are also reported.

3) Embryo sampling variation. There is much interest, both technical and biological, in the universality of the cell type composition across embryos over time. The authors can easily add this information and discuss it briefly. How technically robust is whole-embryo sampling? Are there specific lineages/types that can be more prone to under-sampling given tissue handling variation? The authors are showing embryos projected in PC space, but much more informative will be the cell type distribution, color-coded (and possibly, shown for both clusters and select subclusters). In the discussion, some comments on inter-individual differences should be added. I think it is acceptable that this cannot become an analysis goal as so much can be done in this front, but the distinction between universal developmental process and individualized differences is something the authors should mention. Regarding sex-linked differences – I also believe it is acceptable not to analyze this in this paper. But

the authors should add a comment in the discussion prompting others to take such effect into account when using the new data.

Thank you for this insightful comment. Please note that we do present the proportion of cells from individual major cell clusters across timepoints in **Fig. 1e**, but the y-axis is scaled to the estimated number of cells in the whole mouse embryo at that timepoint. For selected developmental trajectories, we also present the estimated absolute number of cells over time (these are in scattered supplementary figures, but we have collated them to **Fig. R5** below).

A few additional points of relevance:

First, unfortunately, our sampling strategy does not really allow us to systematically investigate this question, because we only sampled one embryo for most timepoints. That being said, there are some intriguing observations of relevance. For example, the individual embryo staged as E12.25 lacked multiple different types of renal cells (see **Fig. R5a** below, with a blue downward facing arrow at the top to draw your attention to it). We believe that this particular individual may have had aberrant renal development, although we cannot be absolutely certain.

Second, in a recent preprint (currently in revisions), we applied sci-RNA-seq3 to profile 101 embryos of 26 genotypes at embryonic stage E13.5 (Huang et al. 2022). The 26 genotypes include multiple mouse mutants of varying severities as well as wildtype controls, and each genotype was represented by 4 E13.5 embryos. We used several analytical frameworks to detect differences in cell composition across 52 cell types that were identified from the dataset. We also performed a statistical power analysis based on simulation data to determine the number of embryos required for each group to detect a difference in cell composition. That study (and that data) are likely a better place to assess interindividual variation because we have multiple individuals sampled per genotype at the same developmental stage.

Third, we acknowledge that we have not performed any sex-linked analysis, although we tried to profile embryos from both sexes by alternating wherever possible (**Fig. 1d**). The main reason for this is that we do not want to extend the paper to an unwieldy length, so we decided to leave it open (particularly as many sex-differences will first manifest postnatally, which we hope to sample next).

We have added the following paragraph to the Discussion to reflect these points:

New text: “A limitation of our sampling strategy is that we only profiled a single embryo for most timepoints, such that we are unable to conduct a systematic analysis of interindividual variation at any given timepoint. We do observe hints of such variation, e.g. multiple different types of renal cells were not detected at E12.25, which may reflect aberrant renal development in that individual embryo (**Supplementary Fig. 10b**). However, such analyses may be better pursued through other datasets, e.g. our profiling of 101 embryos (of 26 genotypes) staged at E13.5, also by sci-RNA-seq3 (Huang et al. 2022). On a related point, although both sexes are represented in the dataset (as we alternated between adjacent timepoints), we have not yet delved into sex differences, and this remains one of many avenues of investigation for which we hope researchers in the field will find these data useful.”

Reviewer Figure 5. Transcriptional heterogeneity in renal, mesodermal, and retinal development. a, Reproduced from **Supplementary Fig. 10b**. The predicted absolute number (log2 scale) of cells of each renal cell type at each timepoint. The predicted absolute number was calculated by the product of its sampling fraction in the overall embryo and the predicted total number of cells in the whole embryo at the corresponding timepoint (**Fig. 1e**). For each row, the first timepoint with at least 10 cells assigned that cell type annotation is labeled, and all observations prior to that timepoint are discarded. Blue arrows highlight an individual embryo staged as E12.25 lacked multiple different types of renal cells. **b,** Reproduced from **Supplementary Fig. 15a**. The predicted absolute number (log2 scale) of cells of each mesoderm cell type at each somite count. The predicted absolute number was calculated by the product of its sampling fraction in the overall embryo and the predicted total number of cells in the whole embryo at the corresponding timepoint. Because cell numbers were only predicted for the broader bins (**Fig. 1e**), rather than individual somite counts, these were used for roughly corresponding sets (0-12 somite stage: E8.5; 14-15 somite stage: E8.75; 16-18 somite stage: E9.0; 20-23 somite stage: E9.25; 24-26 somite stage: E9.5; 27-31 somite stage: E9.75; 32-34 somite stage: E10.0). For each row, the first somite count with at least 10 cells assigned that cell type annotation is labeled, and all observations prior to that somite count are discarded. **c,** Reproduced from **Supplementary Fig. 16e**. The predicted absolute number (log2 scale) of cells of each retinal cell type at each timepoint. The predicted absolute number was calculated by the product of its sampling fraction in the overall embryo and the predicted total number of cells in the whole embryo at the corresponding timepoint (**Fig. 1e**). For each row, the first timepoint with at least 10 cells assigned that cell type annotation is labeled, and all observations prior to that timepoint are discarded.

Referee #3 (Remarks to the Author):

Qiu et al., A single-cell transcriptional timelapse of mouse embryonic development, from gastrula to pup

In this study, the authors focus on using single-nuclei RNA-sequencing to profile mouse development from early organogenesis through to early postpartum. The study represents a rich dataset that will no doubt be of high use to the wider scientific community – albeit, at present, this use is hindered by the lack of a high-quality website that allows the data to be easily accessible. Overall, the paper is well written and easy to follow. The biological insights are relatively high level and consistent with prior expectations – as the authors state, their aim was to provide a snapshot of different aspects of the data. Nevertheless, the lack of depth in some sections or an attempt to validate, more comprehensively, some of the novel findings (e.g., around the cell type lineage) using external data is a little disappointing. Additionally, for such a data rich paper, the computational methods employed were often highly heuristic, with a lack of robust quantification of results – several statements / insights (e.g., trajectories / comment 15 below) really do require some statistical quantification. Similarly, I was disappointed at the terseness of the Methods section – much more information about how specific analyses were performed is needed. I strongly feel the authors need to do a more rigorous and careful job on this front. Similarly, it was extremely disappointing that, although a link to the code was provided, for half of the paper (from Figure 4 onwards) the GitHub repository simply states ‘TBD’.

Despite these concerns, I am certain that this dataset will be of value to the community but would, in particular, strongly urge the authors to put the computational analyses on a firmer footing during the revision process.

Thank you for your feedback on our manuscript. We sincerely apologize that the Methods were terse on analytical details and the GitHub repository incomplete. In the course of our revisions, we have expanded the Methods, completed the GitHub repository, and created a new website for data visualization.

1. Limits of profiling nuclei vs whole cells? Is this a particular challenge for neuronal tissues?

We use nuclei instead of cells because, to our knowledge, there is no protocol to dissociate whole, post-gastrulation embryos at the cellular level without resulting in serious biases. Although this can be overcome to some extent at pre-gastrulation stages (e.g. as was done in Pijuan-Sala *et al.* 2019), that work also required immediate processing of freshly harvested embryos through dissociation and onwards to single cell profiling, with no stopping point.

In contrast, in our experience, nuclear isolation protocols can be performed on flash-frozen whole embryos from later stages (or even postnatal stages) without resulting in overt biases. The flash-freezing was critical, because it allowed us to preserve embryos immediately after harvesting and photographing them, to ship them to a separate site, and to process them through nuclear isolation and single cell profiling in a limited number of batches. We are essentially freezing them in time, which enables precise staging, and conducting nuclear isolation on these frozen samples is a much simpler process than dissociation of fresh tissue into whole cells, especially for neurons, whose long axons make whole-cell isolation more difficult.

The downside is of course that we lose cytoplasmic transcripts, but sci-RNA-seq3 has been greatly improved for nuclear profiling since the original protocol, and now consistently yields thousands of UMIs per nucleus from diverse tissues (Martin et al. 2022). We could have obtained more UMIs by sequencing more deeply (the average PCR duplicate rate across all the 15 sci-RNA-seq3 experiments is 50.65%, **Supplementary Table 2**), but we did not feel that this factor was limiting for our analyses.

To the second question, in the present data, we did not observe an appreciable difference in UMI counts for neuronal vs. non-neuronal tissues. In fact, the UMI counts for neuronal cells (e.g. CNS neurons, intermediate neuronal progenitors, and neural crest-PNS neurons) were higher (median 3,259; mean 4,183) than the UMI counts for non-neuronal cells (median 2,547; mean 3,181).

2. For the postpartum samples, could some of the changes be driven by the environment? If you alter the environment, do the results differ?

We agree that it seems likely that the transition from the uterine to extrauterine environment is driving these changes through various physiological triggers. For example, in hepatocytes, genes involved in gluconeogenesis are sharply upregulated after birth, including *Ppargc1a*, which encodes Pgc-1 α , a master regulator of hepatic gluconeogenesis in the liver (Liang and Ward 2006), which is plausibly driven by changes in blood glucose levels after the newborn is cut off from maternal circulation. In brown adipocytes, *Irf4* and *Ppargc1a* are upregulated. *Irf4* is a cold-induced master regulator of thermogenesis, while Pgc-1 α partners with *Irf4* to drive the expression of *Ucp1* and uncoupled respiration (Kong et al. 2014). These changes are plausibly driven by the sharp drop in temperature upon birth (Rowland et al. 2015).

Of note, one major environmental difference between the original series vs. replicate birth-series experiment, is that in the birth-series experiment, all newborns (both naturally and C-section delivered) mice did not nurse. In contrast, we presume that the original P0s had nursed given that the mother had completed delivery and had settled down with her litter. However, the dramatic changes were still observed in certain cell types, meaning that we can rule out nursing as driving those changes. Stepping back, we imagine that changes in metabolite and hormone levels, oxygen levels, temperature, and possibly other environmental differences between the uterine and extrauterine environment (and most likely, some combination of these factors) are driving the rapid changes that we observe. Although we plan to systematically investigate this in the future in greater depth, we argue that this is a major undertaking and beyond the scope of the current manuscript.

3. Variability between embryos within a stage? Only profiling a single embryo could lead to additional noise? How were the 75 selected embryos chosen?

The embryos were selected to maximize our temporal resolution, i.e. generally one embryo per timepoint, alternating between males and females wherever possible. This is discussed in more detail in the Methods section ("Generating data using an optimized version of sci-RNA-seq3") of the paper. The question about interindividual variability and noise is a good one, although not systematically addressable through this dataset. Please see our response to Comment #3 from Reviewer #2, on pages 12-13 of this document.

4. One embryo failed entirely?

Correct, one E14.5 sample was excluded from our downstream analysis. During initial quality checking, we found that this sample was almost entirely missing neuronal samples, when co-embedded with five samples from E14.0 to E14.75 (**Fig. R6**). We suspect that this particular sample had been divided during pulverization, and that the anterior-most portion of the frozen embryo was somehow lost during processing. Therefore, we excluded it from all downstream analyses.

Reviewer Figure 6. E14.5 had a grossly reduced proportion of neuronal cells. Re-embedded 3D UMAP of 1,195,179 cells from five different timepoints of embryos (E14.0 to E14.75). All of the samples were profiled in the same sci-RNA-seq3 experiment. The cells were plotted in five separate panels, with each panel containing only cells from a single timepoint. The circled region in the E14.5 plot corresponds to neuronal cells, which are grossly depleted in that sample relative to other timepoints.

The reviewer might be wondering why we did not also exclude the E12.25 embryo referenced in our response to Comment #3 from Reviewer #2, in which we noticed that multiple types of renal cells, including proximal tubule cells and metanephric mesenchyme (**Fig. R5a**), were missing. The difference is that in that case, the apparently absent cell types were only expected to be present in small numbers, other related cell types were not missing, and the relevant tissue was nascent, internal and well protected. In contrast, the massive underrepresentation of neurons in the E14.5 embryo was easily explained by technical loss of a gross portion of the embryo during pulverization.

5. SF3 – label for panel b should be red and green – or better, change the green to blue in the plot for colour-blind readers. Also, why do you see a rather different profile in the LH and RH plots in SF3b? Was the quality in run 22 lower than in run 19?

Thank you for your suggestion. We have adjusted the plot to adjust the green to blue for color-blind readers, as reproduced below (**Fig. R7**).

To remind the reviewer and as discussed in the figure legend, for a handful of timepoints, we profiled extra nuclei in some sci-RNA-seq3 experiments to ensure sufficient coverage. Here we sought to leverage those instances to check for potential batch effects across experiments. For this, on the embedding learned from all of the data, we asked whether these cells' profiles are more similar to cells from the same experiment or, alternatively, cells from the same time window. The percentages of individual cells' nearest neighboring cells from the two groups (cells from the time window vs. cells from the same experiment) are presented in a histogram. For the left plot, the contrast was between E14.75 and E17-E17.75 (run 22), while for the right plot, the contrast was between E13.5 & E13.75 and E10.5-E11.0 (run 19). The simplest explanation for the difference between the profiles is that although a

similar amount of absolute time (~2-3 days), the rate of change is much, much greater during early development, such that greater differences are expected between cells of different absolute ages. In other words, the contrast between an E10.5 and E13.5 cell of the same type is expected to be much greater than the contrast between an E14.75 and E17.75 cell of the same time.

Reviewer Figure 7. A knn-based analysis of sci-RNA-seq3 data suggests that batch effects are relatively minor. This is reproduced from **Supplementary Fig. 3b**. For two examples, we performed a k -nearest neighbors (k NN, $k = 10$) approach in the global 3D UMAP to find the nearest neighboring cells either from the same experiment (red) or the same time window but a different experiment (blue). The percentages of the nearest neighboring cells from the two groups for individual cells are presented in the histogram. In both examples, we observe that nearest neighbors are overwhelmingly cells from a different experiment (but the same time window), rather than cells from the same experiment (but a different time window).

6. Is the harvesting of the postpartum pups potentially a traumatic event that could influence the environment / change the transcriptome? Can this be controlled for? The number of replicates is relatively small for this crucial window (if I understand the methods correctly) – does this give any cause for concern?

The reviewer raises an excellent point regarding the potential for harvest-associated trauma related to the periparturition samples. We were cognizant of this potential and sought to minimize handling and undue stress to all pups during harvest. Naturally birthed pups were kept with the dam in the birthing cage until harvest, when pups were removed and quickly euthanized by decapitation, and then immediately snap-frozen in liquid nitrogen. We also made every effort to ensure that the process of C-section harvest did not exceed the physical stresses pups typically experience via natural birth. C-section pups for the time series were maintained on a warming plate, received gentle physical stimulation to simulate maternal interaction, and were similarly euthanized and frozen as rapidly as possible.

We have added the sentence below to be more transparent:

Original text: "It is plausible that rapid changes in transcriptional programs might be physiologically necessary due to the profound differences between the placental and extrauterine environments."

Revised text: "Although we cannot fully rule out technical variables associated with sacrificing pups, we took care to minimize handling and stress prior to euthanasia and immediate snap-freezing, both for naturally and C-section delivered pups. Moreover, it is plausible that rapid changes in transcriptional programs might be physiologically necessary due to the profound differences between the placental and extrauterine environments."

7. The third doublet removal strategy is interesting, albeit rather heuristic. Moreover, it filters out a large fraction of cells (up to 13.2% in some experiments). I have several questions about this approach: i) were cell cycle genes removed prior to performing the analysis?; ii) in step 10, how were 'subclusters showing low expression of the differentially expressed genes identified in step 5' identified? Was this analysis performed in the normalized count space? lii) For clusters of cells that fall along a developmental trajectory, does this approach remove intermediate cells? Depending upon the granularity of the clustering, I could imagine this being a severe problem for continuous cell trajectories. More generally, I would like to see more information about this approach and more detail on the type of cells excluded.

We would like to clarify that the percentage of doublets detected by step 3 is actually lower than suggested in the original submission, since we only reported the range of doublet percentages (0.5-13.2%). The mean percentage of cells detected as doublets in all 16 sci-RNA-seq3 experiments was 8.3% after the first two steps, and 3.4% after step 3 (**Fig. R8a**). However, one experiment (run_13) had a much higher percentage of doublets detected after step 3 (13.2%). The mean percentage of doublets for the other 15 experiments was only 2.5%.

To answer each of the reviewer's questions in turn:

i) Throughout our manuscript, we did not remove cell cycle genes or regress cell cycle index out. However, we carefully checked for any extra cell states that may be driven by specific cell cycle phases each time we performed embedding.

ii) In step 10, subclusters that showed low expression of target cell-partition markers and enriched expression of non-target cell-partition markers were identified as doublet-driven clusters. This was done by manually visualizing gene expression in the UMAP, after the gene expression data was normalized by size factor and log10 transformed.

iii) We agree with the reviewer and acknowledge the potential concern, and believe this is a general concern by any doublets detection strategy based purely on transcriptional profiles, such as *Scrublet*. Below, we highlight several aspects of our approach that we believe at least mitigate this concern:

First, except for three experiments that profiled embryos before E10, which are less heterogeneous, we performed either subclustering or identified differentially expressed genes on each major cell partition, rather than the cell clusters. Cell partitions were detected using the *partitionCells* function implemented in *Monocle/3-alpha*, which applies algorithms that automatically partition cells to learn disjoint or parallel trajectories based on ideas from "approximate graph abstraction" (Wolf et al. 2019). As shown in **Fig. R8b**, cells from six selected experiments were visualized by UMAP before removing doublets by the third strategy. The cells are colored by their partitions, and we can see that the partitions appear to be disjointed in the UMAP embedding, suggesting that cells between the partitions are less likely to be connected during development.

Second, the first two steps were largely automated, with global thresholds set to guide the process. However, the third step required more manual intervention. We identified doublet-driven subclusters by looking for subclusters that met the following two criteria: 1) low expression of target cell partition-specific markers and enriched expression of non-target cell partition-specific markers; 2)

relatively higher doublet scores. As an example, we looked at the sub-clustering result of partition 4 in experiment run_16. We identified one subcluster that expressed high levels of the top 10 marker genes of partition 3. This subcluster also had relatively higher doublet scores. Therefore, we nominated it as a doublet-driven subcluster (**Fig. R9a-c**).

After repeating this approach for individual partitions, we identified doublet-driven subclusters. We then highlighted these subclusters in the original UMAP plot (**Fig. R9d**). We can see that the doublets detected by strategy 3 (5,440 cells, ~0.5% of the whole dataset) are distributed sparsely, rather than enriched at any potential trajectories between partitions.

To provide more details on our approach to the reader, we have revised the **Methods** section regarding the third strategy of detecting doublets by adding more details. We acknowledge that this strategy may potentially exclude cells that are at an intermediate stage of cell state transitions. Additionally, we have provided a script in `step1_Removing_doublets.R` on GitHub that shows the steps we took to detect doublets. This will allow others to replicate and potentially improve on our approach. In the below quote from the revised Methods section, new text is bolded:

Revised Methods: “This step consists of a series of ten substeps. 1) We reduced each cell’s expression vector to retain only protein-coding genes, lincRNAs, and pseudogenes. 2) Genes expressed in fewer than 10 cells and cells in which fewer than 100 genes were detected were further filtered out. 3) The dimensionality of the data was reduced by PCA (50 components) first on the top 5,000 most highly dispersed genes and then with UMAP (max_components = 2, n_neighbors = 50, min_dist = 0.1, metric = 'cosine') using *Monocle/3-alpha*. 4) **Cell clusters were identified in UMAP 2D space using the Louvain algorithm implemented in *Monocle/3-alpha* (resolution = 1e-06). Cell partitions were detected using the *partitionCells* function implemented in *Monocle/3-alpha*. This function applies algorithms that automatically partition cells to learn disjoint or parallel trajectories based on concepts from "approximate graph abstraction" (Wolf et al. 2019).** 5) We took the cell partitions identified by *Monocle/3-alpha* (cell clusters were used instead for three experiments that profiled embryos before E10), downsampled each partition to 2,500 cells, and computed differentially expressed genes across cell partitions with the *top_markers* function of *Monocle/3* (reference_cells=1000). 6) We selected a gene set combining the top ten gene markers for each cell partition (filtering out genes with fraction_expressing < 0.1 and then ordering by pseudo_R2). 7) Cells from each main cell partition were subjected to dimensionality reduction by PCA (10 components) on the selected set of top partition-specific gene markers. 8) Each cell partition was further reduced to 2D using UMAP (max_components = 2, n_neighbors = 50, min_dist = 0.1, metric = 'cosine'). 9) The cells within each partition were further subclustered using the Louvain algorithm implemented in *Monocle/3-alpha* (res = 1e-04 for most clustering analysis). 10) **Subclusters that expressed low levels of the genes that were found to be differentially expressed in step 5, had high levels of markers specific to a different partition, and had relatively high doublet scores, were labeled as doublet-derived subclusters and removed from the analysis. On average, this procedure eliminated 3.4% of cells from each experiment (range 0.5-13.2%) of the cells in each experiment (Supplementary Figs. 24-25).**”

Reviewer Figure 8 (Supplementary Figure 24 in the revised manuscript). Three-step doublet detection workflow for sci-RNA-seq3 experiments. **a**, We performed three steps to detect and remove potential doublets from each single sci-RNA-seq3 experiment. First, we used *Scrublet* to calculate a doublet score for each cell. Cells with a doublet score over 0.2 were annotated as detected doublets. Second, we clustered and subclustered the entire dataset. Subclusters with a detected doublet ratio over 15% were annotated as doublet-derived subclusters. Third, after removing doublets detected by the first two steps, we performed clustering again to identify the major cell partitions (*i.e.* disjoint trajectories). Three experiments (runs 4, 15, and 17) that profiled embryos before E10 used cell clusters instead of cell partitions. We then generated a union gene list by combining the top 10 differentially expressed genes from each cell partition. This gene list was used to perform subclustering on each cell partition. Subclusters that showed low expression of target cell partition-specific markers and enriched expression of non-target cell cluster-partition markers were identified as doublet-driven clusters. More details are provided in the **Methods**. The percentage of cells detected and removed as doublets by each of the three steps in individual sci-RNA-seq3 experiments is shown. **b**, The labeled cell partitions for each of six selected experiments are shown, after removing doublets from the first two steps.

Reviewer Figure 9 (Supplementary Figure 25 in the revised manuscript). Example of detection of doublet-driven subclusters via step 3. a, Re-embedded 2D UMAP of 986,264 cells from experiment run_16, after removing doublets detected in the first two steps. Cells were colored by each of the 12 partitions detected by the *partitionCells* function implemented in *Monocle/3-alpha*. **b**, Re-embedded 2D UMAP of cells from partition 4, with cells colored by subclusters. The same UMAP is shown below, with cells colored by doublet score calculated by *Scublet*. **c**, The same UMAP as in panel b, colored by the normalized gene expression of the top 10 differentially expressed genes in either partition 3 (top) or partition 4 (bottom). **d**, The same UMAP as in panel a, highlighted by doublets detected in step 3 (red).

8. When the data were pooled computationally (pp33-34) was any batch correction strategy applied? From the text, it read as if samples were merged without any additional correction. I do not find the analysis presented in Fig 1d compelling evidence for the lack of a batch effect: I would prefer to see full plots of the embedding for samples processed from adjacent time points in different experiments. Additionally, it would be illustrative to consider the data that were processed in multiple runs.

We did not perform any batch correction in this project for three reasons. First and most importantly, we used sci-RNA-seq3 technology instead of 10X Genomics technology. Sci-RNA-seq3 uses a split-pool barcoding strategy, which means that cells from different samples are pooled together after the first round of barcoding, which minimizes variation between samples within the same experiment. Furthermore, in our experience to date, sci-RNA-seq3 is associated with markedly less batch effects between experiments than data from 10X Genomics (assuming a consistent protocol, which was the case here). Second, since we included samples from roughly adjacent timepoints in most sci-RNA-seq3 experiments, batch variations are confounded with temporal information. Given that we could not find any evidence for batch effects, we erred on the side of preserving temporal information that might be lost during batch correction. Third, we profiled extra nuclei in some sci-RNA-seq3 experiments at a handful of timepoints to ensure sufficient coverage. We used these instances to perform a kNN based analysis, checking for potential batch effects across experiments. In both examples shown in **Supplementary Fig. 3c**, we observed that the nearest neighbors of the cells were overwhelmingly from a different experiment (but the same time window), rather than from the same experiment (but a different time window).

To further check for potential batch effects, as suggested by the reviewer, we generated “full plots of the embedding for samples processed from adjacent timepoints in different experiments”, without any batch correction (**Fig. R10a**). Consistent with our general experience with sci-RNA-seq3 data, we could not discern any batch effects between experiments from these plots. The transcriptional separation of some specific cell clusters was observed for cells from E18.75 vs. P0, as discussed in the paper.

To follow on the second suggestion about data processed in multiple runs, we compared cells from run_23_A and run_23_B, which profiled the same sci-RNA-seq3 experiment but were sequenced on different NovaSeq runs (*i.e.* different cells from the same experiment, sequenced on different runs; **Fig. R10b**). Once again, we could not identify any potential batch effects based on the co-embedding. In both cases (and again, consistent with our prior experience with sci-RNA-seq3), this sharply contrasts with our experience with 10X Genomics, where batch effects are often immediately obvious when such plots are generated without any correction procedures.

We have added the following text (and included these figures in the supplement):

Original text: “Further analyses suggested that batch effects were relatively minimal (**Supplementary Fig. 3b**).”

Revised text: “We did not perform batch correction across experiments, as various analyses suggested that batch effects were relatively minimal. In particular, cells from the same time window but profiled by different experiments, or cells from adjacent timepoints but profiled by different experiments, were well-integrated (**Supplementary Fig. 3b; Supplementary Fig. 4**).”

Reviewer Figure 10 (Supplementary Figure 4 in the revised manuscript). Cells processed in different experiments are well-integrated without batch correction. a, To further check for potential batch effects, we generated co-embeddings of samples processed from adjacent timepoints in different experiments, without batch correction. **b**, We also generated a co-embedding of cells from run_23_A (red) and run_23_B (green), which derived from the same sci-RNA-seq3 experiment but were sequenced on different NovaSeq runs.

9. Methods p34: you mean '12 timepoints at 1 day increments' not '1 hour increments' I think.

Thank you for catching this typo. We fixed it in the revised manuscript.

10. The resolution of clusters in SF4 is relatively low given the number of cells. While I understand the argument of the authors that this will be refined further moving forward, I think a somewhat deeper clustering and annotation would have been helpful for the community and would substantially increase the utility of the dataset.

The annotation of the full dataset took about one year. We agree that deeper clustering and annotation would add more value, but at the cost of significantly more time and the greater likelihood of errors and ambiguity as we go deeper. We also feel that we are at the point where further clustering and annotation requires the identification and engagement of domain-specific experts for each physiological system, which would constitute a major organizational effort to run centrally. Alternatively, this may simply happen on its own as we get the data out there. This possibility is well-stated by Reviewer #2: *"This resource can change the way by which we approach embryonic development of specific lineages and tissues given a much-needed global context. This is possible even given the present form of the data, and will be intensified after much more analysis, tools, integration with spatial data and epigenomics profiles will be added on top of it by the authors (in future studies) and the community."*

Our experience to date has been that each successive publication in this field (from us and others) both adds new data/timepoints, while also refining and advancing annotations from past publications even for earlier timepoints (e.g. this work builds on (Mittnenzweig et al. 2021) and (Qiu et al. 2022), which built on (Pijuan-Sala et al. 2019) and (Cao et al. 2019). Continuing that pattern, we anticipate that followup studies in which we go beyond P0 will also report progress in deepening, refining and correcting prenatal timepoints.

11. Were the authors able to explore the intercalation of cells (e.g., work from Kat Hadjantonakis and others) from the extra-embryonic endoderm when forming the gut-tube (Figure 2)? Some such cells must contribute to the posterior portion of the gut tube.

Thank you for this comment. In a recent publication (Qiu et al. 2022), we re-analyzed several mouse datasets from early embryogenesis, including that of (Pijuan-Sala et al. 2019), which profiled mice from E6.5 to E8.5 with a temporal resolution of 6 hours. In this analysis, we split the cells from each time point and annotated the cell clusters based on their original annotation as well as the expression of specific marker genes. In the figure below, we highlight five cell types: extraembryonic visceral endoderm (*Ttr+*), embryonic visceral endoderm (*Hhex+* and also *Ttr+*, though to a lesser extent than extraembryonic visceral endoderm), definitive endoderm (*Cer1+*), gut (*Apela+*), and notochord (*Noto+*), in UMAPs of embeddings of data from E7.25 to E8.25 (Fig. R11a).

Reviewer Figure 11. Temporal dynamics of visceral endoderm, definitive endoderm, and gut cell states. a, In a recent publication (Qiu et al. 2022), we re-analyzed data from E6.5 to E8.5 embryos from (Pijuan-Sala et al. 2019), by splitting cells by timepoint and labeling according to the original annotations or expression of marker genes for embryonic visceral endoderm (*Hhex*+), extraembryonic visceral endoderm (*Ttr*+), definitive endoderm (*Cer1*+), gut (*Apela*+), or notochord (*Noto*+). **b,** The left panel is reproduced from **Fig. 2j** of the manuscript. Re-embedded 2D UMAP of cells from cluster 3 in **Fig. 2a**. Cells are colored by either their initial annotations (top) or somite counts (bottom). Different subpopulations of gut cells are highlighted by black circles. The same UMAP is shown multiple times on the right, with cells colored by normalized expression of the same marker genes shown in panel a. A subpopulation of early-somitogenesis cells with elevated *Ttr* expression is highlighted by a red circle.

We tracked the changes in cell states over time and found that at E7.25, the visceral endoderm and definitive endoderm were readily distinguishable. From E7.5 to E8.0, the embryonic visceral endoderm and definitive endoderm converged, consistent with the findings of Hadjantonakis and colleagues. By E8.25, the gut was again largely distinct from the (extraembryonic) visceral endoderm, although we did identify *Ttr* gene expression in a subset of gut cells, which we infer likely correspond to the subset that derive from the embryonic visceral endoderm. Overall, these results suggest that the principal window in which intercalation occurs is between E7.5 and E8, with visceral endoderm-derived gut cells potentially expressing higher levels of *Ttr* (**Fig. R11a**).

Turning to the data presented in this manuscript, in our focused analyses of the early gut (**Fig. 2j**), we do observe an early-somitogenesis cell population that expresses elevated levels of *Ttr* relative to other early-somitogenesis cells in this sub-analysis (red circle in *Ttr* sub-panel of **Fig. R11b**). We believe that these cells may correspond to gut cells derived from the visceral endoderm.

12. The result in Fig 2l does not seem especially surprising – AP patterning is expected to be conserved across different developmental trajectories, so I am not sure why the authors find this ‘striking’. I would suggest toning down the language in the text both here and when referring to Fig 2m. This reuse of regulatory programs is well known – it is a nice demonstration but not especially surprising and should be rephrased accordingly.

Thank you for this comment. We agree and have removed three instances “striking” from this section:

Revised text 1: “In comparing genes whose expression patterns are highly correlated with the inferred A-P axis between notochord (PC1; n=591) and gut (PC1; n=502), we observe striking overlap and directional concordance (198 overlapping genes, 86% of which are consistently associated with the inferred anterior or posterior aspect of the notochord and gut; $p < 1e-28$, χ^2 -test; **Fig. 2l**; **Supplementary Table 10**).”

Revised text 2: “A second striking overlap between germ layers involves genes highly correlated with early vs. late somite counts in NMPs (n=257) vs. the gut (PC2; n=502). Once again, we observe striking overlap and directional concordance (82 overlapping genes, 70 (85%) of which are consistently associated with early or late somite counts; $p < 1e-15$, χ^2 -test) (**Fig. 2m**; **Supplementary Table 11**).”

13. Are the manually inferred relationships shown in Figure 3c consistent with those obtained from application of computational approaches?

Below we show the results of two different methods for inferring cell relationships in the developing kidney. The left image shows the results of manual inference (**Fig. R12a**), while the right image shows the results of computational reconstruction using the mutual nearest neighbors (MNN) method (**Fig. R12b**). The overall patterns between the two images are very similar, with three exceptions:

1. With the MNN-method, anterior intermediate mesoderm (AIM) contributes to posterior intermediate mesoderm (PIM), which may reflect an anterior-posterior (A-P) elongation process during mouse early organogenesis.
2. With the MNN-method, AIM contributes to nephron progenitors.

- With the MNN-method, the ureteric bud stalk has been split from the ureteric bud (we made minor changes to some cell annotations while creating the developmental graph), and renal pericytes and mesangial cells (which were originally included in the lateral late & intermediate mesoderm cluster) have been added.

In summary, the overall patterns between the two methods are consistent. The minor differences appear to correspond to subtleties that we did not take into account in our manual reconstruction (#1 above), a potentially incomplete understanding of some lineages (#2 above), or the consequences of annotation changes made while constructing the developmental graph (#3 above).

Reviewer Figure 12. Inferred developmental trajectories between annotated renal cell types. a, This is reproduced from **Fig. 3c**. Manually inferred relationships between annotated renal cell types. Dashed circle highlights posterior & anterior intermediate mesoderm. Dashed line highlights the expected spatial ordering of annotated cell types from proximal (left) to distal (right) aspect of nephron. **b**, A subview of the graph presented in **Fig. 6g**, corresponding to the renal subsystem of mouse development, spanning E0 to P0 (yFiles Hierarchic layout in *Cytoscape/v3.9.1*).

14. Some experimental validation of the tangram inferred location of cell types (Figure 3g) would be important – this is a relatively straightforward experiment, given the marker genes that are available and would support the computational inference. For example, can you validate the expression of markers associated with the two LPM-subsets mapping to the inside and outside of the kidney?

Although we agree that this is a straightforward experiment to conduct anew, publicly available databases of ISH images for mouse embryos contain overlaps with the genes that we would have sought to examine with new experiments. ISH for these and other markers of various mesenchymal subpopulations for which we inferred spatial distributions are available and reproduced below (**Fig. R13**). The ISH images were obtained from www.informatics.jax.org, and the relevant references are provided as part of this figure as well.

Reviewer Figure 13 (Supplementary Figure 12 in the revised manuscript). Published *in situ* hybridization (ISH) images support our annotations of lateral plate and intermediate mesoderm derivatives. In each subpanel (defined by dotted rectangles), three rows are shown for one or two lateral plate and intermediate mesoderm derivative cell types. Notably, each of these cell types was annotated based on spatial mapping analysis, as shown in **Fig. 3g**. Top: The same UMAP as in **Fig. 3f**, colored by gene expression of marker genes which appear specific to the given cell type. Middle: Virtual *in situ* hybridization (ISH) images of individual genes

from one selected section (E1S1) from E14.5 of the *Mosta* data (<https://db.cngb.org/stomics/mosta/>). Bottom: *In situ* hybridization (ISH) images of individual genes were obtained from the Jackson Laboratory Mouse Genome Informatics (MGI) website (<https://www.informatics.jax.org/>). The original references for these images are listed in the middle right of the overall figure.

For example, from our data, we claim that *Pax2* and *Cfh* specify the subsets of organ-specific mesenchyme located outside and inside the kidney, respectively, based on mapping our data to *MOSTA* via *Tangram*. A more detailed explanation of this finding is provided in our response to Comment #4 from Reviewer #1. These predictions are supported by published ISH images, as shown above (bottom left of **Fig. R13**). Several additional examples of ISH-based validation of other subsets of organ-specific mesenchyme are also shown. This figure has been added to the manuscript, and we have revised the following text as follows:

Original text: “Through a combination of spatial inference and marker gene analysis, we were able to assign annotations to 22 subtypes of the LPM & intermediate mesoderm major cell type (**Fig. 3f-g; Supplementary Fig. 8; Supplementary Table 12**).”

Revised text: “Through a combination of spatial inference and marker gene analysis, we were able to assign annotations to 22 subtypes of the LPM & intermediate mesoderm major cell type (**Fig. 3f-g; Supplementary Fig. 11; Supplementary Table 12**). Many of these assignments were supported by publicly available *in situ* hybridization images (**Supplementary Fig. 12**).”

15. How were the solid black lines drawn in Fig 4c? The legend talks about inference, but this feels like a very manual connection, which is challenging given such a complex series of cell states. Can the authors put these relationships on a more quantitative footing?

We apologize for the confusion. In this graph, connections were manually inferred, and are not automated nor quantitative. It is only in a later section that we apply a more systematic approach to generate the comprehensive developmental graph (which includes the eye). In the revision, we have clarified this in the figure legend:

Original legend: “**Fig. 4c**, Schematic of retinal cell types emphasizing the timing at which they first appear and their inferred developmental relationships from E8-P0. The gray lines indicate subsets of the eye field and RPE subsequently annotated as the optic stalk (label 16) and iris pigment epithelium (label 17), respectively. Cell types are positioned along the x-axis at the timepoint at which they are first observed (**Supplementary Fig. 10e**).”

Revised legend: “**Fig. 4c**, Schematic of retinal cell types emphasizing the timing at which they first appear and their inferred developmental relationships from E8-P0, **based on manual review of the trajectories**. The gray lines indicate subsets of the eye field and RPE subsequently annotated as the optic stalk (label 16) and iris pigment epithelium (label 17), respectively. Cell types are positioned along the x-axis at the timepoint at which they are first observed (**Supplementary Fig. 16e**).”

16. Fig 5c: the authors state that the ‘extensive heterogeneity of the patterned neuroectoderm progressively diminishes as differentiating neurons become more similar with respect to their transcriptional states’ – this is hard to observe from the figure – can the authors quantify this observation?

This is a good comment, especially as regardless of whether the claim is correct with respect to the figure shown, distances in UMAP should be interpreted with caution (or not at all). We therefore attempted to apply several different metrics to quantify this. Unfortunately, the results were inconsistent, with some metrics suggesting that transcriptional heterogeneity was decreasing over time (e.g. angular distance in PC space with 30 dimensions) but others suggesting that it was increasing (e.g. Euclidean distance in PC space with 30 dimensions, or Jaccard similarity calculated from Euclidean distance in PC space). As it is not clear to us which of these is the “correct” metric for this kind of a claim, we have deleted this statement from the manuscript.

17. The description of the cell type tree felt like it belonged as a more detailed supplementary note, with more discussion being warranted around a number of factors in the heuristic algorithm. For example, are the MNNs between cell types enriched for cells that are collected from the same embryo? If so, this might suggest a technical factor could be driving a particular edge. More generally, a discussion on parameter choices and robustness of the approach would be welcome; also a discussion around known complex transitions would provide confidence in the result; in particular, in cases where there is cell type intercalation (gut tube) is this observed in the mapping? It would provide confidence in the results.

Thank you for the comments. To assess the robustness of our approach to technical factors and parameter usage, we conducted the following analyses, which we have added as a supplementary note (**Supplementary Note 1**) in the revised manuscript, reproduced below. Note that the similarity of correlation coefficient ranges between the second and third approaches is a coincidence (i.e. we double-checked that this was not an error).

Supplementary Note 1: “To evaluate whether our approach is robust to technical factors or parameter choices, we took the following three approaches. First, we examined whether the MNNs that we identified between different cell types were enriched for cells from the same embryo. Since the data from pre-gastrulation and gastrulation were generated from pooled samples, we only investigated this phenomenon for later stages, i.e. E8-P0 data generated via sci-RNA-seq3. Overall, we found that only 16.4% of MNNs from different cell types were between cells from the same embryo. However, we notably only profiled one embryo for most timepoints, which may inflate this value relative to what it might have been if we had profiled multiple embryos per timepoint. This is supported by the fact that when we look at windows with multiple embryos profiled per timepoint (E8-E10 and E13-E13.75), the proportion of MNNs from different cell types that connect cells from the same embryo was only 10.5% for E8-E10, and only 2.4% for E13-E13.75 (**Supplementary Fig. 27a**). Overall, the fact that MNNs spanning cell types overwhelmingly connect cells from different embryos (and different timepoints) is reassuring.

Second, to assess the robustness of MNNs to cell sampling, we randomly subsampled 80% of cells from each developmental system during organogenesis & fetal development (except for notochord, which is a relatively rare cell type). We then repeated our MNN approach on the subsamples and compared the resulting numbers of MNNs obtained for each edge to those obtained when using the full dataset. This process was repeated 100 times for each developmental system. The resulting correlation coefficients ranged from 0.92 to 0.99, with an average of 0.98 (**Supplementary Fig. 27b**). This suggests that the MNNs we identified are robust to cell sampling.

Third, the k parameter is critical when using kNNs to identify MNNs between cell types. The original k value was selected based on the \log_2 -transformed median number of cells across cell types ($k = 10$ neighbors for pre-gastrulation and gastrulation subsystems, $k = 15$ for organogenesis & fetal development subsystems). To determine the effect of k parameter choice on the MNNs identified between cell types,

we examined different k values ($k = 5, 10, 20, 30, 40, 50$) for k NN to identify MNNs for each developmental system during organogenesis & fetal development. We then compared the results to the original result, which was based on $k = 15$. The resulting Spearman correlation coefficients ranged from 0.92 to 0.99, with an average of 0.98 (**Supplementary Fig. 27c**). This suggests that the MNNs we identified are robust to the choice of k parameter.”

To answer the question about the gut tube, we indeed identified two edges that give rise to the gut in our graph. One of these originates from the definitive endoderm and the other from the embryonic visceral endoderm. Both are part of the gastrulation sub-graph, and were very well supported by the MNN approach (see rows 18/19 and 56/57 of **Supplementary Table 21**). In fact, the edge going back to the embryonic visceral endoderm edge was even more strongly supported than the edge going back to the definitive endoderm. We hope that our automated detection of this complex transition reassures the reviewer that the approach is reasonable.

Reviewer Figure 14 (Supplementary Figure 27 in the revised manuscript). The MNN approach used for graph construction is robust to subsampling and choice of the k parameter. a, The percentage of MNNs between different cell types, from the same embryo (blue) or from different embryos (red), is shown for each developmental system during organogenesis & fetal development, for all cells (left), cells from E8.0 to E10.0 (middle), or cells from E13.0 to E13.75 (right). **b**, The Spearman correlation coefficients of the normalized number of MNNs between cell types, comparing random subsampling of 80% of the cells to the full set of cells. The subsampling was repeated 100 times. The number of MNNs between cell types were normalized by the total number of possible MNNs between them. **c**, The Spearman correlation coefficients of the normalized number of MNNs between cell types, comparing various choices for k parameter ($k = 5, 10, 20, 30, 40, 50$) and the choice of k parameter ($k = 15$) when applying k NN to the developmental systems during organogenesis & fetal

development. The number of MNNs between cell types were normalized by the total number of possible MNNs between them.

18. As an alternative to the cell type analysis presented in Fig 7b, perhaps the authors could consider connecting neighbourhoods of cells? Could the annotation of cell types across this interval be more challenging than across other parts of the developmental trajectory?

Following this suggestion, we attempted an alternative strategy that built a neighborhood graph on cells from each cell type, by connecting cells' neighborhoods based on transcriptional similarity. This was implemented with the *scanpy.pp.neighbors* function. After that, we took the subset of edges that consisted of cells from a given timepoint, and then computed the ratio of those edges that connect cells from the same timepoint or from different timepoints. In this framing, a low proportion of edges from different timepoints corresponds to a relatively abrupt change in transcriptional state. The results for each cell type (**Fig. R15a**) are highly comparable to previous **Fig. 7b**. Although this adds confidence, we decided to stick with the original approach, as we believe that it is more straightforward.

To answer the second question, when we annotated cell types, we carefully checked if the cell state heterogeneity was due to other factors, including (but not limited to) cell cycle phase, A high ratio of reads mapping to the mitochondrial genome, sex, and enrichment of cells from a specific timepoint. For example, the “extra” cell state identified in the kidney subanalysis is enriched with cells from only the P0 sample, but still very clearly expressed marker genes of proximal tubule cells, such as *Lsc27a2* and *Lrp2* (**Fig. R15b**).

Reviewer Figure 15. An alternative strategy to systematically identify which cell types exhibit abrupt transcriptional changes before vs. after birth. a, As an alternative strategy to what was done for **Fig. 7b**, we built a neighborhood graph on cells from each cell type, by connecting cells' neighborhoods based on their transcriptional similarities. This was implemented with the *scanpy.pp.neighbors* function. After that, we took the subset of edges that consisted of cells from a given timepoint, and then computed the ratio of those edges that connect cells from the same timepoint or from different timepoints. In this framing, a low proportion of edges from different timepoints corresponds to a relatively abrupt change in transcriptional state. **b,** This is reproduced from **Supplementary Fig. 10c & d**. Top: The same UMAP as **Fig. 3a** is shown three times, with colors highlighting cells from before E18.75 (left), E18.75 (middle), or P0 (right). Dotted cycles highlight cells which appear to correspond to the proximal tubule. Bottom: The same UMAP as in **Fig. 3a**, colored by expression of marker genes which appear specific to proximal tubule cells (*Slc27a2*+, *Lrp2*+). References for marker genes are provided in **Supplementary Table 5**.

19. The lack of a user-friendly interface for exploring the data is a major hindrance and will substantially limit its utility. This needs to be resolved.

We apologize for our previous lack of care in making our data more accessible. We have taken this feedback to generate a new version of our website that is more user-friendly and accessible to all. Please take a look at the links below:

<https://atlas.gs.washington.edu/jax/>
<https://atlas.gs.washington.edu/jax/public/index.html>

In brief, we have added interactive 3D visualizations to our website, not just for the entire dataset, but also for various subsets (e.g. major cell clusters or the specific subsets that are presented in our main figures). The side-by-side views are synchronized in terms of the viewpoint on the 3D UMAP. Each view can be labeled by cell type, timepoint, or the raw or normalized expression level of any gene of interest.

For subsets with more than 100,000 cells, we used random downsampling to facilitate fast response times for the interactive aspect of the portal. However, each full subset (and/or its metadata) can be easily downloaded for focused analyses by users via links at the bottom of the page that are updated as different subsets are selected.

We plan to continue improving this interactive website and welcome any suggestions. We are also in discussions with CZ about how to make these data available at the CZ “CELL by GENE” data portal.

20. The GitHub repository is incomplete (e.g., TBD for Fig 4 etc) – this needs to be resolved.

Thank you for pointing this out, and we sincerely apologize for this oversight. We have completed the GitHub repository in the course of our revisions. Please take a look at the link below, in which scripts have been split up by section of the manuscript:

https://github.com/ChengxiangQiu/JAX_code

- Section_1_basic_analysis
- Section_2_posterior_embryo
- Section_3_kidney_mesenchyme
- Section_4_eye
- Section_5_neuroectoderm
- Section_6_development_tree
- Section_7_key_TFs
- Section_8_birth_series

For example, there is a script (“step1_Removing_doublets.R”) in the first section that describes how to perform quality control and remove doublets.

On a related point, to address one of this reviewer’s general comments that was not explicitly listed again as a specific comment, we have also expanded the Methods section, in particular adding sections titled “Whole mouse embryo analysis”, “Spatial mapping with Tangram”, “Generating tree of cell types for mouse development”, “Nominating key TFs and genes”, and “Identifying cell types with abrupt transcriptional changes before vs. after birth”.

References

- Cao, Junyue, Malte Spielmann, Xiaojie Qiu, Xingfan Huang, Daniel M. Ibrahim, Andrew J. Hill, Fan Zhang, et al. 2019. "The Single-Cell Transcriptional Landscape of Mammalian Organogenesis." *Nature* 566 (7745): 496–502.
- Costantini, F., and R. Shakya. 2006. "GDNF/Ret Signaling and the Development of the Kidney." *BioEssays: News and Reviews in Molecular, Cellular and Developmental Biology* 28 (2). <https://doi.org/10.1002/bies.20357>.
- Diez-Roux, G., S. Banfi, M. Sultan, L. Geffers, S. Anand, D. Rozado, A. Magen, et al. 2011. "A High-Resolution Anatomical Atlas of the Transcriptome in the Mouse Embryo." *PLoS Biology* 9 (1). <https://doi.org/10.1371/journal.pbio.1000582>.
- Hie, Brian, Hyunghoon Cho, Benjamin DeMeo, Bryan Bryson, and Bonnie Berger. 2019. "Geometric Sketching Compactly Summarizes the Single-Cell Transcriptomic Landscape." *Cell Systems* 8 (6): 483–93.e7.
- Huang, Xingfan, Jana Henck, Chengxiang Qiu, Varun K. A. Sreenivasan, Saranya Balachandran, Rose Behncke, Wing-Lee Chan, et al. 2022. "Single Cell, Whole Embryo Phenotyping of Pleiotropic Disorders of Mammalian Development." *bioRxiv*. <https://doi.org/10.1101/2022.08.03.500325>.
- Kong, Xingxing, Alexander Banks, Tiemin Liu, Lawrence Kazak, Rajesh R. Rao, Paul Cohen, Xun Wang, et al. 2014. "IRF4 Is a Key Thermogenic Transcriptional Partner of PGC-1 α ." *Cell* 158 (1): 69–83.
- Liang, Huiyun, and Walter F. Ward. 2006. "PGC-1 α : A Key Regulator of Energy Metabolism." *Advances in Physiology Education* 30 (4): 145–51.
- Majumdar, Arindam, Seppo Vainio, Andreas Kispert, Jill McMahon, and Andrew P. McMahon. 2003. "Wnt11 and Ret/Gdnf Pathways Cooperate in Regulating Ureteric Branching during Metanephric Kidney Development." *Development* 130 (14): 3175–85.
- Martin, Beth K., Chengxiang Qiu, Eva Nichols, Melissa Phung, Rula Green-Gladden, Sanjay Srivatsan, Ronnie Blecher-Gonen, et al. 2022. "Optimized Single-Nucleus Transcriptional Profiling by Combinatorial Indexing." *Nature Protocols*, October. <https://doi.org/10.1038/s41596-022-00752-0>.
- Mittnenzweig, Markus, Yoav Mayshar, Saifeng Cheng, Raz Ben-Yair, Ron Hadas, Yoach Rais, Elad Chomsky, et al. 2021. "A Single-Embryo, Single-Cell Time-Resolved Model for Mouse Gastrulation." *Cell* 184 (11): 2825–42.e22.
- Pijuan-Sala, Blanca, Jonathan A. Griffiths, Carolina Guibentif, Tom W. Hiscock, Wajid Jawaid, Fernando J. Calero-Nieto, Carla Mulas, et al. 2019. "A Single-Cell Molecular Map of Mouse Gastrulation and Early Organogenesis." *Nature* 566 (7745): 490–95.
- Qiu, Chengxiang, Junyue Cao, Beth K. Martin, Tony Li, Ian C. Welsh, Sanjay Srivatsan, Xingfan Huang, et al. 2022. "Systematic Reconstruction of Cellular Trajectories across Mouse Embryogenesis." *Nature Genetics* 54 (3): 328–41.
- Ransick, Andrew, Nils O. Lindström, Jing Liu, Qin Zhu, Jin-Jin Guo, Gregory F. Alvarado, Albert D. Kim, Hannah G. Black, Junhyong Kim, and Andrew P. McMahon. 2019. "Single-Cell Profiling Reveals Sex, Lineage, and Regional Diversity in the Mouse Kidney." *Developmental Cell* 51 (3): 399–413.e7.
- Rowland, Leslie A., Naresh C. Bal, Leslie P. Kozak, and Muthu Periasamy. 2015. "Uncoupling Protein 1 and Sarcolipin Are Required to Maintain Optimal Thermogenesis, and Loss of Both Systems Compromises Survival of Mice under Cold Stress." *The Journal of Biological Chemistry* 290 (19): 12282–89.
- Schifferl, Dennis, Manuela Scholze-Wittler, Lars Wittler, Jesse V. Veenvliet, Frederic Koch, and Bernhard G. Herrmann. 2021. "A 37 Kb Region Upstream of Brachyury Comprising a Notochord Enhancer Is Essential for Notochord and Tail Development." *Development* 148 (23). <https://doi.org/10.1242/dev.200059>.
- Wolf, F. Alexander, Fiona K. Hamey, Mireya Plass, Jordi Solana, Joakim S. Dahlin, Berthold Göttgens, Nikolaus Rajewsky, Lukas Simon, and Fabian J. Theis. 2019. "PAGA: Graph Abstraction Reconciles Clustering with Trajectory Inference through a Topology Preserving Map of Single Cells." *Genome Biology* 20 (1): 59.

Reviewer Reports on the First Revision:

Referees' comments:

Referee #1 (Remarks to the Author):

The authors have made a good effort to improve the manuscript in many parts. However, the authors have not created the interactive viewer enabling gene-query searching of these data that one would have expected of this group. As one of the development foci of the paper, the authors have chosen the kidney. This is an excellent choice given robust ontologies and understanding of development and the example of other groups for the community benefit of creating gene query sites such as KidneyCellExplorer (<https://cello.shinyapps.io/kidneycellexplorer/>) and KidneyInteractiveTranscriptomics (<http://humphreyslab.com/SingleCell/>). With the kidney focus, let me illustrate a question I would be driven to ask by the authors data and some problems in not following standards in the field for integrating findings here with other efforts.

In the response to the reviewers, the authors highlighted an e12.25dpc sample missing a number of renal cell types (reviewer Figure 5). In looking harder at this data in panel a, I notice that podocytes and proximal tubule cells are both annotated from e10.5 despite the fact their first appearance in the metanephric kidney is not until e13.0-13.5 and the metanephric kidney is only coalescing at e10.0-10.25. Assuming the annotation is correct, the earlier podocytes and proximal tubule annotation may reflect mesonephric tubules, set to degenerate later in development. If so, these would co-express podocyte and proximal tubule markers along with Hox10 paralogs as the most "posterior" Hox genes, while metanephric counterparts may express similar sets of podocyte and proximal tubule gene markers (but likely only similar) together with more posterior Hox11 paralogs. With the expected website addition, I could simply have searched relevant genes amongst the temporal kidney datasets and seen whether this simple explanation held? If the data is to be maximally useful, and science is democratized so that data strives to be as reasonably as accessible as possible, this type of query should be resolved at a good web accessible site.

The authors discuss Foxd1 lineage which is not as abstract as it need be given extensive lineage mapping studies (eg Kobayshi et al. 2014). GUDMAP (www.gudmap.org) has set the community bar for kidney annotation. I understand "mesangial" but terms "pericyte" and "renal stroma" are confusing. There are blood vessels throughout the kidney and stroma in some definitions is everything non-epithelial so wouldn't pericytes be a subset of stroma. The Carroll lab have highlighted the complexity of what has been broadly described as interstitial cells or stroma, much of it descended from Foxd1 progenitor cells, though clearly not all, particularly the smooth muscle progenitor/smooth muscle regions in the deep medullary region, the focus of studies from the Kispert lab. The bottom line here is that the authors could do a much better job with the terms employed for cell clusters and with bringing in insight from the literature.

One minor point in this sentence in "revised text" on page 9, and in several others, the authors use "it" instead of the relevant noun and its confusing and need not be: "However, analysis of genes correlated with Npm1 found it to be declining". An "it" extraction throughout will help the reader.

Referee #2 (Remarks to the Author):

I have no further comments.

Referee #3 (Remarks to the Author):

I would like to thank the authors for the clear and comprehensive way that they have tackled my previous comments. I am satisfied by the responses and now feel that this paper justifies publication in Nature.

Author Rebuttals to First Revision:

Response to Reviewers

We thank Reviewer #1 for their constructive additional feedback on the revised manuscript and website. Below, the original reviewer comments are replicated in full in blue text, while our responses are in black text.

Reviewer #1:

The authors have made a good effort to improve the manuscript in many parts. However, the authors have not created the interactive viewer enabling gene-query searching of these data that one would have expected of this group. As one of the development foci of the paper, the authors have chosen the kidney. This is an excellent choice given robust ontologies and understanding of development and the example of other groups for the community benefit of creating gene query sites such as KidneyCellExplorer (<https://cello.shinyapps.io/kidneycellexplorer/>) and KidneyInteractiveTranscriptomics (<http://humphreyslab.com/SingleCell/>). With the kidney focus, let me illustrate a question I would be driven to ask by the authors data and some problems in not following standards in the field for integrating findings here with other efforts.

We are grateful for this reviewer's positive comments on the revision as a whole, as well as for the additional feedback on the website. In response, we have added additional functionalities to the website (now at <https://omg.gs.washington.edu/>). In particular, the GeneExp tab allows one to query any gene in any single cell type, and to then view bar-plots showing that gene's expression over time (either as log-scaled normalized expression or as % of cells in which the gene was detected), together with the # of cells assigned that cell-type label over time (either as absolute # of cells profiled or estimated proportion of the embryo). The user can also choose to utilize this functionality at 6 hour resolution for the entire time-course (E8 to P0) or 2 hour resolution for late gastrulation (E8 to E10, based on somite counts). As described in response to the next comment, we leverage this new functionality to address the kidney-specific question raised by the reviewer.

We are also working with CZI to have these data deposited to CELLxGENE in such a manner that the same subsets described in our paper and made accessible via our browser will also be explorable via the CELLxGENE browser. Because of the size of the dataset, this is taking some time but they are willing to take the data in this form. However, we anticipate that this will further serve to make the data more accessible/usable by the community, above and beyond our own website.

In the response to the reviewers, the authors highlighted an e12.25dpc sample missing a number of renal cell types (reviewer Figure 5). In looking harder at this data in panel a, I notice that podocytes and proximal tubule cells are both annotated from e10.5 despite the fact their first appearance in the metanephric kidney is not until e13.0-13.5 and the metanephric kidney is only coalescing at e10.0-10.25. Assuming the annotation is correct, the earlier podocytes and proximal tubule annotation may reflect mesonephric tubules, set to degenerate later in development. If so, these would co-express podocyte and proximal tubule markers along with Hox10 paralogs as the most "posterior" Hox genes, while metanephric counterparts may express similar sets of podocyte and proximal tubule gene markers (but likely only similar) together with more posterior Hox11 paralogs. With the expected website addition, I could simply have searched relevant genes amongst the temporal kidney datasets and seen whether this simple explanation held? If the data is to be maximally useful, and science is

democratized so that data strives to be as reasonably as accessible as possible, this type of query should be resolved at a good web accessible site.

Our understanding of your comment is that if your hypothesis is correct, one would expect to see:

* At E10-E11, the cells that we label as podocytes and proximal tubule cells express both their own marker genes as well as Hox10 paralogs but not Hox11 paralogs.

* At E10-E11, the cells that we label as metanephric mesenchyme also express podocyte/proximal tubule marker genes (albeit similar, not identical ones) but also both Hox10 and Hox11 paralogs.

Leveraging the new functionality of the website, we explored this as illustrated in the panels below, all of which are essentially screenshots of views that we pulled up using the GeneExp tab described above.

In **Reviewer Figure 1a**, we highlight the cells that we are focused on, showing the absolute numbers detected and their estimated proportional contribution to the whole embryo over time, for podocytes, proximal tubule, and metanephric mesenchyme.

In **Reviewer Figure 1b**, we show that the early cells assigned as proximal tubule cells express appropriate markers, e.g. *Lrp2* and *Slc27a2*. Interestingly, early podocyte-labeled cells do not express *Nphs1/2* highly, but do express *Synpo* and *Wt1*.

In **Reviewer Figure 1c**, we show that early cells assigned as metanephric mesenchyme express some (e.g. *Lrp2*, *Synpo*, *Wt1*) but not all of these same markers, as do the late cells assigned as metanephric mesenchyme.

In **Reviewer Figure 1d**, we show that the early cells assigned as proximal tubule cells express *Hoxa10* but not *Hoxa11*. Hoxa paralogs are not detected in early cells assigned as podocytes, but again these are much fewer in number (6 cells altogether) than the early cells assigned as proximal tubule cells (see **Reviewer Figure 1a**). In contrast, later cells assigned as proximal tubule cells or podocytes express both *Hoxa10* and *Hoxa11*. In further contrast, both *Hoxa10* and *Hoxa11* are well detected in both early and late subsets of cells assigned as metanephric mesenchyme.

We believe, but are not fully confident that this set of patterns matches the reviewer's hypothesis. But regardless of whether they do or not, we hope that it illustrates how the updated website can be used by the community to explore these kinds of questions.

Reviewer Figure 1. Gene expression profiles of proximal tubule cells, podocytes, and metanephric mesenchyme. **a**, Log₂-scaled cell numbers (top row) and estimated proportional contribution to the whole embryo (bottom row) for podocytes (left), proximal tubule cells (middle), and metanephric mesenchyme (right) are plotted for each 6-hour bin. **b**, Natural-log-scaled normalized expression of selected marker genes specifically expressed in proximal tubule cells (*Slc27a2*, *Lrp2*) and podocytes (*Nphs1*, *Nphs2*, *Synpo*, *Wt1*) are plotted for proximal tubule cells and podocytes, respectively. See titles of each subplot for which gene and cell type is shown. The expression levels are categorized by a 6-hour bin. **c**, Natural-log-scaled normalized expression of the same selected marker genes as shown in panel **b** are plotted for metanephric mesenchyme. See titles of each subplot for which gene and cell type is shown. The expression levels are categorized by a 6-hour bin. **d**, Natural-log-scaled normalized expression of two posterior Hox genes (*Hoxa10* on top row, and *Hoxa11* on bottom row) are plotted for podocytes (left), proximal tubule cells (middle), and metanephric mesenchyme (right). The expression levels are categorized by a 6-hour bin. All plots are screenshots from our website (<https://omg.gs.washington.edu/>).

The authors discuss *Foxd1* lineage which is not as abstract as it need be given extensive lineage mapping studies (eg Kobayshi et al. 2014). GUDMAP (www.gudmap.org) has set the community bar for kidney annotation. I understand “mesangial” but terms “pericyte” and “renal stroma” are confusing. There are blood vessels throughout the kidney and stroma in some definitions is everything non-epithelial so wouldn’t pericytes be a subset of stroma. The Carroll lab have highlighted the complexity of what has been broadly described as interstitial cells or stroma, much of it descended from *Foxd1* progenitor cells, though clearly not all, particularly the smooth muscle progenitor/smooth muscle regions in the deep medullary region, the focus of studies from the Kispert lab. The bottom line here is that the authors could do a much better job with the terms employed for cell clusters and with bringing in insight from the literature.

Thank you for the insightful comments. We re-examined the two cell populations, previously termed renal pericytes & mesangial cells (termed "A") and renal stromal cells (termed "B").

1. Both A and B express *Foxd1*, as well as stromal marker genes including *Prrx1*, *Pdgfra*, and *Pdgfrb*, indicating they are both renal stromal cells derived from *Foxd1*⁺ progenitor cells (**Reviewer Fig. 2a**).
2. Spatial mapping places A in the cortex, and B in the medulla (**Fig. 3d-e**). Moreover, the cap mesenchyme marker gene *Six2* is only expressed in A. Other genes that are highly expressed in A, such as *Eya1* and *Pax2*, are also detected in the cortical region of the kidney by *in situ* hybridization (ISH). Cell populations that overlap in *Foxd1* and *Six2* expression have been discussed in the literature (Kobayashi et al. 2014), but these overlaps seem to occur at early stages of organogenesis (before the onset of ureteric branching), in contrast to our findings, where we identified overlap at much later stages, even close to P0.
3. We identified heterogeneity within B. If you look at the cluster on the right in **Reviewer Fig. 2a-b**, the “bottom” cells express *Foxd1*, which is also expressed in the cortex, while the top cells express *Lrrig1*, which is also expressed in the medulla. This was also verified by published ISH of several other genes that are highly expressed in either cell population. Additionally, the top cells express interstitial cell markers (more specifically, smooth muscle cell markers) *Acta2*, *Pparg*, and *Myh11*. Thus, B indicates a developmental trajectory from cortical stromal cells (*Foxd1*⁺) to medullary interstitial cells (note in time-annotated version of this UMAP, that the the earliest cells (pre-E10) are at the “bottom” of the B cluster, and thus correspond to the cortical *Foxd1*⁺ subset of B).
4. However, the A and B populations are very clearly distinct from one another, even at the earliest stages of the timelapse.
5. Integrating these two cell populations with their potential origins in the mesoderm, we found that A appears to be derived from the intermediate mesoderm and metanephric mesenchyme, while B appears to be derived from the lateral plate mesoderm. However, without lineage tracing studies, we refrain from making any definitive conclusions about the origins of these two subpopulations of renal stromal cells.

In summary, we decided to rename A to "renal cortical stromal cells" and B to "renal medullary stromal cells". We have revised the text as follows:

Original text (initial submission): “We can also distinguish two subsets of LPM-derivatives mapping to the kidney, one to the cortex and the other more heterogeneously, which may correspond to renal stroma and the renal pericytes and mesangial cells, respectively (**Fig. 3g**).”

Original text (first revision): “We can also distinguish two subpopulations of LPM-derivatives mapping to the kidney, one to the cortex and the other more heterogeneously distributed within the renal mesenchyme, which we believe correspond to renal pericytes & mesangial cells and renal stromal cells, respectively. Although both subpopulations express *Foxd1*, supporting their assignment to the kidney, focused analyses are consistent with their having distinct origins (**Supplementary Fig. 13**). However, lineage tracing experiments would be necessary to test this hypothesis. Of note, renal stromal cells exhibited gene expression heterogeneity along what may be the cortical-medullary spatial axis, of genes including *Foxd1* (cortical), *Netrin-1* (cortical) and *Zeb2* (medullary) (**Supplementary Fig. 14**).”

Revised text: “Two subtypes spatially mapped to the kidney, one to the cortex and the other heterogeneously, which we term renal cortical stromal cells and renal medullary stromal cells, respectively (**Fig. 3d-e; Extended Data Fig. 7a-c**). Although both express *Foxd1*⁺, focused analyses suggest distinct origins, with renal cortical stromal cells appearing to derive from the intermediate mesoderm and metanephric mesenchyme, and renal medullary stromal cells appearing to derive from LPM (**Extended Data Fig. 7d-e**). However, lineage tracing experiments would be necessary to conclusively prove this. Of note, renal medullary stromal cells exhibited heterogeneity along what may be a cortical-medullary spatial axis (**Extended Data Fig. 7f**).”

Reviewer Figure 2 (Extended Data Fig. 7 in the revised manuscript). Assessing the potential origins of LPM subsets annotated as renal cortical & medullary stromal cells. **a**, Re-embedded 2D UMAP of 39,468 cells from renal cortical & medullary stromal cells. Cells are colored by either annotation (top) or timepoint

(bottom, after downsampling to a uniform number of cells per time window). **b**, Top: The same UMAP as in panel a, colored by gene expression of marker genes which appear specific to renal cortical & medullary stromal cells. Both cell types express *Foxd1*, *Prrx1*, *Pdgfra*, and *Pdgfrb*, but only renal cortical stromal cells express *Six2*. Middle: Virtual *in situ* hybridization (ISH) images of individual genes. Bottom: ISH images of individual genes. **c**, Top: The same UMAP as in panel a, colored by gene expression of marker genes which appear specific to renal cortical stromal cells (*Eya1+*, *Pax2+*), and renal medullary stromal cells (*Lrriq1+*, *Acta2+*, *Pparg+*, *Myh11+*). Middle: Virtual ISH images of individual genes. Bottom: ISH images of individual genes. **d**, Re-embedded 2D UMAP of 206,908 cells from renal cortical & medullary cells, anterior intermediate mesoderm, posterior intermediate mesoderm, metanephric mesenchyme, and splanchnic mesoderm. Cells are colored by either their initial annotations (left) or timepoint (right, after downsampling to a uniform number of cells per time window). **e**, The average normalized expression of *Foxd1* over time is shown for renal cortical stromal cells (left) and renal medullary stromal cells (right). Gene expression was normalized by the size factor estimated by Monocle/3. **f**, Top: The same UMAP as in panel a, colored by gene expression of marker genes which appear specific to two subsets of renal stromal cells: medullary renal stromal cells (*Zeb2+*, *Picb1+*) and cortical renal stromal cells (*Ntn1+*, *Zbtb7c+*, *Sema3d+*), respectively. Middle: Virtual ISH images of individual genes. Bottom: ISH images of individual genes. In panel b, c, and f, virtual ISH images of individual genes were obtained from one selected section (E1S1) from E14.5 of the *Mosta* data (<https://db.cngb.org/stomics/mosta/>). ISH images were obtained from the Jackson Laboratory Mouse Genome Informatics (MGI) website (<https://www.informatics.jax.org/>). The original reference for these ISH images are (Diez-Roux et al. 2011; Hoffman et al. 2008; Visel, Thaller, and Eichele 2004).

One minor point in this sentence in “revised text” on page 9, and in several others, the authors use “it” instead of the relevant noun and its confusing and need not be: “However, analysis of genes correlated with *Npm1* found it to be declining”. An “it” extraction throughout will help the reader.

We apologize for any confusion. We have revised the text to clarify our meaning.

Original text (first revision): “Other genes such as *Npm1* and *Hsp90* isoforms are plausibly associated with batch effects. However, analysis of a module of genes correlated with *Npm1* found it to be declining with developmental time across the entire time series, rather than correlated with batch variables.”

Revised text: “Other genes such as *Npm1* and *Hsp90* isoforms are plausibly associated with batch effects. However, analysis of a module of genes correlated with *Npm1* revealed that this module declined with developmental time across the entire time series, rather than being correlated with batch variables.”

References

- Diez-Roux, G., S. Banfi, M. Sultan, L. Geffers, S. Anand, D. Rozado, A. Magen, et al. 2011. “A High-Resolution Anatomical Atlas of the Transcriptome in the Mouse Embryo.” *PLoS Biology* 9 (1). <https://doi.org/10.1371/journal.pbio.1000582>.
- Hoffman, Brad G., Bogard Zavaglia, Joy Witzsche, Teresa Ruiz de Algora, Mike Beach, Pamela A. Hoodless, Steven J. M. Jones, Marco A. Marra, and Cheryl D. Helgason. 2008. “Identification of Transcripts with Enriched Expression in the Developing and Adult Pancreas.” *Genome Biology* 9 (6): R99.
- Kobayashi, Akio, Joshua W. Mugford, A. Michaela Krautzberger, Natalie Naiman, Jessica Liao, and Andrew P. McMahon. 2014. “Identification of a Multipotent Self-Renewing Stromal Progenitor Population during Mammalian Kidney Organogenesis.” *Stem Cell Reports* 3 (4): 650–62.
- Visel, Axel, Christina Thaller, and Gregor Eichele. 2004. “GenePaint.org: An Atlas of Gene Expression Patterns in the Mouse Embryo.” *Nucleic Acids Research* 32 (Database issue): D552–56.